# Reconstruction of proto-vertebrate, proto-cyclostome and proto-gnathostome genomes provides new insights into early vertebrate evolution

Yoichiro Nakatani [1,4,5], Prashant Shingate [2,5], Vydianathan Ravi [2], Nisha E. Pillai[2], Aravind Prasad [2], Aoife McLysaght [1 ✉] & Byrappa Venkatesh [2,3 ✉]

Ancient polyploidization events have had a lasting impact on vertebrate genome structure, organization and function. Some key questions regarding the number of ancient polyploidization events and their timing in relation to the cyclostome-gnathostome divergence have remained contentious. Here we generate de novo long-read-based chromosome-scale genome assemblies for the Japanese lamprey and elephant shark. Using these and other representative genomes and developing algorithms for the probabilistic macrosynteny model, we reconstruct high-resolution proto-vertebrate, proto-cyclostome and proto-gnathostome genomes. Our reconstructions resolve key questions regarding the early evolutionary history of vertebrates. First, cyclostomes diverged from the lineage leading to gnathostomes after a shared tetraploidization (1R) but before a gnathostome-specific tetraploidization (2R). Second, the cyclostome lineage experienced an additional hexaploidization. Third, 2R in the gnathostome lineage was an allotetraploidization event, and biased gene loss from one of the subgenomes shaped the gnathostome genome by giving rise to remarkably conserved microchromosomes. Thus, our reconstructions reveal the major evolutionary events and offer new insights into the origin and evolution of vertebrate genomes.

[1] Smurfit Institute of Genetics, Trinity College Dublin, University of Dublin, Dublin 2, Ireland. [2] Comparative and Medical Genomics Laboratory, Institute of Molecular and Cell Biology, A*STAR, Biopolis, Singapore, Singapore. [3] Department of Paediatrics, Yong Loo Lin School of Medicine, National University of Singapre, Singapore, Singapore. [4] Present address: Graduate School of Medicine, Osaka University, Osaka, Japan. [5] These authors contributed equally: Yoichiro Nakatani, Prashant Shingate. ✉email: aoife.mclysaght@tcd.ie; mcbbv@imcb.a-star.edu.sg

The emergence of morphologically complex vertebrates from invertebrate chordates is considered a major evolutionary transition that led to the emergence of more than 70,000 vertebrate species (http://vgpdb.snu.ac.kr/splist), including humans. The common ancestor of vertebrates that originated during the Lower Cambrian[1] diverged to give rise to the two extant lineages of vertebrates, the cyclostomes (jawless vertebrates) and gnathostomes (jawed vertebrates). Cyclostomes are a monophyletic group[2] comprising lampreys and hagfishes, while gnathostomes include cartilaginous fishes (Chondrichthyes, represented by chimaeras, sharks and rays) and bony vertebrates (Osteichthyes, represented by ray-finned fishes and lobe-finned fishes, including tetrapods). Cyclostomes are sometimes thought to be morphologically primitive as compared to gnathostomes as they lack hinged jaws, paired appendages and nostrils, mineralized tissue, and a discrete pancreas[3–5]. However, recent studies suggest that the lamprey and hagfish lineages independently acquired their seemingly simplified as well as their specialized morphology, and that the ancestral cyclostome already had a complex morphology and physiology distinct from the gnathostome lineage[6,7]. For example, although cyclostomes lack the major histocompatibility complex and immunoglobulin-based adaptive immune system (AIS) of gnathostomes, they have independently evolved somatically diversifying variable lymphocyte receptors for antigen recognition[8].

Evolutionary innovations at the origin of vertebrates have been proposed to be the result of ancient tetraploidization events that generated additional copies of the entire genome[9,10]. This view is now widely accepted because genome-wide synteny and paralogy analyses[11–15] have provided convincing evidence for two rounds of tetraploidization (known as 1R and 2R, respectively) during early vertebrate evolution (see for review refs. [16,17]; see also refs. [18,19]). However, the timing of 1R and 2R relative to the cyclostome–gnathostome divergence has remained contentious—a significant gap in our knowledge considering the important implications for the genetic basis of the shared and derived features of these two lineages. Previous studies have produced conflicting results supporting each of the three possibilities[18–26], i.e. divergence occurring prior to 1R, between 1R and 2R, and after 2R (Fig. 1). This uncertainty has been further compounded by the discovery of six Hox clusters in both lampreys and hagfish[19,22,27] compared to four clusters in most gnathostome lineages, suggesting the possibility of an additional tetraploidization or chromosome-scale segmental duplications in the cyclostome ancestor[18,19,22].

Resolving these alternative scenarios using gene trees has proved to be challenging. This is partly due to the presence of multiple 'ohnologues' (paralogous genes generated by polyploidy) created by successive rounds of tetraploidization; lineage-specific secondary losses of some ohnologues[28,29]; as well as the confounding effects of asymmetric evolutionary divergence between ohnologues[30]. The possibility of delayed rediploidization after a tetraploidization event[28,31,32] has further complicated the interpretation of gene trees, as it uncouples gene duplication time from the divergence time of the ohnologues. In addition, the tendency of lamprey ohnologues to cluster outside gnathostome gene clades due to high GC-content and consequent codon bias[22,26,33] has impeded the use of gene trees for determining the timing of 1R and 2R.

An alternative and more effective strategy for the identification of ancient polyploidy is the macrosynteny-based reconstruction of ancestral genomes[12–15]. In particular, this strategy has the potential to reveal chromosome fusion/fission events that occurred in the interval between 1R and 2R and/or after 2R[13,15]. Whether such genome rearrangements are shared by cyclostomes and gnathostomes would be potentially informative for determining the timing of the cyclostome–gnathostome divergence in relation to 1R and 2R. A prerequisite for such comparisons is the high-resolution reconstruction of the proto-cyclostome and the proto-gnathostome genomes which require high-quality, chromosome-scale genome assemblies from the most basal vertebrate lineages such as cyclostomes and cartilaginous fishes.

In the present study, we generate de novo chromosome-scale genome assemblies of a cyclostome, the Japanese lamprey (*Lethenteron japonicum*; also known as the Arctic lamprey *Lethenteron camtschaticum*) and a cartilaginous fish, the elephant shark (*Callorhinchus milii*), based on long single-molecule reads and chromatin conformation capture (Hi-C) data. These two species represent two crucial divergence points in the evolution of vertebrates (Fig. 1). We use our recently developed probabilistic macrosynteny model[34] to reconstruct the proto-vertebrate and proto-cyclostome genomes. The major advantage of our method is that it has a high tolerance to reconstruction uncertainty caused by small-scale rearrangements that have accumulated over a long evolutionary time[34]. Using our strategy, we are able to reconstruct the proto-cyclostome genome, in which we integrate information from the Japanese lamprey genome, the sea lamprey genome and the Pacific lamprey linkage markers[19]. In addition, using the elephant shark genome, we reconstruct the proto-gnathostome genome with a higher coverage of extant gnathostome genomes than previous reconstructions (including 19,343 human genes as compared to 12,137 human genes in ref. [13], and 8,434 human genes in ref. [15]).

Our high-resolution reconstructions resolve the number and timing of polyploidization events during early vertebrate evolution and provide new insights into the genetic basis underlying evolutionary innovations during the origin of early vertebrates. In addition, our reconstructions serve as a reliable reference for accurate annotation of ohnologues, which will be especially important for ohnologues with low sequence similarity[30] which are difficult to identify by standard approaches.

## Results

**Genome sequencing, assembly and annotation.** We generated de novo chromosome-scale genome assemblies for elephant shark and Japanese lamprey using a combination of PacBio single-molecule real-time (SMRT) sequencing (68- and 87-fold coverage, respectively), and 'Chicago'[35] and Hi-C data aided scaffolding (see Supplementary Note 1). The resultant genome assemblies of elephant shark and Japanese lamprey span 991 Mb (N50 contig, 1.6 Mb and N50 scaffold, 69 Mb) and 1.07 Gb (N50 contig, 1.6 Mb and N50 scaffold, 10.7 Mb), respectively. These assemblies contain a substantially higher amount of repetitive sequences (42 and 50%) compared to the previous short-read assemblies of elephant shark (28%)[36] and Japanese lamprey (21%)[22], presumably due to the higher contiguity of the long-read assemblies. Using the MAKER pipeline (v2.31.8)[37] and evidence-based and ab initio gene predictions, we predicted 18,747 protein-coding genes in the elephant shark genome assembly and 19,455 protein-coding genes in the Japanese lamprey genome assembly, respectively.

**Reconstruction and validation of the proto-vertebrate genome.** We reconstructed the proto-vertebrate genome structure by employing the probabilistic macrosynteny model[34] and comparing the Japanese lamprey, sea lamprey (*Petromyzon marinus*)[19], amphioxus (*Branchiostoma floridae*)[14], and four gnathostome genomes including human, chicken[38], spotted gar[39] and elephant shark (see 'Methods'). In our reconstruction procedure, the lamprey genomes were partitioned into segments of conserved

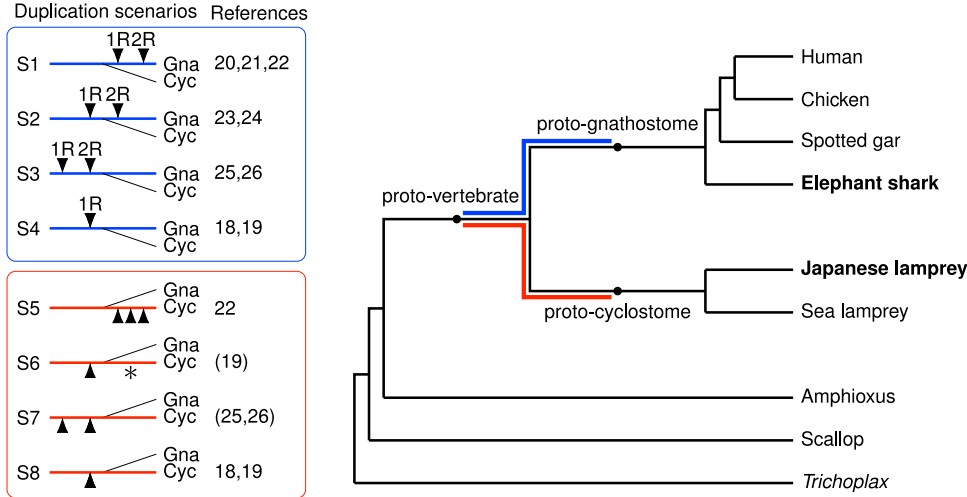

**Fig. 1 Phylogenetic tree and previously proposed whole-genome duplication scenarios.** Scenarios S1–S4 show the proposed timing of tetraploidization events occurring in the interval between the proto-vertebrate genome and the proto-gnathostome genome. Scenarios S5–S8 show polyploidization events between proto-vertebrate and proto-cyclostome. Triangles and asterisks indicate tetraploidization and hexaploidization events, respectively. The possibility of cyclostome-specific hexaploidization (S6) was discussed briefly in ref. [19]. Reference [25] discussed the lack of clear evidence for cyclostome-specific polyploidization, and ref. [26] reported the absence of recent polyploidization events in the cyclostome lineage (S7). References [18,19] argued that chromosome duplications and segmental duplications occurred after 1R (S4 and S8). Reference [24] suggested one shared tetraploidization before the gnathostome–cyclostome split and two lamprey-specific tetraploidization events after the hagfish–lamprey split. It is presently considered that the hagfish and lamprey lineages share the same duplication history[27], but this argument should eventually be confirmed by sequencing the hagfish genome. Few previous studies discussed both the timing of gnathostome-cyclostome split and the possibility of cyclostome-specific polyploidization events, because cyclostome genomes were unavailable until recently. Gna gnathostome, Cyc cyclostome.

macrosynteny (191 Japanese lamprey segments and 198 sea lamprey segments), where synteny breakpoints were detected using the Japanese lamprey, sea lamprey, elephant shark, spotted gar, chicken and human genomes. Then, our Bayesian inference algorithm reconstructed the proto-vertebrate genome, assuming that individual proto-vertebrate chromosomes (Pvcs) have distinct orthologue distributions over the lamprey segments[34]. The reconstructed proto-vertebrate genome comprises 18 putative chromosomes (designated as Pvc1–18, with Pvc18 exhibiting only weak macrosynteny conservation in the amphioxus genome) and is largely consistent with previous reconstructions with 17 putative chromosomes[14,15] (Supplementary Fig. 3 and Supplementary Table 9 in Supplementary Note 3).

As a validation of our reconstruction we examined conserved macrosynteny with representative invertebrate and gnathostome genomes including the scallop *Chlamys farreri*[40], the placozoan *Trichoplax adhaerens*[41], human and elephant shark (Fig. 2, also see Supplementary Fig. 4). These lineages have been shown to possess relatively slow rates of genome structure evolution[12,36,41–43]. We therefore expect that a reliable reconstruction should show a highly non-random distribution of orthologues in these genomes. Indeed, we noted that the orthologues are not randomly scattered throughout the modern genomes, but are clustered into a small number of chromosomes (evident as concentration of blue dots in Fig. 2). The fact that we find strong macrosynteny conservation in these invertebrate genomes, which were not used in the proto-vertebrate reconstruction, supports the validity of our reconstruction and indicates that all 18 reconstructed chromosomes existed as separate chromosomes in early metazoan lineages.

**Reconstruction of the proto-cyclostome chromosomes and evidence for sixfold duplication of the genome.** The generation of a long-read-based high-quality genome assembly for the Japanese lamprey, in addition to the existing 'hybrid' genome assembly of the sea lamprey[19], permitted us to investigate unresolved issues in cyclostome genome evolution. In particular, the

evolutionary steps between the proto-vertebrate and proto-cyclostome genomes have remained contentious, even after the sequencing of the sea lamprey genome[18,19,26]. For example, the presence of six Hox clusters in two species of lampreys and the inshore hagfish could be due to more than two rounds of tetraploidization (S5 in Fig. 1); alternatively, they could be the result of a single tetraploidization event followed by chromosome duplication events (S8 in Fig. 1). Another possibility is that the cyclostome lineage experienced a hexaploidization event (whole-genome triplication) in addition to a tetraploidization (whole-genome duplication) event (S6 in Fig. 1). These alternative evolutionary models have been discussed in previous studies[18,19,22,26] but remained unresolved even with the chromosome-level assembly of the sea lamprey genome.

In the present study, we have generated the first reconstruction of the proto-cyclostome genome by combining lamprey segments (described in the previous section) into 104 proto-cyclostome chromosomes (see 'Methods' for details). Our algorithm enumerates possible combinations of lamprey segments (see Supplementary Movie 1), and reconstructs proto-cyclostome chromosomes by choosing the combination with the most significant (i.e. non-random) distribution of paralogues and orthologues (partly illustrated in Fig. 3a–c). Importantly, the algorithm explores all alternative models including segmental duplications, chromosome duplications/losses, tetraploidization and hexaploidization, under the assumption that duplicated chromosomes share significantly large numbers of paralogues. The major advantage of this reconstruction method is its robustness against lineage-specific rearrangements and fragmentation of genome assemblies. For example, Japanese lamprey Scaffold2 was partitioned into two segments (Fig. 3a) because each of the segments showed conserved synteny with two different sea lamprey scaffolds; in our reconstruction (Fig. 3b), the two segments on Scaffold2 were assigned to different proto-cyclostome chromosomes because they share a significantly large number of paralogues (dots in Fig. 3c). Thus, our reconstruction-based analysis is more reliable

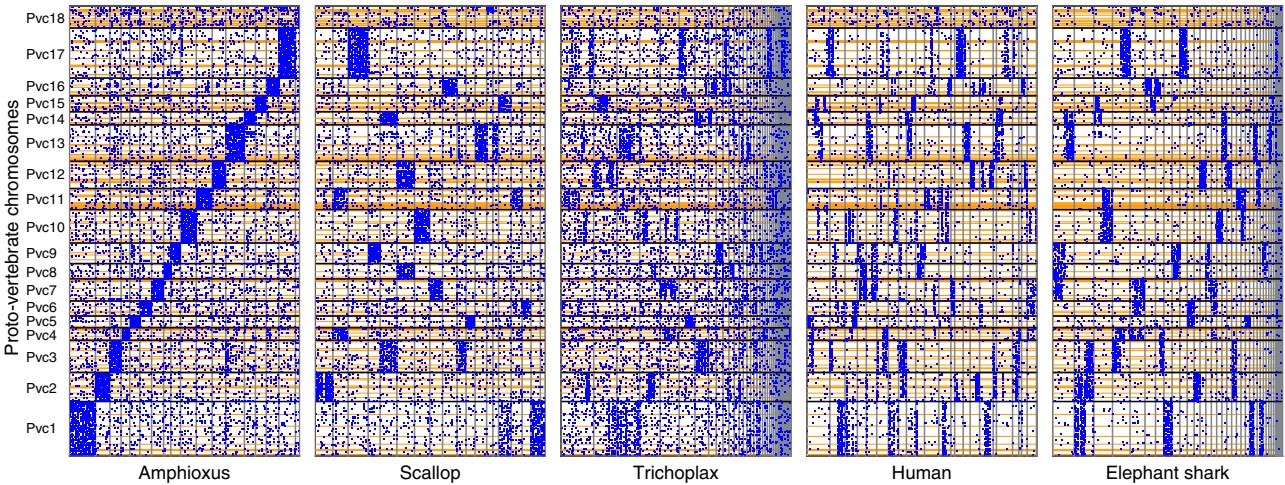

**Fig. 2 Each of the reconstructed proto-vertebrate chromosomes exhibits a distinct orthologue distribution in invertebrate and gnathostome genomes.** The reconstructed proto-vertebrate chromosomes, as represented by the Japanese lamprey segments (y-axis), were compared with the amphioxus, scallop, *Trichoplax*, humans, and elephant shark genomes (x-axes), and their orthologues are plotted (blue dots). The proto-vertebrate chromosomes (Pvc1–Pvc18) are shown from bottom to top along the y-axis, and their boundaries are indicated by horizontal black lines. The boundaries of Japanese lamprey segments are indicated by the horizontal orange lines. The amphioxus scaffolds are organized into 18 groups representing the proto-vertebrate chromosomes. The scallop chromosomes are shown from chr1 to chr19. The *Trichoplax* and elephant shark scaffolds were sorted by the number of genes and the largest 50 scaffolds are shown. Human chromosomes are shown from chr1 to chrY.

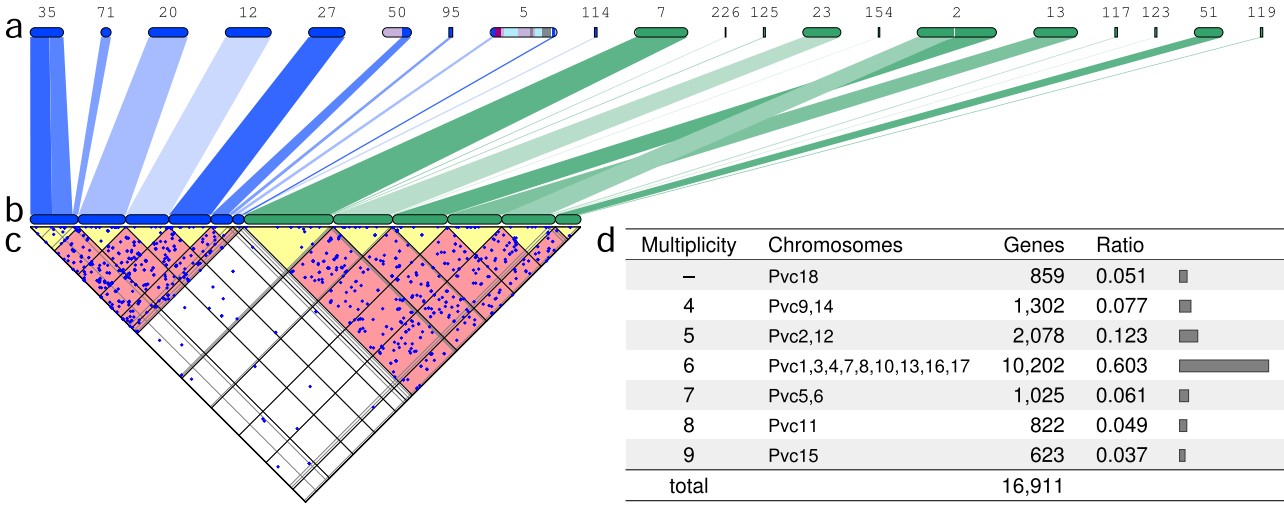

| Multiplicity | Chromosomes | Genes | Ratio | |
|---|---|---|---|---|
| – | Pvc18 | 859 | 0.051 | |
| 4 | Pvc9,14 | 1,302 | 0.077 | |
| 5 | Pvc2,12 | 2,078 | 0.123 | |
| 6 | Pvc1,3,4,7,8,10,13,16,17 | 10,202 | 0.603 | |
| 7 | Pvc5,6 | 1,025 | 0.061 | |
| 8 | Pvc11 | 822 | 0.049 | |
| 9 | Pvc15 | 623 | 0.037 | |
| total | | 16,911 | | |

**Fig. 3 Reconstruction of the proto-cyclostome chromosomes and evidence of sixfold duplication of the genome. a** Japanese lamprey scaffolds are illustrated with the scaffold IDs. These scaffolds were partitioned into segments of conserved synteny, and segments corresponding to proto-vertebrate chromosome Pvc3 (blue) and Pvc17 (green) are shown for illustrative purposes. **b** Groups of segments of the same colour were organized into several subgroups representing proto-cyclostome chromosomes based on the distribution of paralogues and orthologues. **c** The triangular plot is a 45°-rotated graph of the paralogue distribution between the 12 proto-cyclostome chromosomes that correspond to Pvc3 and Pvc17. This shows large numbers of paralogues between the chromosomes (dots in the red regions) but few paralogues within each chromosome (dots in the yellow regions). **d** The table classifies the proto-vertebrate chromosomes with respect to the number of duplicated proto-cyclostome chromosomes as shown in the 'Multiplicity' column (Pvc18 was retained as a single proto-cyclostome chromosome because Pvc18 had too many segments for the reconstruction algorithm). The table also shows the numbers and ratios of Japanese lamprey genes that were mapped to the proto-cyclostome chromosomes originating from the proto-vertebrate chromosomes shown in the 'Chromosomes' column.

than scaffold-based analyses used in previous studies[18,19,26] and provides the first opportunity to conclusively resolve the controversy over the origin of the proto-cyclostome genome.

To distinguish between alternative polyploidization models (i.e. S5–S8 in Fig. 1), we followed ref. [13] and used a measure we have called multiplicity, i.e. the number of proto-cyclostome chromosomes originating from individual proto-vertebrate chromosomes (Fig. 3d), and counted the numbers of Japanese lamprey genes that map to these chromosomes. If the proto-cyclostome genome was shaped by three rounds of tetraploidization (S5 in Fig. 1), it

should be covered by chromosomes of multiplicity eight. Instead, if it experienced a single tetraploidization with subsequent chromosomal duplications (S8 in Fig. 1), the multiplicity should peak at two with gradual decrease toward larger multiplicities. The third possibility is that if the genome went through a single tetraploidization and a hexaploidization (genome triplication) (S6 in Fig. 1) the majority of the genome should be covered by chromosomes of multiplicity six. Our analysis indicates that 9 out of the 18 proto-vertebrate chromosomes were duplicated into six paralogous proto-cyclostome chromosomes, and that the

majority (60.3%) of the proto-cyclostome genome was covered by the sixfold duplicated chromosomes. In addition, we confirmed by statistical test (see 'Methods') that the observed peak of multiplicity (Fig. 3d) is unlikely to have been created by accumulation of chromosome scale or segmental duplications after one ($P < 4 \times 10^{-5}$) or two ($P < 0.05$) tetraploidization events. Thus, the clear peak at multiplicity of six is compelling evidence of sixfold duplication of the entire genome, probably through a tetraploidization and a hexaploidization event.

Although the current lamprey genomes might still be incomplete and some chromosomes might be fragmented, such limitations are unlikely to have substantially biased our analysis. First, if the proto-cyclostome genome was shaped by three rounds of tetraploidization, that would additionally require a large number of subsequent chromosome fusions to explain the current genome arrangement (e.g., 45 post-tetraploidization fusions are required to obtain the chromosome number of sea lamprey germline cells: $18 \times 8 - 45 = 99$). However, we found that the lamprey lineage had remarkably low rates of inter-chromosomal rearrangement (Supplementary Fig. 5) over ~500 million years[44] of cyclostome evolution. Specifically, our proto-cyclostome genome reconstruction shows large-scale fusions and translocations affecting only 22 out of 141 Japanese lamprey scaffolds and only 19 out of 151 sea lamprey scaffolds that have at least 10 genes. The exceptionally low rate of inter-chromosomal rearrangement and the haploid chromosome number of ~99 in the germline sea lamprey genome[45] are consistent with our evolutionary scenario in which the lamprey chromosome number is explained approximately as $18 \times 6 = 108$ with several subsequent fusions. Second, even though some tiny chromosomes might be missing in the current proto-cyclostome reconstruction, large chromosomes (e.g. Hox-bearing chromosomes duplicated from Pvc1) are unlikely to be missing entirely; therefore, our reconstruction is particularly reliable for the largest five proto-vertebrate chromosomes (i.e. Pvc1, 3, 10, 13 and 17), which consistently exhibited a multiplicity of six. Thus, the high coverage (60.3%) of the Japanese lamprey genome by sixfold duplicated proto-cyclostome chromosomes suggests that extant cyclostome genomes are paleo-dodecaploids (i.e. the chromosome number increased as $18 \times 6$ due to tetraplodization and hexaploidization), which might be similar to the situation in sturgeon where a species (*Acipenser brevirostrum*) with ~180 chromosomes is considered to be a hexaploid of a tetraploid ancestor with ~60 chromosomes[46–48].

**Proto-gnathostome genome and the origin of microchromosomes**. Previous reconstructions of the proto-gnathostome genome[12–15] included members of only bony vertebrates (Osteichthyes) and lacked representatives of its sister group, the cartilaginous fishes (Chondrichthyes). Here, we produced a substantially improved reconstruction of the proto-gnathostome genome structure with a higher coverage of modern genomes by taking advantage of our newly sequenced, chromosome-scale genome assembly of the elephant shark, in addition to the spotted gar, zebra finch, turkey, chicken, opossum, dog, mouse and human genomes (see 'Methods', Fig. 4 and Supplementary Fig. 6). The reconstruction provided additional support for the previous finding of two rounds of tetraploidization between the proto-gnathostome and its invertebrate ancestor[11–15].

Analysis of this proto-gnathostome genome also revealed the origin of microchromosomes found in some modern gnathostomes. Microchromosomes are tiny chromosomes (typically smaller than 20 Mb), characterized by high GC-content, high gene density and high recombination rate[38,49]. Although there are no microchromosomes in the human genome, they are present in

other tetrapod lineages such as birds and reptiles. Whether microchromosomes were recently created by chromosome fission or were present in the gnathostome ancestor has been controversial (see Supplementary Note 4 for a short review). Although several recent studies supported the ancient origin of microchromosomes[13,36,39,49–52], it was still unknown (1) if chromosomal features characteristic to modern avian microchromosomes (i.e. high GC-content, high gene density and high recombination rate) were already present in the ancestral gnathostome genome (cf. the chromosomal features were previously reported to be conserved between the spotted gar and chicken genomes[39]), and (2) why microchromosomes have been conserved in distantly related gnathostome species such as the chicken, spotted gar and elephant shark.

In the present study, our reconstruction shows that at least 15 proto-gnathostome chromosomes have remained intact as microchromosomes in some modern gnathostome genomes such as chicken, spotted gar and elephant shark (Supplementary Fig. 7) even after ~450 million years of gnathostome evolution[44,53]. Furthermore, we observed that specific sequence features (namely, chromosome length and gene density) are shared by modern gnathostome chromosomal regions that were derived from such proto-gnathostome chromosomes (Fig. 5). First, the total length of segments originating from individual proto-gnathostome chromosomes is highly conserved in chicken, spotted gar and elephant shark, suggesting that the ancestral gnathostome already possessed the tiny microchromosomes and the large macrochromosomes (Fig. 5a). Second, smaller proto-gnathostome chromosomes tend to have higher gene densities in all species, suggesting that the ancestral gnathostome genome consisted of small chromosomes with high gene densities and large chromosomes with low gene densities (Fig. 5b). Third, smaller proto-gnathostome chromosomes tend to have higher ohnologue densities in individual species, suggesting that the ancestral gnathostome genome had small chromosomes with high ohnologue densities (Fig. 5c). These observations suggest that many of the proto-gnathostome chromosomes might have already exhibited distinctive features (e.g. diminutive chromosomes with high gene density) that are considered characteristics of avian microchromosomes[38,49]. Thus, the proto-gnathostome lineage might have already possessed many microchromosomes with high gene density, many of which are still retained in several modern gnathostome genomes due to low rates of inter-chromosomal rearrangement. On the other hand, macrochromosomes, large genome sizes and high rates of rearrangement are likely to be derived characteristics of lineages that experienced substantial expansion of repetitive sequences.

The persistence of intact microchromosomes in modern gnathostome genomes is intriguing, and raises questions about the possible mechanism and evolutionary forces maintaining them over such a long evolutionary time[49]. One possibility is the presence of a high density of genomic regulatory blocks (GRBs) comprising long-range interacting regulatory elements and/or topologically associating domains (TADs) that require long-range linkage to be maintained intact. To test this possibility we analysed the density of GRBs and TADs[54] and observed no obvious difference between macrochromosomes and microchromosomes (see Supplementary Fig. 8 and Supplementary Note 4). An alternative possibility is that the persistent synteny conservation is a by-product of the small size and high gene density of microchromosomes[49], which is corroborated by previous arguments that gene density and ohnologue density are major factors in decreasing the rates of evolutionary breakage[55] and inter-chromosomal rearrangement[56], respectively. Consistent with this hypothesis, we find evidence for high density of genes (including ohnologues, which were identified with the method described in

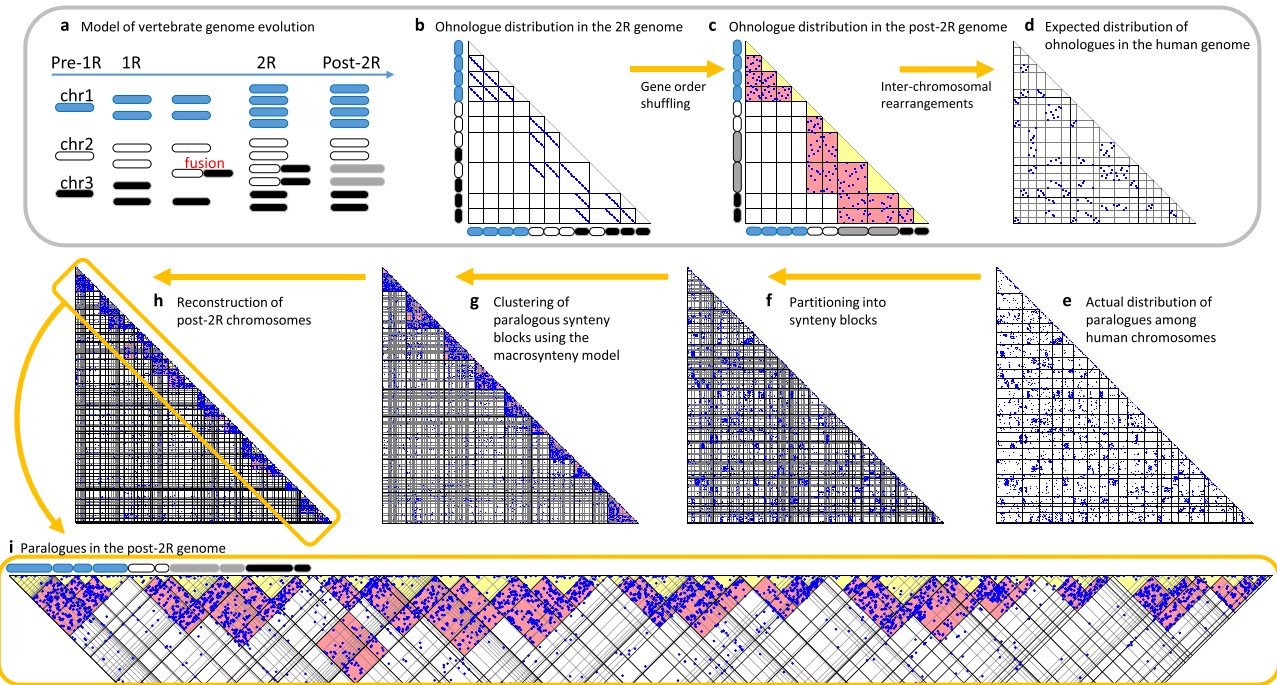

**Fig. 4 Model of vertebrate genome evolution and reconstruction of proto-gnathostome chromosomes. a** Hypothetical scenario of an ancestral genome of three chromosomes that experiences two rounds of WGD (labelled 1R and 2R) and rearrangements. **b** Scatter plot of ohnologues created by the two WGD events. The hypothetical 2R genome is along each axis. Dots represent ohnologue pairs whose genome positions are indicated by the *x*- and *y*-axes. **c** Inversions result in extensive shuffling of gene order within a chromosome, scattering the dots over the entire chromosome. Red rectangles indicate regions that contain many ohnologue gene pairs between duplicated chromosomes. Yellow triangles represent intra-chromosome comparisons and are empty. **d** Inter-chromosomal rearrangements shuffle the genome further. The aim of our reconstruction analysis is to infer the post-2R genome structure (**c**) from modern genomes (**d**). **e** As expected, the actual ohnologue distribution in the human genome is similar to Panel **d**. Lines indicate chromosome boundaries, and human chromosomes 1–22, *X* and *Y* are ordered along the axes. **f** The human chromosomes were partitioned into 151 segments of conserved synteny based on comparison with other gnathostome genomes. **g** The segments were grouped into 10 clusters (red triangular regions) using the probabilistic macrosynteny model. Segments in individual clusters are expected to derive from a single pre-1R chromosome (no fusions), or alternatively derive from multiple pre-1R chromosomes if they involve fusions. **h** Optimal partitioning of each cluster of human segments into several paralogous subgroups (yellow triangles of low ohnologue density, representing 49 post-2R chromosomes). **i** Magnification of the diagonal part of **h**. Examples of fusion presence/absence similar to panel **a** are present, indicated by the chromosome cartoons above the plot shaded as per panel **a**. For example, the leftmost group of segments make up the Hox-bearing chromosomes and have not experienced fusion (blue chromosomes).

Supplementary Note 2) in the proto-gnathostome chromosomes that gave rise to the modern microchromosomes (Fig. 5b, c).

**Timing of gnathostome–cyclostome divergence relative to 1R and 2R.** The timing of gnathostome–cyclostome divergence relative to the two basal vertebrate tetraploidization events (i.e. 1R and 2R) remains an unresolved issue in the field of vertebrate genome evolution. In order to resolve the divergence timing conclusively, we searched our reconstructions of the proto-vertebrate, proto-cyclostome and proto-gnathostome genomes for evidence of large-scale genomic changes that help distinguish between three alternative divergence models, i.e., divergence before 1R, between 1R and 2R, or after 2R. Our reconstructions revealed nine major fusion events that occurred during the interval between 1R and 2R (see Supplementary Note 3 and Supplementary Fig. 6), but none of these fusions is shared with the proto-cyclostome lineage (Supplementary Fig. 15b, c and Fig. 2), suggesting that the two lineages diverged before the chromosome fusion events and thus before 2R. Furthermore, the orthologue distribution between proto-gnathostome and proto-cyclostome chromosomes demonstrates four-to-six correspondence and a quasi-random gene retention pattern (Supplementary Fig. 15a). This lack of one-to-one or two-to-three orthology relationships indicates that the two lineages diverged shortly after 1R but before rediploidization.

In order to verify the timing of duplications and the gnathostome–cyclostome divergence, we performed a gene tree analysis by inserting lamprey genes into Ensembl gene trees or re-computing the gene trees (see Supplementary Note 5). Then, we classified human and lamprey paralogue pairs by their duplication timing and plotted vertebrate paralogues (i.e. paralogues duplicated before the gnathostome–cyclostome split), gnathostome-specific paralogues and cyclostome-specific paralogues on the proto-gnathostome and proto-cyclostome genomes (Supplementary Figs. 9–15). Intriguingly, we observed a mixture of vertebrate paralogues and cyclostome-specific paralogues between most pairs of homoeologous proto-cyclostome chromosomes, making it difficult to conclusively determine the duplication timing of individual chromosomes. This observation may be explained by (1) difficulties in gene tree inference due to the high GC content and strong codon bias in the lamprey genomes[22,26,33], (2) differential gene loss between cyclostome and gnathostome lineages[29], (3) delayed rediploidization[28,31,32] creating cyclostome-specific paralogues between proto-cyclostome chromosomes duplicated by 1R, and (4) tetraploidization through hybridization and doubling[57–59], which may have created both vertebrate-specific and cyclostome-specific paralogues due to recurrent hybridization among genetically diverse subpopulations[57,58] and subsequent genetic drift[60]. Although these factors may have obscured the duplication timing, the

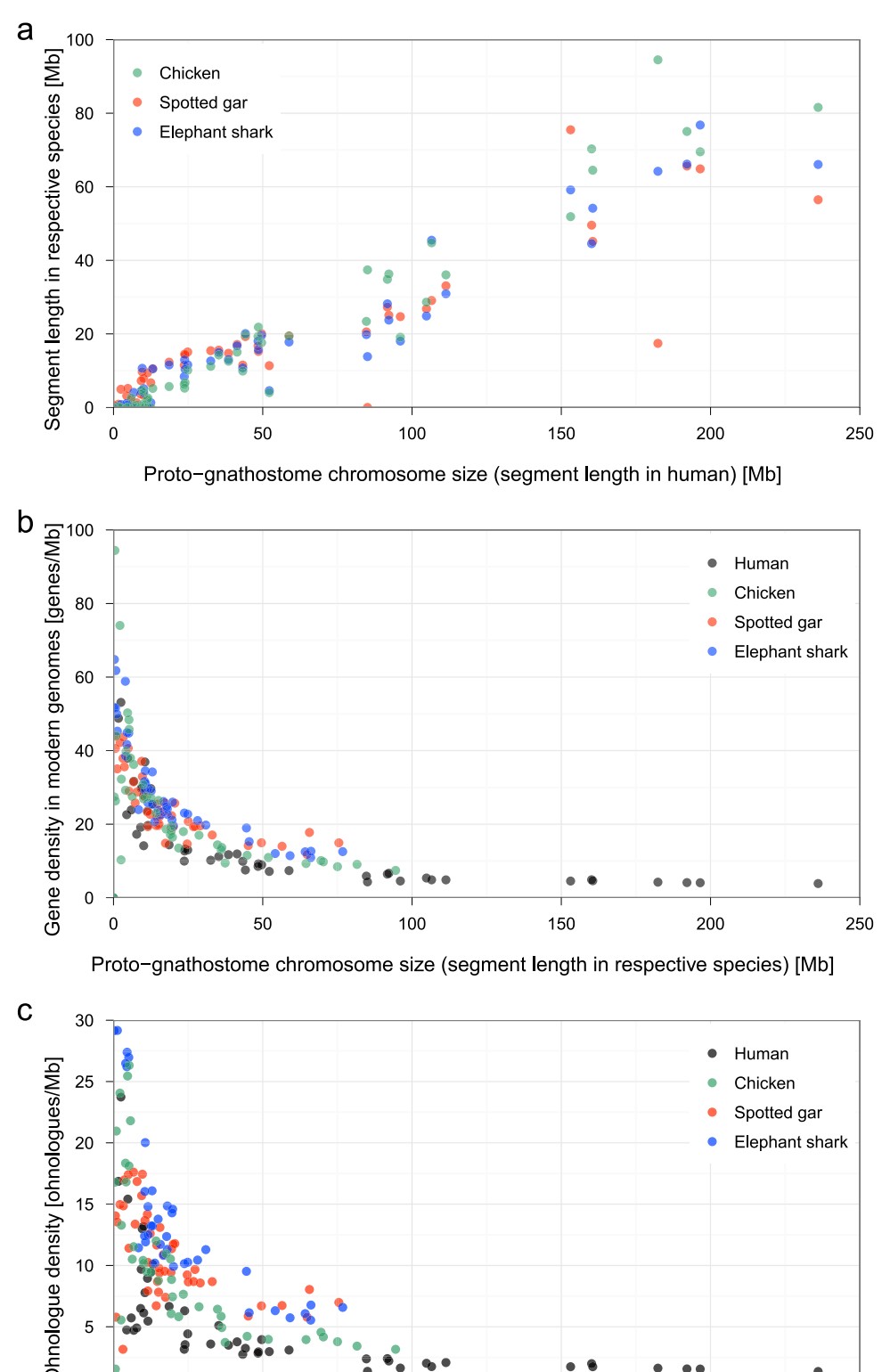

**Fig. 5 Persistent conservation of chromosomal features associated with microchromosomes in gnathostomes. a** Each proto-gnathostome chromosome, consisting of multiple segments, was mapped to modern genomes, and the total segment length in the human genome is shown on the *x*-axis, whereas the total segment length in the chicken, spotted gar and elephant shark genomes are shown on the *y*-axis. **b** Gene density (i.e., number of genes per megabase) was calculated in human, chicken, spotted gar and elephant shark segments derived from individual proto-gnathostome chromosomes. The *x*-axis shows the chromosome length in individual species. **c** Ohnologue density instead of gene density is shown on the *y*-axis.

presence of chromosome pairs enriched either with vertebrate-specific paralogues or cyclostome-specific paralogues are consistent with the model that the proto-cyclostome lineage diverged from the proto-gnathostome lineage shortly after 1R.

**Inferred scenario of early vertebrate genome evolution**. The findings described above can be brought together into a model describing the steps of early vertebrate genome evolution (Fig. 6). First, our reconstruction indicates that the proto-vertebrate genome (with 18 chromosomes) was similar in structure to the ancestral bilaterian animal genome, as suggested by the strong macrosynteny conservation between the proto-vertebrate and scallop genomes (Fig. 6a), e.g. Pvc2, 5, 6, 7, 9, 10, 16, 17 and 18 retain one-to-one correspondence with chr1, 11, 17, 8, 4, 7, 9, 3 and 13 in scallop, respectively. Second, our analysis suggests that the gnathostome–cyclostome divergence occurred shortly after 1R (Fig. 6b) but before rediploidization (Supplementary Fig. 15). This was followed by nine gnathostome-specific chromosome fusions, which were not shared with the proto-cyclostome lineage (Fig. 6c and Supplementary Fig. 6). Third, the 2R event in the proto-gnathostome was an allotetraploidization event, as our reconstruction shows biased gene loss/retention between duplicated chromosomes (Fig. 6e and Supplementary Note 4)[61–63]. Indeed, the ratio of retained genes between the two subgenomes in the proto-gnathostome genome is 2.25, which is considerably larger than previously reported ratios of paleo-allopolyploids: 1.47 for *Brassica*, 1.46 for maize, 1.24 for sorghum, 1.17 for *Arabidopsis* and 1.35 for *Xenopus laevis*[61,64]. A comparison with the modern gnathostome genomes (Fig. 6f and Supplementary Fig. 7) shows that a pair of chromosomes duplicated by 2R typically gave rise to a large chromosome (dashed lines) and a microchromosome (solid lines) in elephant shark and chicken, which suggests that the proto-gnathostome ancestor already possessed microchromosomes as a result of biased fractionation between the subgenomes. (A paper published after the submission of this manuscript suggested a similar evolutionary scenario[65].) Fourth, we present evidence that there was a cyclostome-specific hexaploidization (Fig. 6g) that gave rise to the proto-cyclostome genome with $18 \times 2 \times 3$ chromosomes, most of which are still retained in the modern lamprey genomes with ~99 chromosomes[45] due to remarkably low rates of interchromosomal rearrangement.

## Discussion

To the best of our knowledge our reconstruction is the first reported genome-scale evidence for hexaploidy in the cyclostome lineage. There are several documented examples of hexaploidy giving rise to new evolutionary lineages. Perhaps the most well-known example is wheat, a domesticated crop with three sub-genomes (A, B and D). The formation of hexaploid wheat is believed to be a multi-step process where there was an initial tetraploid genome formed by hybridization, and a subsequent hybridization of the tetraploid with a diploid, generating a hexaploid[66]. Hexaploidy has also been shown in early dicot plant evolution[67], the shortnose sturgeon (*Acipenser brevirostrum*)[46], and in the Prussian carp (*Carassius gibelio*)[68,69]. In most instances the mechanism of hexaploidy origin has been inferred to have been by serial hybridizations[46,66].

These instances suggest that genome hybridization may have played a significant role in the origin and evolution of early vertebrates, as already discussed in previous studies[9,70]. A few possible mechanisms have been suggested for explaining the establishment of allopolyploid species. First, heterosis, or hybrid vigour, confers selective advantages to the newly formed

allopolyploids[71,72]. Second, asymmetric and unequal contribution from the subgenomes has been reported to have facilitated the evolution of complex phenotypes in some allopolyploids: for example, it was suggested that the allotetraploid cotton produces high-quality fibres by combination of long fibres from the A-genome and short fibres from the D-genome[73]; in the allohexaploid wheat, the A-genome is responsible for the morphological traits, while the B- and D-genomes contain most genes for response to biotic and abiotic factors[74]. In line with this argument, previous studies have shown an example of asymmetric contribution from quadruple paralogous regions in the human genome[75]. Our reconstruction suggests that such asymmetric evolution may not be limited to specific gene clusters but is a genome-scale phenomenon due to the hybrid origin of the allopolyploid proto-gnathostome genome.

In particular, our reconstruction suggests that genome hybridization might have contributed to the origin of the adaptive immune system (AIS), which is a prime example of a major evolutionary innovation in early vertebrates. The human AIS is an intricate defence system characterized by the B cell and T cell receptors and the major histocompatibility complex (MHC), which are highly conserved throughout most gnathostomes, including cartilaginous fishes, but are missing in invertebrates, including the closest relatives of vertebrates, such as sea squirts and amphioxus[76,77]. The seemingly abrupt emergence of such a complex molecular machinery of AIS has been described as an evolutionary 'Big Bang' triggered by macroevolutionary events, including the two rounds of tetraploidization[76,78–80].

Of particular interest with regard to the origin of the AIS is a previous hypothesis on the evolution of highly polymorphic genes encoded in the MHC, natural killer gene complex (NKC) and leucocyte receptor complex (LRC), which are essential for the mammalian AIS. It has been proposed that the precursors of MHC, NKC and LRC were physically linked on the proto-MHC chromosome, and the tight linkage facilitated co-evolution of highly polymorphic receptors and ligands, giving rise to a putative 'immune supercomplex' that was subsequently fragmented as MHC, NKC and LRC in the human genome[76,78–81]. However, our reconstruction shows that the hypothesized proto-MHC chromosome was an artefact due to inter 1R–2R chromosome fusions and additional rearrangements in the mammalian lineages. In our reconstruction, the putative proto-MHC chromosome is divided into several proto-vertebrate chromosomes (i.e. Pvc5, 11, 13, 14, 15, 17 and 18) that are clearly separated in the lamprey, amphioxus and scallop genomes (Fig. 2), and the MHC, NKC and LRC gene clusters are derived from these distinct proto-vertebrate chromosomes including Pvc5, 15 and 17.

Intriguingly, our reconstruction shows that MHC, NKC and LRC were located on microchromosomes in the proto-gnathostome genome (i.e. Pgc38, 12 and 27 in Supplementary Data 1). This observation suggests that the post-1R tetraploid species might have already had co-evolving genes encoding the precursors of MHC, NKC and LRC, and the post-2R allopolyploid preserved this interaction network within a subgenome despite the higher rate of gene loss in microchromosomes. This view is also consistent with the previous observation that functionally linked genes involved in 'response to stimulus' (e.g. genes involved in adaptive immunity) tend to be retained in *cis* after 2R, suggesting that interacting gene clusters were preserved despite extensive gene loss[82,83]. In addition, we observed functional biases between the two subgenomes: the human genes in the segment derived from the shorter subgenome were enriched with 'defense/immunity protein' in PANTHER Protein Class (FDR $q = 2.75 \times 10^{-13}$, see Supplementary Note 4). Overall, our reconstruction suggests a possible role of asymmetric

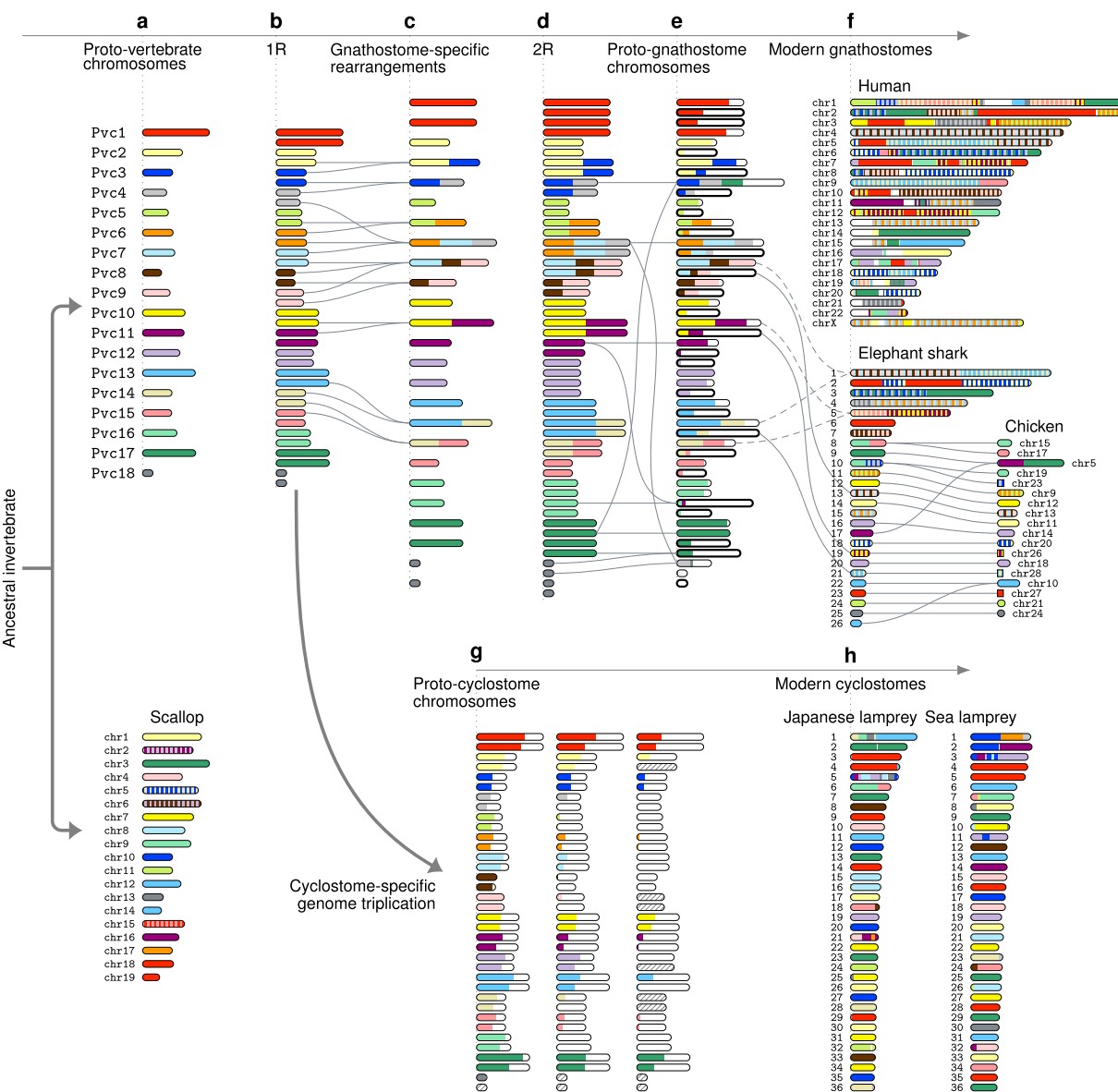

**Fig. 6 Inferred scenario of the early vertebrate genome evolution. a** The 18 proto-vertebrate chromosomes show strong macrosynteny conservation in the scallop genome. Scallop chromosomes are coloured to indicate homology with proto-vertebrate chromosomes, where stripes indicate homology to multiple proto-vertebrate chromosomes. **b** The first tetraploidization (1R) doubled the proto-vertebrate chromosomes. **c** The post-1R genome experienced nine chromosome fusions, which were not shared with the proto-cyclostome lineage. **d** The second tetraploidization (2R) established the proto-gnathostome genome with 49 chromosomes after a few post-2R fusions. **e** The duplicated proto-gnathostome chromosomes exhibit biased gene loss/retention, as indicated by different lengths of coloured bars (see Supplementary Note 4 for details). Chromosomes surrounded by a thick line belong to the shorter subgenome (with a higher rate of gene loss). **f** Chromosomal regions in modern gnathostome genomes were painted according to the originating proto-gnathostome chromosomes, where stripes indicate fusion chromosomes in the proto-gnathostome genome (e.g. human chr9 was derived from Pvc13, 14 and 15). The largest 26 elephant shark scaffolds and selected chicken chromosomes demonstrate the persistent conservation of microchromosomes. The 2R event duplicated fused chromosomes, each of which gave rise to a large chromosome (dashed lines) and a microchromosome (solid lines) in elephant shark and chicken, which suggests that microchromosomes were created by biased gene fractionation after 2R. **g** The cyclostome lineage diverged from the gnathostome lineage shortly after 1R, and the proto-cyclostome genome was formed by hexaploidization. Coloured bars show biased gene loss/retention, and chromosomes missing in our reconstruction are shown as hatched bars. **h** The largest 36 scaffolds in modern lamprey genomes were painted according to their originating proto-vertebrate chromosomes.

contribution from subgenomes for the emergence of gnathostome-like AIS, and corroborates the view that a primordial 'adaptive' immune system emerged in the ancestral vertebrate genome and later turned into the intricate gnathostome-like AIS through 2R[76,77,80].

Finally, our reconstruction of the proto-gnathostome genome has implications for understanding the intrinsic evolutionary constraints on gnathostome genomes. It has previously been

shown ohnologues are frequently dosage-sensitive and resistant to evolutionary duplication and loss[84–86], and that the distribution of ohnologues constrains copy-number variations among human populations[87]. Interestingly, our reconstruction suggests that the high gene and ohnologue densities are conserved features (Fig. 5) associated with microchromosomes created by biased gene loss after 2R (Fig. 6), and that human chromosomal regions with high ohnologue densities originated from microchromosomes in the

proto-gnathostome genome. Thus, ohnologue-rich regions that are susceptible to pathogenic copy-number variations may be regarded as a legacy from the allopolyploid proto-gnathostome genome and subsequent asymmetric evolution between the sub-genomes. In addition, by referring to the evolutionary origins (Fig. 6), we can (1) identify ohnologue relationships between genes with low sequence similarity[30] that might otherwise remain cryptic and (2) prioritize copy-number variations in personal genomes for potential pathogenicity.

In conclusion, we have generated high-quality, chromosome-scale genome assemblies for two phylogenetically opportune organisms, and inferred the genome structures of early vertebrate lineages. This is the first effort to reconstruct the proto-cyclostome genome, which was critical for determining the cryptic origins of the proto-cyclostome and proto-gnathostome genomes. Consequently, our reconstruction resolved several important issues including the number and relative timings of polyploidization events that occurred during the early origin of vertebrates. The resulting model offers unique perspectives on the origin and evolution of vertebrate genomes.

## Methods

**Probabilistic macrosynteny model**. We reconstructed the proto-vertebrate genome by employing the probabilistic macrosynteny model[34], which was previously used for inferring the structure of the pre-TGD genome (TGD stands for teleost-specific genome duplication). The details including the probability model, definitions of parameters/variables, algorithm and estimation accuracy can be found in ref. [34] (open access). In short, the macrosynteny model assumes that the individual pre-WGD chromosomes have distinct orthologue distributions over the present-day post-WGD genomes; then, the pre-WGD genome structure can be reconstructed by employing the variational Bayesian inference algorithm. In the present study, we used an algorithm called collapsed variational Bayes (CVB)[88,89], which is more efficient than the variational Bayesian expectation-maximization algorithm described in ref. [34].

The CVB algorithm is derived as follows using the same model and variables as defined in ref. [34]. In the framework of the probabilistic macrosynteny model, we infer the pre-WGD genome structure as the posterior (i.e. $p_{\Theta,X|Y}$) of the model parameters ($\Theta$) and latent variables ($X$) conditioned by the orthologue information ($Y$). Since exact computation of the posterior is infeasible, the posterior needs to be approximated by tractable probability density functions. For deriving the CVB algorithm, the posterior is approximated by $q_{\hat{\Theta},\hat{X}}(\theta, x)$ that can be factorized as

$$q_{\hat{\Theta},\hat{X}}(\theta, x) = q_{\hat{\Theta}|\hat{X}}(\theta|x)\prod_{s=1}^{S}\prod_{g=1}^{G_s} q_{\hat{X}_{s,g}}\left(x_{s,g}\right), \quad (1)$$

where $S$ is the number of non-WGD segments and $G_s$ is the number of genes in segment $s$. Then, assuming this factorization and following the derivation of the CVB (or CVB0) algorithm[88,89] for the probabilistic topic model[90,91], we obtain Algorithm 1 with the following update formula:

$$\log\left(q_{\hat{X}_{s,g}}(k)\right) = \log\left(\hat{a}_k^{(s)} - q_{\hat{X}_{s,g}}(k)\right) + \sum_{t=1}^{T}\sum_{c=1}^{C_t}\sum_{i=1}^{n_{t,c}^{s,g}}\log\left(i - 1 + \hat{\beta}_c^{(k,t)} - q_{\hat{X}_{s,g}}(k)n_{t,c}^{s,g}\right)$$
$$- \sum_{t=1}^{T}\sum_{i=1}^{n_t^{s,g}}\log\left(i - 1 + \sum_{c=1}^{C_t}\hat{\beta}_c^{(k,t)} - q_{\hat{X}_{s,g}}(k)n_t^{s,g}\right) + C, \quad (2)$$

where $\hat{a}_k^{(s)}$ and $\hat{\beta}_c^{(k,t)}$ are given by Eqs. (11) and (12) in ref. [34] and $C$ is a constant that cancels by normalizing $q_{\hat{X}_{s,g}}(k)$ so that $\sum_{k=1}^{K} q_{\hat{X}_{s,g}}(k) = 1$. In the actual computation, we avoided early convergence of $q_{\hat{X}}$ to suboptimal values by starting from a less extreme prior distribution with a slightly larger value of $\alpha_k$ as follows: at the $j$th iteration, we replaced $\alpha_k$ with $\alpha_k + 0.8^{j-1}$ while $j < 98$. We continued updating $q_{\hat{X}}$ until after 100 iterations or until $q_{\hat{X}}$ converges, satisfying

$$\sum_{s=1}^{S}\sum_{g=1}^{G_s}\sum_{k=1}^{K}\left|q_{\hat{X}_{s,g}}(k) - q'_{\hat{X}_{s,g}}(k)\right| < 0.001, \quad (3)$$

where $q'$ denotes the estimate in the previous iteration.

Below is a pseudocode for the CVB0 algorithm.

**Input:** Analysis constants, non-WGD segments, observed orthologues $y$ and parameters $\alpha$ and $\beta$.

**Output:** Approximate posterior distribution of $X$ conditioned by $Y = y$.

1: Initialize variational parameters.

2: **repeat**

3: **for** segment $s = 1, \ldots, S$ **do**

4: **for** gene $g = 1, \ldots, G_s$ **do**

5: **for** pre-WGD chromosome $k = 1, \ldots, K$ **do**

6: Update $q_{\hat{X}_{s,g}}(k)$ using Equation (2).

7: Normalize $q_{\hat{X}_{s,g}}(k)$ for $k = 1, \ldots, K$.

8: **until** all $q_{\hat{X}_{s,g}}(k)$ converge.

**Reconstruction of the proto-vertebrate genome**. We reconstructed the structure of the proto-vertebrate genome in two steps: first, we partitioned the lamprey genomes into blocks of conserved synteny by comparing the lamprey genomes with each other and also with four gnathostome genomes (i.e. human, chicken, spotted gar and elephant shark); second, we inferred the structure of the pre-WGD genome by applying the macrosynteny model to the amphioxus and lamprey genomes. These steps are described below.

**Segmentation of the lamprey genomes**. We partitioned the lamprey scaffolds (with at least ten genes) into blocks of conserved synteny as described in ref. [34]. Specifically, we employed the Bayesian segmentation model[92] and computed the optimal segmentation using a dynamic programming algorithm[93]. Segmentation was performed in two steps: first, we compared the Japanese lamprey and sea lamprey scaffolds, and identified lineage-specific synteny breakpoints; second, we compared the lamprey genomes with human, chicken, spotted gar and elephant shark genomes, and identified breakpoints occurring between the gnathostomes and cyclostomes. Then we merged the two sets of breakpoints and obtained 191 Japanese lamprey segments and 198 sea lamprey segments. These segments have homogeneous distributions of orthologues in the other genomes under comparison, and thus they are likely to have been unaffected by large-scale inter-chromosomal rearrangements in the cyclostome lineages.

**Inference of the pre-WGD genome structure**. We analysed the 1-to-4 orthologue distribution among the amphioxus scaffolds[14] and the Japanese lamprey and sea lamprey segments, by applying the macrosynteny model and CVB0 algorithm with the following parameter values: post-WGD species $T = 2$, numbers of post-WGD segments $C_1 = 191$ and $C_2 = 198$, maximum number of co-orthologues $D^{(t)} = 4$ for all $t$, $L = 10$, $\alpha_k = 0.1$ for all $k$, and $\beta_c^{(t)} = 0.1$ for all $c$ and $t$ (see ref. [34] for details of these parameters). As described in ref. [34], individual amphioxus scaffolds were associated with mixture distributions over the proto-vertebrate chromosomes, which represent reconstruction confidence scores. For the sake of simplicity in visualization, we assigned each amphioxus scaffold to the proto-vertebrate chromosome with the largest reconstruction confidence score (i.e. $\text{argmax}_k \mathbb{E}[\hat{U}_{s,k}]$, where $\mathbb{E}$ denotes expectation), which is calculated by using Eq. (11) in ref. [34]. In addition, we assigned each lamprey segment to the proto-vertebrate chromosome with the largest reconstruction confidence score (i.e. $\text{argmax}_k \mathbb{E}[\hat{V}_{t,k,c}]$), which is calculated by using Eq. (12) in ref. [34]. See also Fig. 1 in ref. [34] for an intuitive explanation.

**Number of proto-vertebrate chromosomes**. In the macrosynteny model, the number of proto-vertebrate chromosomes is treated as an input parameter ($K$) for inferring the optimal pre-WGD genome structure. The previous studies estimated the number of proto-vertebrate chromosomes to be 10–13 in refs. [12,13,18,19,94] or 17 in refs. [14,15], but the exact number is unknown. In order to decide the optimal number of $K$, we reconstructed the proto-vertebrate chromosomes with $K = 10, \ldots, 20$, and evaluated the quality of those reconstructions by comparing their paralogue distributions as follows.

The underlying assumption is that most lamprey paralogues were created by WGDs (or by chromosome-scale duplications as proposed in ref. [18]); then, the paralogue distribution should be highly non-random, with most paralogues found between lamprey segment pairs both deriving from the same proto-vertebrate chromosome. We quantified such non-randomness by using the hypergeometric distribution under the null hypothesis in which paralogues are randomly distributed over the entire genome as described below.

Let $g$ and $p$ be the number of gene pairs and paralogue pairs in the genome, respectively, and $v$ be the number of gene pairs both of which derive from the same proto-vertebrate chromosomes. Let $X$ be a random variable representing the

number of paralogue pairs both of which derive from the same proto-vertebrate chromosome, and $x$ be the observed number of such paralogue pairs. Then, the significance of $x$ is given as follows:

$$\mathbb{P}(X \geq x) = \sum_{i=x}^{v} \binom{g-v}{p-i}\binom{v}{i} \Big/ \binom{g}{p} \qquad (4)$$

where $\mathbb{P}$ denotes the probability and () denotes the binomial coefficient.

In both the Japanese lamprey and sea lamprey genomes, the reconstruction with $K = 18$ was the most significant in this criterion (Supplementary Table 8). Then we labelled the proto-vertebrate chromosomes as Pvc1–Pvc18 (although we observed that Pvc18 has no clear synteny in gnathostome and invertebrate genomes).

**Reconstruction of the proto-cyclostome genome.** Although the 2R hypothesis was resolved by a genome-wide synteny analysis[11] and reconstruction analyses[12–14], the origins of the proto-gnathostome and proto-cyclostome genomes have remained contentious. In particular, the timing of gnathostome–cyclostome divergence and possibility of cyclostome-specific WGD have remained topics of debate even after sequencing of the sea lamprey genome[18,22,26]. For example, six Hox clusters were found in the cyclostome genomes[19,22,26,27], but it was not clear if the number of Hox clusters should be explained by additional cyclostome-specific WGD followed by the loss of two entire clusters, or by chromosome-scale duplications in the cyclostome lineage[18,19,22].

We considered that a reconstruction of the proto-cyclostome chromosomes would provide a conclusive answer to this question. In our macrosynteny model analysis, the Hox-bearing proto-vertebrate chromosome comprises 10 Japanese lamprey segments and 11 sea lamprey segments, which are likely to be parts of proto-cyclostome chromosomes fragmented due to inter-chromosomal rearrangements or limited scaffold length in the current genome assemblies. Thus, the reconstruction of proto-cyclostome chromosomes can be formulated as finding the correct combination from a large number of possible combinations of the lamprey segments. The enumeration of all combinations is called 'set partitioning' (i.e. partitioning of a set of segments into non-empty subsets), which is computationally infeasible because the number of all set partitions, known as the Bell number, can be extremely large: for example, the Bell number for the 21 lamprey segments is $B_{21} = 474,869,816,156,751$. To address this problem, we performed clustering of lamprey segments and reduced the number of set partitions as follows.

**Step 1**: Paralogous lamprey segments do not originate from the same proto-cyclostome chromosome. Therefore, we calculated the paralogue significance for each segment pair, and significant pairs were not allowed to be assigned to the same cluster in the subsequent steps.

**Step 2**: Orthologous segments between Japanese lamprey and sea lamprey originate from the same proto-cyclostome chromosome. Therefore, we performed a single linkage clustering of lamprey segments to make clusters of orthologous segments. First, we defined individual segments as initial clusters. Second, we sorted segment pairs by the significance of the number of orthologues between them. Third, we repeated choosing the most significant segment pair and merged the two clusters if they did not have paralogous segments.

**Step 3**: Some lamprey scaffolds are expected to be over-fragmented by the synteny segmentation algorithm or by lineage-specific rearrangements. In order to address such over-fragmentation of lamprey scaffolds, we merged two clusters if (i) they had segments on the same scaffold and (ii) they did not have paralogous segments.

**Step 4**: In addition to Step 3, we utilized the Pacific lamprey linkage markers[19] and merged two clusters if (i) the clusters shared a pair of sea lamprey segments having linkage markers on the same Pacific lamprey linkage group and (ii) the clusters did not have paralogous segments.

**Step 5**: Reliable reconstruction is difficult for short segments having few orthologues and paralogues. Therefore, clusters of lamprey segments were excluded from the proto-cyclostome reconstruction if the clusters had fewer than five genes.

For each of Pvc1–Pvc17, we enumerated all set partitions of the clusters, and chose the optimal set partition with the most significant distribution of orthologues and paralogues as the proto-cyclostome chromosomes. During this analysis we found that some Japanese lamprey scaffolds are likely to be haplotype sequences that were not removed from the primary assembly by FALCON during the final stage of the assembly; we therefore excluded the following Japanese lamprey scaffolds from the proto-cyclostome reconstruction: Scaffolds 110, 190, 198, 105, 104, 82, 163, 69, 133, 74, 115, 139, 86, 70, 72, 171 and 192. We left Pvc18 as a single proto-cyclostome chromosome because computation of the optimal set partitioning was infeasible for Pvc18 consisting of 34 segments.

Significance of paralogues, orthologues and set partitions were calculated as follows.

**Significance of the number of paralogues.** The significance of the number of paralogues between two lamprey segments is calculated as follows. Let $g$ and $p$ be the number of gene pairs and paralogue pairs in the genome, respectively, and $n$ be the number of gene pairs between the two segments. Let $X$ be a random variable representing the number of paralogue pairs between the two segments, and $x$ be the observed number of such paralogue pairs. Then, the probability of

observing $x$ paralogue pairs between the two segments is given by

$$\mathbb{P}(X = x) = \binom{g-n}{p-x}\binom{n}{x} \Big/ \binom{g}{p}, \qquad (5)$$

and the number of paralogue pairs was considered significant if $\mathbb{P}(X \geq x) < 10^{-5}$.

**Significance of the number of orthologues.** The significance of the number of orthologues between two lamprey segments is calculated as follows. Let $g$ and $o$ be the number of gene pairs and orthologue pairs between the Japanese lamprey and sea lamprey genomes, respectively, and $n$ be the number of gene pairs between the two segments. Let $X$ be a random variable representing the number of orthologue pairs between the two segments, and $x$ be the observed number of such orthologue pairs. Then, the probability of observing $x$ orthologue pairs between the two segments is given by

$$\mathbb{P}(X = x) = \binom{g-n}{o-x}\binom{n}{x} \Big/ \binom{g}{o} \qquad (6)$$

and the number of orthologue pairs was considered significant if $\mathbb{P}(X \geq x) < 10^{-5}$.

**Significance of a reconstruction.** For each proto-vertebrate chromosome $c$ ($c = 1, \dots, 17$ for Pvc1,...,17, respectively) and each species $s$ ($s = 1$ for Japanese lamprey and $s = 2$ for sea lamprey), the significance of paralogues was calculated as follows. Let $g_{c,s}$ be the number of gene pairs and $p_{c,s}$ be the number of paralogue pairs in species $s$ such that both genes derive from proto-vertebrate chromosome $c$. Let $n_s (\leq g_{c,s})$ be the number of gene pairs between different proto-cyclostome chromosomes (i.e. inter-chromosome gene pairs). Let $X_s (\leq p_{c,s})$ be a random variable representing the number of inter-chromosome paralogue pairs, and $x_s$ be the observed number of such paralogue pairs. Then, the significance of $x_s$ inter-chromosome paralogue pairs is given by

$$\mathbb{P}(X_s \geq x_s) = \sum_{i=x_s}^{p_{c,s}} \binom{g_{c,s}-n_s}{p_{c,s}-i}\binom{n_s}{i} \Big/ \binom{g_{c,s}}{p_{c,s}} \qquad (7)$$

In addition, we calculated the significance of the number of orthologues between species $s$ and $t$. Let $g_{c,s,t}$ be the numbers of gene pairs and $o_{c,s,t}$ be the number of orthologue pairs between the two species such that both genes derive from proto-vertebrate chromosome $c$. Let $n_{s,t} (\leq g_{c,s,t})$ be the number of gene pairs between species $s$ and $t$, where both genes derive from the same proto-cyclostome chromosome. Let $X_{s,t} (\leq o_{c,s,t})$ be a random variable representing the number of orthologue pairs, deriving from the same proto-cyclostome chromosome, and $x_{s,t}$ be the observed number of such orthologue pairs. Then, the significance of $x_{s,t}$ orthologue pairs is given by

$$\mathbb{P}(X_{s,t} \geq x_{s,t}) = \sum_{i=x_{s,t}}^{o_{c,s,t}} \binom{g_{c,s,t}-n_{s,t}}{o_{c,s,t}-i}\binom{n_{s,t}}{i} \Big/ \binom{g_{c,s,t}}{o_{c,s,t}} \qquad (8)$$

Finally, we defined the significance of the set partition for proto-vertebrate chromosome $c$ as

$$\mathbb{P}(X_1 \geq x_1)\mathbb{P}(X_2 \geq x_2)\mathbb{P}(X_{1,2} \geq x_{1,2}). \qquad (9)$$

This method is an extension of the reconstruction of post-2R chromosomes in ref. [13], which was developed for verifying if genome quadruplication occurred in the proto-vertebrate lineage: in the previous study, set partitioning into 2, 3, 4 and 5 post-2R chromosomes were enumerated for showing that quadruplication was the most significant; we extended it to also enumerating set partitions into more than five proto-cyclostome chromosomes.

**Reconstruction of the proto-gnathostome genome.** We reconstructed the proto-gnathostome chromosomes by comparing the amphioxus[14] and several gnathostome genomes including elephant shark. As illustrated in Fig. 4, we performed reconstruction in three steps: first, we partitioned the gnathostome chromosomes into blocks of conserved synteny (Fig. 4f); second, we applied the CVB0 algorithm and made groups of gnathostome segments that share large numbers of paralogues (Fig. 4g); third, segments in individual groups were further partitioned into several subgroups representing proto-gnathostome chromosomes (Fig. 4h).

**Segmentation of the gnathostome genomes.** We partitioned the gnathostome genomes as described in ref. [34]. Specifically, we performed genome segmentation twice for each gnathostome genome: one with four teleost genomes (i.e. zebrafish, stickleback, medaka and *Tetraodon*) to identify blocks of doubly conserved synteny, and the other with chicken, turkey, zebra finch, anole lizard and spotted gar to identify additional synteny breakpoints in individual lineages. Then we merged the two sets of synteny breakpoints to define 151 human segments. Similarly, we partitioned the mouse, dog, opossum, chicken, turkey, zebra finch, spotted gar and elephant shark genomes into 258, 212, 163, 70, 69, 60, 78 and 132 segments, respectively. These numbers of segments are slightly different from the previous study[34], because we used the spotted gar genome in addition to the non-mammalian amniotes in the synteny segmentation step.

**Analysis with the macrosynteny model**. We used the macrosynteny model and applied the CVB0 algorithm with the following parameter values: number of post-WGD species $T = 9$, numbers of post-WGD segments $C_t = 151, 258, 212, 163, 70, 69, 60, 78$ and $132$ for $t = 1, \ldots, 9$ respectively, maximum number of co-orthologues $D^{(t)} = 4$ for all $t$, $L = 10$, $\alpha_k = 0.1$ for all $k$, and $\beta_c^{(t)} = 0.1$ for all $c$ and $t$. Then we set $K = 7, \ldots, 20$ and evaluated the reconstruction quality by comparing the significance of paralogue distribution as described for the reconstruction of proto-vertebrate genome. We found that clustering into 10 groups of segments was optimal, which is consistent with the previous study[13]. However, as we argue in Fig. 4, individual groups might represent multiple proto-vertebrate chromosomes due to inter-chromosomal rearrangements. Therefore, we reconstructed proto-gnathostome chromosomes by employing a rearrangement-aware method as follows.

**Reconstruction of proto-gnathostome chromosomes**. In this step, we used segments from the human, mouse, dog, opossum, chicken, turkey, zebra finch, spotted gar and elephant shark genomes, and enumerated set partitions for each of the 10 groups after reducing the number of set partitions by clustering of gnathostome segments as in the reconstruction of proto-cyclostome chromosomes:

**Step 1**: Paralogous gnathostome segments do not originate from the same proto-gnathostome chromosome. Therefore, we calculated the paralogue significance for each segment pair, and significant pairs were not allowed to be assigned to the same cluster in the subsequent steps.

**Step 2**: Orthologous gnathostome segments originate from the same proto-gnathostome chromosome. Therefore, we performed a single linkage clustering of gnathostome segments to make clusters of orthologous segments. First, we defined individual segments as initial clusters. Second, we sorted segment pairs by the significance of the number of orthologues between them. Third, we repeated choosing the most significant segment pair and merged the two clusters if they did not have paralogous segments.

**Step 3**: Reliable reconstruction is difficult for short segments having few orthologues and paralogues. Therefore, clusters of gnathostome segments were excluded from the proto-gnathostome reconstruction if the clusters had fewer than five genes.

For the reconstruction of proto-gnathostome chromosomes, we assumed that individual gnathostome segments might have derived from multiple proto-vertebrate chromosomes, since it was reported that several rearrangements occurred between the two WGD events[13]. Figure 4a illustrates the case of a chromosome fusion occurring between the two WGD events. As the result of the fusion, the grey post-2R chromosomes share large numbers of ohnologues with the black and white chromosomes (represented by red regions in Fig. 4c); on the other hand, there are no ohnologues between black and white chromosomes (white regions). In addition to the case of a chromosome fusion between the two WGD events, our reconstruction method considered other rearrangement scenarios, namely, (A) a chromosome fission event occurring in the period between 1R and 2R and (B) a fusion or translocation after 2R. Scenario A results in the same paralogue distribution pattern as in the case of a fusion between the two WGD events, but the two scenarios can be distinguished by checking the orthologue distribution in invertebrate genomes. In Scenario B, the paralogue distribution is different from Scenario A, since the chromosome created by post-2R fusion is paralogous to the other six post-2R chromosomes. In general, we expect to see a large number of paralogues between a pair of proto-gnathostome chromosomes, only if the two chromosomes (1) are duplicated chromosomes or (2) inherit duplicated chromosomes or duplicated segments through rearrangements (fusions, fissions and translocations). These proto-gnathostome chromosome pairs are called 'red chromosome pairs' (as in Fig. 4c) in the subsequent texts.

Then, for each cluster $c$ ($c = 1, \ldots, 10$ for the ten clusters) and for each species $s$ ($s = 1, \ldots, 9$ for human, mouse, dog, opossum, chicken, turkey, zebra finch, spotted gar and elephant shark, respectively), the significance of paralogues was calculated as follows. Let $g_{c,s}$ be the number of gene pairs and $p_{c,s}$ be the number of paralogue pairs in species $s$ such that both genes are in cluster $c$. Let $n_s (\leq g_{c,s})$ be the number of gene pairs between red proto-gnathostome chromosome pairs. Let $X_s (\leq p_{c,s})$ be a random variable representing the number of paralogue pairs between red chromosome pairs, and $x_s$ be the observed number of such paralogue pairs. Then, the significance of $x_s$ intra-chromosome paralogue pairs is given by

$$\mathbb{P}(X_s \geq x_s) = \sum_{i=x_s}^{p_{c,s}} \binom{g_{c,s} - n_s}{p_{c,s} - i} \binom{n_s}{i} \bigg/ \binom{g_{c,s}}{p_{c,s}}. \quad (10)$$

Significance of orthologues between two gnathostome species was given in the same way as in the proto-cyclostome reconstruction. Then, we calculated the significance of a set partition for cluster $c$ by multiplying the significance values for all species and all species pairs:

$$\left( \prod_s \mathbb{P}(X_s \geq x_s) \right) \left( \prod_{s \neq t} \mathbb{P}(X_{s,t} \geq x_{s,t}) \right). \quad (11)$$

In the last step of the proto-gnathostome reconstruction, we chose the most significant set partition and filtered out small unreliable subgroups. This filtering step was necessary because reliable subgroups are expected to have segments from all (or most) of the gnathostome species, whereas reconstruction errors result in spurious small subgroups having short segments from few species. For this reason, we filtered out small subgroups with segments from only few (<3) species, which filtered out nine small subgroups consisting of four mouse segments, one chicken segment and nine elephant shark segments. This filtering step should have little influence on the analysis results because the number of genes on the filtered segments is only 288 in total. Finally, the remaining subgroups were defined as the proto-gnathostome chromosomes.

**The proto-cyclostome genome was shaped by sixfold genome duplication**. Here, we introduce a framework for calculating the probability that multiplicities of independently duplicating chromosomes converge toward a given ploidy level, where the convergence is measured in terms of the deviation ($\delta$) from the given ploidy level. Application to the proto-cyclostome genome shows that the observed peak of multiplicity at six is unlikely to be created by chance through accumulation of chromosome-scale duplications.

Let us consider the following situation. The proto-vertebrate genome with $K$ chromosomes underwent one or two polyploidization events, producing $X_k (k = 1, \ldots, K)$ duplicates for each proto-vertebrate chromosome ($X_k = 2$ for all $k$ after 1R or $X_k = 4$ after two rounds of tetraploidization). Subsequently, those $X = \sum_{k=1}^{K} X_k$ chromosomes were duplicated by a series of independent chromosome-scale duplications, eventually creating $Y_k$ duplicates for each proto-vertebrate chromosome ($k = 1, \ldots, K$). As a measure of deviation from a polyploidization-only model, we define $\delta(Y_k) = \sum_{k=1}^{K} |Y_k - M|$, where $M$ is the expected multiplicity ($M = 6$ in our model). Assuming that all chromosomes are equally likely to be duplicated, we calculate $\mathbb{P}(\delta(Y_k) \leq D \mid \sum_{k=1}^{K} Y_k = Y)$, the probability that the deviation is smaller than or equal to the observed deviation $D$ (i.e. $D = 13$ in our reconstruction) conditioned by the total number of proto-cyclostome chromosomes $Y$ (i.e. $Y = 103$ in our reconstruction).

The desired probability is calculated as follows. First, the total number of duplication scenarios is given by $T = \Gamma(Y)/\Gamma(X)$, where $\Gamma(n) = (n-1)(n-2)\cdots 1$ is the gamma function. Second, for given $Y_k (k = 1, \ldots, K)$, the number of duplication scenarios in which individual proto-vertebrate chromosomes are eventually duplicated into $Y_k$ proto-cyclostome chromosomes is given by

$$S(Y_1, \ldots, Y_K) = (Y_1 - X_1, \ldots, Y_K - X_K)! \prod_{k=1}^{K} \Gamma(Y_k)/\Gamma(X_k), \quad (12)$$

where $(\cdot, \ldots, \cdot)!$ is the multinomial coefficient. Then, by enumerating all $Y_k$ values, we can calculate the desired probability (i.e. independently duplicating proto-vertebrate chromosomes converging to multiplicity $M$ by chance alone) as

$$\mathbb{P}\left( \delta(Y_k) \bigg| \sum_{k=1}^{K} Y_k = Y \right) = \sum_{\{Y_k\}} S(Y_1, \ldots, Y_K)/T, \quad (13)$$

where the summation is taken over all $Y_k$ that satisfy $\delta(Y_k) \leq D$ and $\sum_{k=1}^{K} Y_k = Y$.

In our reconstruction, we have $K = 17$, $Y = 103$, $D = 13$ and $M = 6$ (see Supplementary Table 10 in Supplementary Note 3). We evaluated the following five evolutionary scenarios: (A) chromosome-scale duplications with no tetraploidization, (B) one tetraploidization followed by chromosome-scale duplications, (C) two tetraploidizations followed by chromosome-scale duplications, (D) chromosome-scale duplications followed by one tetraploidization, and (E) first tetraploidization followed by chromosome-scale duplication followed by second tetraploidization. In these scenarios we assume that $X_k = N$ for all $k$, where we set $N = 1$ and $M = 6$ for Scenario A; $N = 2$ and $M = 6$ for Scenario B; $N = 4$ and $M = 6$ for Scenario C; $N = 1$ and $M = 3$ for Scenario D; and $N = 2$ and $M = 3$ for Scenario E. We set $(Y_1, \ldots, Y_{17}) = (6, 5, 6, 6, 7, 7, 6, 6, 4, 6, 8, 5, 6, 4, 9, 6, 6)$ for Scenarios A/B/C and $(Y_1, \ldots, Y_{17}) = (3, 2, 3, 3, 3, 3, 3, 3, 2, 3, 4, 2, 3, 2, 4, 3, 3)$ for Scenarios D/E, based on the proto-cyclostome genome reconstruction. In addition, we evaluated the case of $K = 18$ by setting $Y_{18} = \max(1, N)$, since our model requires $Y_k \geq N$ for all $k = 1, \ldots, K$; we also evaluated the case of $K = 5$, $Y = 30$ and $D = 0$ since larger proto-vertebrate chromosomes are more reliable in our reconstruction and the largest five proto-vertebrate chromosomes have multiplicity six.

Supplementary Table 10 in Supplementary Note 3 shows small probabilities of observing convergence of multiplicities through independent chromosome-scale duplications. Thus, it is unlikely that the proto-cyclostome genome was shaped by a series of independently occurring chromosome-scale duplications.

**Reporting summary**. Further information on research design is available in the Nature Research Reporting Summary linked to this article.

## Data availability
The Japanese lamprey and elephant shark genome sequences generated in this study have been deposited at DDBJ/ENA/GenBank under the accession numbers WFAB00000000 and WEZY00000000, respectively. The reconstruction dataset including information of orthologues, paralogues and gene names in individual chromosomal segments is available as Supplementary Data 1.

## Code availability
The reconstruction software/code is available on request.

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

## Acknowledgements

This work is supported by funding from the European Research Council Grant Agreements 309834 and 771419 to A.McL. and the Biomedical Research Council of A*STAR, Singapore to B.V. We acknowledge the National Supercomputing Centre of Singapore for providing computational resources for this project. We thank Karsten Hokamp for technical assistance for computational analysis and Anthony Redmond for critical reading of the manuscript.

## Author contributions

B.V. and A.McL. conceived and coordinated the project. P.S., V.R., N.E.P., A.P. and B.V. sequenced and annotated the genomes; Y.N. and A.McL. performed genome reconstruction. Y.N., A.McL. and B.V. wrote the manuscript.

## Competing interests

The authors declare no competing interests.
