## [Peer Review File · Nature Communications]

Reviewers' Comments:

Reviewer #1:

Remarks to the Author:

I am happy to recommend that this article be published by Nature Communications as soon as my comments and recommendations, as detailed below and in the attached document, are addressed. I don't think that the study requires additional analyses and I am confident that the methods are appropriate and sufficient, based on my review as well as previous highly-regarded publications by Nakatani and co-authors, as well as Venkatesh and co-authors.

The application of the modified probabilistic macrosynteny model to the question of cyclostome-gnathostome divergence relative to the vertebrate tetraploidizations is especially exciting, as this method has proven its usefulness in previous publications by Nakatani and co-authors. In addition, the sequencing and chromosome-level assembly of a lamprey genome and an elephant shark genome will undoubtedly be useful resources for molecular evolutionary studies in vertebrates and vertebrate genomics. These species hold key taxonomic positions that have previously been under-addressed due to the lack of high-quality genomic resources. Congratulations on a fantastic paper!

However, some methods and procedures are not described clearly, which made some aspects of the analyses difficult to review. I also have some concerns about the conclusion that the cyclostome lineage underwent a hexaploidization event. This is a very novel suggestion, and I want to make sure that the authors have done everything possible to explain their method clearly, so that no serious doubts can be brought forward about the conclusion.

My general comments and suggestions are included below, and more detailed comments have been attached in a separate document.

General comments:

- Will the new genome assemblies be shared as part of any of the commonly used public genome browsers? Is the *Lethenteron camtschaticum* genome assembly, LetJap1.0, the same that has already been shared through NCBI: https://www.ncbi.nlm.nih.gov/assembly/GCA_000466285.1? The submitter of LetJap1.0 matches the home institution of several of the co-authors. If so, the authors should mention in the paper that the genome assembly has been shared, and direct the reader towards the online databases. If a newer assembly has been made, this should be shared in the same way. The BioProject entries for the new genomes mentioned in the paper are not active yet, so I couldn't check them; but presumably these will only include the raw data, not the assembled genomes. Sharing the genome assemblies in an easily browsable/searchable way is crucial.

- The figure legends are inordinately long. Please make sure to only include information relevant for the graphical interpretation of the figure. As they are now, the figure legends include lengthy descriptions of the methodology and descriptions of results. This should not be included in a figure legend. Otherwise, it may look like the authors are not confident that their text is good enough for the reader to understand the figures. Or, perhaps more cynically, that the authors ran out of words in the main text of the paper and are smuggling some of the text into the paper via the figure legends. They can do better. I have suggested some changes in my detailed comments (attached).

- Another smaller issue is the nomenclature of the Japanese or Arctic lamprey, *Lethenteron japonicum* alt. *Lethenteron camtschaticum*. According to the World Register of Marine Species, *L. japonicum* (Martens, 1868) is an unaccepted synonym. Source: <http://www.marinespecies.org/aphia.php?p=taxdetails&id=298380>. The accepted name is *L. camtschaticum* (Tilesius, 1811). Source: <http://www.marinespecies.org/aphia.php?p=taxdetails&id=101173>. This is also the case in the FishBase database (<https://www.fishbase.se/summary/Lethenteron-camtschaticum.html>) and in the

NCBI taxonomy browser (<https://www.ncbi.nlm.nih.gov/Taxonomy/Browser/wwwtax.cgi?id=980415>). Please change all references to the binomial name of this species throughout the manuscript to reflect the accepted nomenclature. At the first reference to the species, on page 4, the common synonym *L. japonicum* should be mentioned. But it's not the accepted name. It's all right if the authors use "Japanese lamprey" as the common name throughout the paper, as long as *L. camtschaticum* is used and the other common name "Arctic lamprey" is mentioned at the first mention of the species.

- The authors consistently mention 18 photo-vertebrate chromosomes throughout the paper. There are several issues with this. 1) These are reconstructed chromosomes, so they are a purely theoretical conjecture about the karyotype of the photo-vertebrate. I understand that it would be cumbersome to clarify this each and every time they are mentioned, and simply writing "chromosome" is a good shorthand, but the authors should be absolutely clear, at crucial points of the text, that these are theoretical constructions. 2) The reconstruction of the "PvcUn" chromosome is a bit more problematic. It seems to consist of a relatively large number of small fragments that could not be assigned to any other of the proto-vertebrate chromosome reconstructions. It is likely that these fragments correspond to other proto-chromosomes and that there is no 18th proto-vertebrate chromosome. Indeed, in comparing their results to those of Sacerdot et al. (2018), PvcUn seems to match Pvc17 (Table S8). The authors should make this clear in the main text, not only the supplementary text. 3) Thus, I suggest that the authors refer to 17 proto-vertebrate chromosomes, not 18, and when necessary refer to "PvcUn" as separate from the set. For example, on page 6, lines 4-5 - I suggest "Our reconstruction of the proto-vertebrate genome comprises 17 ancestral chromosomes, designated as Pvc1-17, as well as PvcUn, which consists of weak macrosynteny segments that could not be assigned to Pvc1-17." Please make sure that this is carried through for the whole text. Regarding the analysis matching PvcUn to scallop chromosome 13 as an argument for PvcUn representing a "true" ancestral chromosome, see my comment for page 23, line 18, in the attached document.

- I have some concerns about the description of the analyses of the proto-cyclostome genome reconstruction, and how the authors arrived at a hexaploidization scenario. My main issue is that these analyses have not been described well enough for me to make a judgment of whether the conclusions seem correct or not. For example, Jeremiah Smith and co-authors have suggested the involvement of a series of segmental duplications in the cyclostome lineage. How did the authors distinguish between genome hexaploidization and genome tetraploidization + segmental duplications? Simply calculating the "multiplicity" of genes would not address this. I have detailed some other concerns in the detailed comments (attached document) for pages 6-8 as well as for the supplements.

- In general, I miss a discussion of alternative scenarios in the paper. The authors mention alternative scenarios proposed by other previous papers like Mehta et al. (2013), Smith & Keinath (2015), Smith et al. (2018) and Sacerdot et al (2018), but I miss a discussion regarding whether any of these alternative scenarios could be possible with another interpretation of the results presented in the paper. In other words, can the authors definitely disprove any of the previous alternative scenarios? It would be helpful to the reader if the authors could discuss at least the one most likely alternative scenario. Why isn't a shared 1R/2R at the base of vertebrates followed by independent fissions/segmentations a likely scenario, for example? Something like this has been proposed by Jeremiah Smith and co-authors, based on the meiotic map of the previous sea lamprey genome assembly, and more recently based on synteny conservation of the latest sea lamprey germline genome. I concede that Smith and co-authors have gone back-and-forth and suggested partly contradictory scenarios, but it seems to boil down to one shared WGD together with chromosome-level segment duplications and fissions, possibly both preceding and following the WGD. Based on the current results presented in the present paper, why are these alternative scenarios less likely?

- Smith et al. (2018) also have the great advantage of dealing with the germline genome of the sea lamprey. As is well-known, lampreys greatly modify their genomes in the mature somatic cells, losing upwards of 20% of the genomic DNA. The authors describe that the DNA for the Japanese lamprey genome assembly was extracted from the mature testis (page 4 of supplementary information), while

Smith et al. (2018) specify that germline DNA was extracted from sperm cells of sea lamprey. I'm not entirely familiar with the methods for SMRT sequencing, but how confident are you that your Japanese lamprey genome assembly reflects the germline genome?

- I also have concerns regarding the annotation of orthologs vs. paralogs. The method is ingenious, although it has some limitations, and the principles behind it make sense. However, there are many pitfalls related to the fact that it is easy to misidentify orthology and paralogy with automatic annotations and gene trees, and with reciprocal BLASTP searches. I would want to make sure that these pitfalls have been avoided to the utmost extent. I would like the authors to describe the methods, the procedures, and the datasets in clearer detail in the supplementary information. As it is right now it would be nigh impossible for anyone to reproduce these analyses. See my comment in the attached document regarding page 18 of the supplementary information.

- The authors consistently write about implications for human disease, however, I cannot identify anything in the study that would further our understanding of the molecular/genetic mechanisms of disease, disease progression, treatment, etc, which is what is clearly implied by centering on human disease. Genetic diseases may reveal some constraints on genome evolution, which the authors discuss in a relevant way. But from this, there is a big step to talking about "implications for human disease". This reference to human disease must be tempered and put into the right context in the revised manuscript. Otherwise, this just looks like a transparent attempt to drive up the significance of the study by linking it to human disease. Surely the readers of Nature Communications can see through this, and I certainly don't think it was the author's intention.

- Finally, my spell checker kept changing "proto" to "photo", "port" or "protocol". I think I have identified the majority of these mistakes, but if there is a "photo-vertebrate" chromosome here and there in my responses, please overlook it.

It was a lot of work going through this manuscript in the detail that it deserves, but it was a pleasure to take part in these results before they are released. I apologize if my ignorance of some specific topics made me ask for a lot of clarification, but think of readers like myself who will benefit from this study without necessarily being experts in the intricacies of ancient genome reconstruction and macrosynteny algorithms.

I wish my colleagues all the best in the publication of this paper and I'm excited for it to come out.

Signed: Daniel Ocampo Daza
University of Uppsala, Department of Organismal Biology
University of California Merced, School of Natural Sciences

Specific comments for manuscript NCOMMS-19-37344-T - “Reconstruction of proto-vertebrate, proto-cyclostome and proto-gnathostome genomes provides new insights into early vertebrate evolution” by Nakatani et al.

Page 2, line 1: Is it necessary to center humans in this conversation? We are after all a very small part of this story. I suggest “The genomes of vertebrates, including humans, have been shaped by...”

Page 2, line 2: I suggest starting a new sentence at “... tetraploidization events. These have had a lasting impact...”

Page 2, line 3: Strike “However,”

Page 2, line 6: The authors suggest that the lack of a proto-cyclostome genome reconstruction has been a limitation in sorting out the timing of the cyclostome-gnathostome divergence relative to the early vertebrate tetraploidizations. The proto-cyclostome genome reconstruction is undoubtedly a great tool to resolve this issue, but the limitations truly lie with the lack of a reliable, mapped, cyclostome genome as well as the unique composition of cyclostome genomes and sequences. The authors discuss these issues in the manuscript. Can the statement in the abstract be tempered to reflect this? I suggest that this sentence can be removed completely without affecting the abstract.

Page 2, line 11-15: I suggest something like “**Our model suggests that** cyclostomes diverged from **the lineage leading to** gnathostomes after a shared tetraploidization...” In this same long sentence I suggest the following grammatical review - “; **that** the cyclostome lineage experienced...” “; **that** 2R in the gnathostome lineage **was an** allotetraploidization **event**...” “; and **that subsequently**, biased gene loss **from one of the** subgenomes...”

Page 2, line 13: It’s a tautology to write “the **cyclostome** lineage experienced a **cyclostome**-specific hexaploidization”.

Page 2, last sentence of Abstract: Again, this centers humans a bit too much in the story. The authors do mention the possibility of their findings informing our knowledge of human disease genes (I have some additional comments about this below), but because the authors have not identified any specific disease genes, not used any specific human disease genes as examples in this study, I think it is misleading to mention human disease genes in the abstract.

Page 3, line 2. The word “simple” can be removed. This is a common pitfall when writing about evolution. “Simple” in relation to what? Surely even these early chordates had some measure of complexity?

Page 3, line 4: Add comma - “... species, including humans.”

Page 3, line 9: Change to “Osteichthyes, represented by ray-finned fishes and lobe-finned fishes, including tetrapods”. The clade of lobe-finned fishes (Sarcopterygii) includes tetrapods, it’s not separate from it.

Page 3, line 10: I'm not sure that this opinion of cyclostomes is so general any more. Perhaps this could be changed to "Cyclostomes are **sometimes** thought to be..."

Page 3, line 13: I suggest "**seemingly** degenerate".

Page 3, line 15: Start a new sentence at "**F**or example,".

Page 3, lines 20-22: This sentence ("Evolutionary innovations...") is very long and tricky to follow. Please break up and clarify.

Page 3, lines 22-23: "This view is now widely accepted" seems to refer to the duplication followed by sub/neo-functionalization scenario, and not to the tetraploidizations themselves, which I think is the point. Please clarify.

Page 4, lines 7-8: Isn't "the tendency of lamprey ohnologs to cluster outside gnathostome gene clades" what is to be expected, i.e. isn't this the position that follows the taxonomy correctly? I know what the authors mean - that cyclostome sequences tend to occupy "paradoxical" positions in gene trees, but surely the position that the authors have described as "paradoxical" is the expected one?

Page 4, line 26: It's misleading to describe the species themselves as "early branching vertebrates". At least the lamprey is a **representative** of an early branching vertebrate lineage, but the cartilaginous fishes are just as "early" as the bony fishes, so this description is incorrect. Please clarify that the two species whose genomes have been sequenced and assembled **represent** two crucial divergence points in the evolution of vertebrates.

Page 4, lines 29-32: This sentence ("The major advantage...") is very long and tricky to follow. Please break up and clarify.

Page 4, line 33: Syntax error - "... we were able to reconstruct **the** first **the** proto-cyclostome genome..."

Page 5, lines 1-2: The statement "In addition, our reconstruction of the proto-gnathostome genome..." comes a bit prematurely. The authors have not yet stated that it was an aim to do this reconstruction, as they stated with the proto-cyclostome genome reconstruction on the previous page. I suggest "In addition, **we reconstructed** the proto-gnathostome genome **using the same strategy, with a higher coverage of extant gnathostome genomes than previous reconstructions...**" The authors have also neglected to mention that their sequencing and assembly of a new elephant shark genome was crucially integrated into this reconstruction. Highlight this fact - it's one of the major advances described in this paper! Similarly, the authors could highlight how crucial a chromosome-level assembly of a lamprey genome, compared with previous lamprey genome assemblies, was to their reconstruction.

Page 5, lines 7-8: The authors write that they "provide new insights into the genetic basis underlying evolutionary innovations". This is an overstatement. Surely, this is a possible future

impact of this study, but as for the present paper there is only a brief and very general discussion about the evolution of the adaptive immune system. That's it. Please temper the tone of this statement to something that reflects the content of this paper more truthfully.

Page 5, lines 8-9: This statement is only true if the authors will share the new genome assemblies in an easily searchable or browsable form, or, even better, share a detailed searchable map of their reconstructions. These possibilities are not mentioned at all in the paper. If the authors do not plan to share these resources, then the reconstructions will not serve as references of any kind.

Page 5, lines 14-16: This is a big overstatement. But to give this statement any credence, the authors should *at the very least* provide some examples and references of where this has been the case (I have more comments about this further down). They have not identified any specific disease genes linked to their findings, nor used any specific human disease genes as examples in this study. It is a pity because the study doesn't need it. There are many of us who follow the author's work and understand its value without centering it on humans and our pathologies.

Page 6, lines 30-32: The second clause of this sentence is tricky to follow. I suggest "... we predicted 18,727 **protein-coding genes in the elephant shark genome assembly** and 19,455 protein-coding genes in the Japanese lamprey genome assembly." This is only 5 words longer.

Page 6, line 3: If it does not make the manuscript exceed the word count, please detail which four gnathostome genomes here. This is important because if the elephant shark is one of them, the authors should highlight how essential their new genome assembly is for their analyses.

Page 6, line 5: Here is the first reference to "18 chromosomes". See my general comment about this above.

Page 6, lines 11-12: Since the names "scallop" and "placozoan" are used as general terms, and not as specific common names, the parenthesis around the binomial names *Chlamys farreri* and *Trichoplax adhaerens* should be removed.

Page 6, lines 12-14: Move this text ("also see Supplementary Fig. S3...") out of the parenthesis and make it a new sentence.

Page 6, line 20: Use commas around the sub-clause "that were not used in the proto-vertebrate reconstruction".

Page 6, line 25: Add "the" for "**the** Japanese lamprey".

Page 6, lines 25-16: Use commas around the sub-clause "in addition to the existing 'hybrid' genome assembly of the sea lamprey".

Page 6, line 28: Add a comma after "contentious".

Page 6, line 29 - page 7, line 2: This section, removing “For example”, should be moved down to just before the paragraph starting “To distinguish between different polyploidization models...” This way, these different models, which are complex scenarios, are still fresh in the mind of the reader. In addition, the alternative models of polyploidization seems as an aside, “just” an example”, the way they are described now. When, in fact, the reader must be familiarized with them to understand the rest of this section. The text can easily go from “... which have remained contentious, even after the sequencing of the sea lamprey genome”, to “In the present study, we have generated...” without losing clarity or jumping to a separate context (the alternative scenarios).

Page 6, line 32: Start a new sentence at “Another possibility...”

Page 6, lines 29-34: It’s not clear that the authors are referring to 1R here, the same tetraploidization (1R) is mentioned in two scenarios but makes it look like they are *different* tetraploidizations. I suggest “... could be due to additional tetraploidization events in the cyclostome lineage; alternatively, they could be the result of one shared tetraploidization event (1R) at the base of vertebrates followed by segmental (chromosome) duplications in cyclostomes. Another possibility is that the cyclostome lineage experienced a hexaploidization event (whole-genome triplication) following the shared 1R, thus giving rise to $1 \times 2 \times 3 = 6$ Hox clusters.

Page 7: Throughout this section of the paper I had a very difficult time distinguishing between blocks, segments, scaffolds and chromosomes. Some times a segment can be the same as a scaffold, right? And several segments can be “assembled” into a proto-chromosome? Where do “blocks” come in? Please define these terms clearly. This confusion is carried over to Figure 2.

Page 7, lines 4-8: This sentence is very long and difficult to follow. The authors should move the parenthesis to a new sentence following this, e.g. “... by combining lamprey genomic segments into 104 proto-cyclostome chromosomes (Figure 2). Genomic segments in this case are blocks of conserved synteny that were inferred...”

Page 7, line 6: Remove “the” from “the cyclostome evolution”.

Page 7, line 11: I suggest “because **each of the segments showed conserved synteny with two** different sea lamprey scaffolds.”

Page 7, lines 11- 16. Start a new sentence here, e.g. “In our reconstruction...” Furthermore, this sentence is very long and tricky to follow, and the references to Fig. 2 interrupt the flow and make it even more difficult to understand. I also have some methodological concerns here. I suggest the following: “In our reconstruction, the linkage of the two segments on Scaffold35 was restored in one of the proto-cyclostome chromosomes (green in Fig. 2b) with support from Pacific lamprey linkage markers. On the other hand, the two segments on Scaffold2 were assigned to different proto-cyclostome chromosomes based on the number of paralogs shared between them, which indicate an origin in a whole-genome duplication” I must say that the count of number of paralogs doesn’t convince me much - I can count (roughly?) the same number of dots, 12, in Fig. 2c between the two Scaffold35 segments and between the two Scaffold2

segments. Where do the authors draw the line for considering a number of paralogs as evidence for or against linkage? In addition - to invoke the linkage on Scaffold 35 as a proof that the segments indeed were part of the same proto-chromosome is a circular argument. Why then wasn't the linkage on Scaffold 2 seen as an argument for the ancestral linkage of these segments? This section of text as well as the paragraph that follows, makes the authors' analyses seem almost arbitrary, with "hand-picked" results, when they should rely on carefully considered algorithms. Please clarify this section of the paper so that the reader isn't left with the same impression.

Page 7, line 21: I've already suggested that the authors should move a section of text from the preceding page to this location of the paper. The paragraph starting here is very tricky to follow, starting with the first sentence. I suggest something like - "To distinguish between **these** alternative polyploidization models, we introduced a measure we have called multiplicity, i.e the number of **reconstructed** proto-cyclostome chromosomes that **correspond to each** of the **reconstructed** proto-vertebrate chromosomes." Avoid writing that multiplicity equals "the number of proto-cyclostome chromosomes **originating** from individual proto-vertebrate chromosomes" - This would be a circular argument. This describes a conclusion from the analysis, not how the analysis was made. The authors have not written here how this multiplicity was calculated, how the correspondence between proto-cyclostome and proto-vertebrate chromosomes was made, and I could not find a clear description of this in the supplementary text either. This again makes the analyses seem arbitrary and circular. It is briefly mentioned on page 33 of the supplement, but that's it. Is it part of section 3.3.3 on pages 27-28 of the supplement? The only reference to this "we extended it to also enumerating set partitions into more than 5 proto-cyclostome chromosomes." Is this it? Was the set partition with 6 proto-cyclostome chromosomes the most significant? In any case, describe briefly how this was done in the main text of the paper, and include a clearly marked "multiplicity calculation" (or similar) description in the supplementary text.

Page 7, line 24: Here is another mention of 18 proto-vertebrate chromosomes. The authors should write that they arrived at 17 proto-vertebrate chromosomes *plus* PrvUn. See my general comment above.

Page 7, line 24-25: The sentence "We found that nine out of the proto-vertebrate chromosomes **were duplicated** into six paralogous proto-cyclostome chromosomes." In my opinion, the authors should not write this conclusively about their results at this point of the paper. This statement is the **conclusion** that they arrive at, but for the reader it does nothing to explain **how** they arrived at this conclusion. What did the results look like? Are there any alternative scenarios that could explain the same results? If so, how were alternative scenarios discarded?

Page 7, line 28: Clarify that this first tetraploidization is 1R. For a moment I thought the authors suggested that both the tetra- and hexa-ploidizations occurred at the base of cyclostomes, which confused my reading of the paper.

Page 7, lines 30-34: This is a very long sentence that is difficult to follow. Please break up and clarify.

Page 8, line 1: The authors have not described how many proto-cyclostome chromosomes their reconstruction resulted in. This would seem like an obvious result to share, especially in the context of discussing the number of chromosomes in extant lampreys.

Page 8, line 8: I suggest changing “obtained” with “produced”.

Page 8, lines 10-11: It’s not clear here that the authors are describing their newly sequenced/assembled elephant shark genome. Highlight the fact that this genome assembly is new to this study.

Page 8, line 13: Change “confirmation” with “support”, or “additional support”.

Page 8, line 13-14: It was not the “proto-gnathostome” lineage that underwent the two tetraploidizations. At least 1R occurred in a “proto-vertebrate”. The authors found the evidence of 1R/2R in their “proto-gnathostome” genome reconstruction, but 1R occurred earlier. The authors should also be very clear to describe that 2R occurring in the lineage leading to gnathostomes is a new finding of this study.

Page 8, lines 13-14: “The proto-gnathostome lineage” could be a confusing term. If the time estimates for 1R and 2R that have been done previously are mostly correct, then it’s not at all certain that crown gnathostomes had emerged by the time 2R happened. A key fossil to date this node is the (likely) lobe-finned fish *Guiyu* at approximately 420 million years ago. The earliest fossil showing a bony jaw is the placoderm *Entelognathus*, a likely stem gnathostome also dated at approximately 420 Mya. This marks the minimum age of gnathostomes. The maximum age of gnathostomes is more difficult to estimate, but is bounded by the emergence in the fossil record of ostracoderms, at approximately 468 Mya. This time window overlaps with the suggested ages for 2R, but again it is not at all clear that crown gnathostomes had emerged at this point. Therefore, I think that it would be more accurate to write “the lineage leading to extant gnathostomes” instead of “the proto-gnathostome lineage”.

Page 8, lines 16-22: This paragraph about microchromosomes seems to interrupt the flow of the text. Perhaps it could be shortened and moved down to the following paragraph, after “... even after ~450 million years of gnathostome evolution.” The first sentence of the paragraph ““Analysis of the proto-gnathostome genome also revealed...””) could then be removed.

Page 9, line 17: Add comma after “hypothesis”.

Page 9, line 18: I suggest “... high density of genes (**including ohnologs**) in **the** proto-gnathostome chromosomes...”

Page 9, lines 16 and 18: Ohnologs are mentioned, but there is no description in the main text of the paper, however brief, of how ohnologs were identified/predicted or differentiated from other forms of orthologous genes. There is a good description in the supplementary information, but the main text of the paper should give *some* understanding of this. Especially because it is mentioned in the introduction that “our reconstructions serve as a reliable reference for accurate annotation of ohnologs.”

Page 9, lines 22-24: This sentence is tricky to follow I suggest - “The timing of gnathostome-cyclostome divergence relative to the two basal vertebrate tetraploidization events (i.e. 1R and 2R) remains an unresolved issue in the field of vertebrate **genome** evolution. Remove the reference to 1R/2R occurring in “proto-gnathostome lineage”. This is incorrect. See also my comment above regarding “the lineage leading to extant gnathostomes” rather than “the proto-gnathostome lineage”.

Page 9, line 24-25: I suggest “we searched **our reconstructions of the** proto-vertebrate...”

Page 9, line 27: Remove the parentheses and insert a comma after “models”.

Page 9, line 32: I suggest “... before 2R, **but after 1R.**”

Page 10, line 2: Regarding the text in parentheses, “or diverged even before 1R”. This is a much bigger discussion and should not be relegated to a parenthesis. If this were true, then the authors’ own proposed scenario would be consistent with independent 1R events in cyclostomes and the lineage leading to gnathostomes. What in their results, and indeed in previously published studies, suggests that this is a possibility? To the best of my knowledge, the evidence points away from this conjecture.

Page 10, line 5: When the authors write “we performed a gene-tree analysis”, it gives the faulty impression that the authors created these gene trees themselves. In fact, the authors have analyzed automatically generated Ensembl gene trees. This is a possible weak point in the analyses, so the authors should clearly describe what they have done.

Page 10, lines 10-22: This section is very difficult to follow. It seems like a substantial part of the description of results and the arguments are missing. The authors state **that** they arrived at certain conclusions, but it is not at all clear to the reader **how or why** they arrived at these conclusions. Not all of the argumentation should be left to the supplementary text. For example, on line 11 the authors describe “homeologous proto-gnathostome and proto-cyclostome chromosomes”, but calling them homeologous is a conclusion in itself. How did they arrive at this. The following subclause, “seemingly suggesting a contradictory model...” is very unclear. How could both quadruple and sextuple chromosomes arise at the same time? I think they authors simply suggest that this is evidence for a shared tetraploidization at the base of vertebrates, i.e. 1R. How is this a “contradictory model”? Contradictory to what? It is near impossible to distinguish between paralogs generated in 1R and those generated in 2R (although the authors have made a good attempt at dating them by analyzing Ensembl gene trees), but a large amount of 1R generated paralogs shared between gnathostomes and cyclostomes is not contradictory to independent chromosome rearrangements in each of the lineages. Or have the authors been able to date the paralogs so precisely that this set of paralogous genes includes both 1R- and 2R-generated paralogs? Also, be sure to clarify that the hypothesis of 2R being a gnathostome-specific event is based on **their** result and this study. The fact that 2R might be gnathostome-lineage-specific doesn’t necessarily mean that it is a **later** event. The estimations of time-points for 2R, the emergence of crown gnathostomes, and the gnathostome-cyclostome divergence all overlap, and the authors have not done a time-estimate calculation of their own.

Page 10, line 17: Add “the” before “establishment”.

Page 10, line 19: I would suggest that polyploidization through hybridization is common “**to some extent**” in animals.

Page 10, line 27: Here is another reference to 18 ancestral chromosomes when it should be 17 (see general comment above).

Page 11, lines 2-3: “, which can be explained by allotetraploidization” is a repetition and can be removed.

Page 11, line 2: Add the indefinite article “A” to “A comparison with modern...”

Page 11, line 9: Another reference to 18 ancestral chromosomes. Also, the formula $18 \times 2 \times 3$ can be misleading. It's not clear here that “x2” refers to 1R. Also, the authors have not revealed how many proto-cyclostome chromosomes their reconstruction ended up in. Was it as neat as $18 \times 2 \times 3 = 108$? If so, they should mention very clearly, somewhere in the text, whether their estimation of the number of proto-cyclostome chromosomes was constrained by the 18 (17, really) proto-vertebrate chromosomes they had already reconstructed.

Page 11, line 16: “Evolutionary hexaploidy” is not an accepted term and could be confusing. Simply removing “evolutionary” would clear it up. Alternatively, I suggest something like “There are several documented examples of hexaploidy giving rise to new evolutionary lineages”.

Page 11, lines 25-26: The authors of this study are not the first to suggest this. See *Vertebrate evolution by interspecific hybridization – are we polyploid?* by Jürgen Spring in *FEBS Letters* 400, 2–8, 1997, for an early-ish example. They are not the first to suggest that hybridization played a role at the early stage of vertebrate evolution. In more general terms, hybridization has been part of the discussion since Susumu Ohno's time - he writes about it in the “Mechanisms of Gene Duplication” chapter of *Evolution by Gene Duplication* in reference to both auto- and allo-tetraploidy, and he mentions triploidy, though he does write that “Such an interesting oddity, however, is a side issue of vertebrate evolution.” At this point of the paper, the authors should perhaps temper their discussion to reflect the long ongoing discussion surrounding the role of hybridization in polyploidization and the origin of vertebrates. In the supplementary text, the authors contrast “their” hybridization scenario against the “octaploidy hypothesis”. This makes a neat and tidy way to launch hybridization as a new hypothesis, but it has in fact been discussed previously. What's exciting about this paper, is that it adds evidence to this ongoing discussion.

Page 12, lines 2-4: This sentence highlights an issue with this whole section of the discussion: suddenly the authors are describing the proto-gnathostome genome rather than the proto-vertebrate genome... Do they mean to say that only 2R, and not 1R, was an allopolyploidization event? Why not 1R? This is especially confusing since the authors started the section talking about the proto-cyclostome genome and hexaploidization. It should be **abundantly** clear which tetraploidization events they are referring to.

Page 12, line 2: I would change “shows” to “suggests”.

Page 12, lines 10-11: I suggest “... throughout most gnathostomes, [comma] including cartilaginous fishes, but are **missing** in invertebrates, [comma] including the closest relatives of **vertebrates**, such as **tunicates** and amphioxus.”

Page 12, line 13: Add a comma after “events”.

Page 12; lines 30-31: It’s not clear whether MHC, NKC and LRC were located on **different** microchromosomes or the same microchromosome. The authors write about *cis*-preserved genes on the next page (line 2), but the context we are in as readers is tetraploidizations, which suggests different chromosomes... The authors use microchromosomes in plural on page 12, line 31.

Page 12, line 30 - page 13, line 7: The authors have traced the **locations** where there would be MHC, NKC and LRC genes back to early vertebrate evolution, but are there any indications that the genes themselves were present at this time? After 1R? After 2R in gnathostomes?

Page 13, lines 9-22: I think this section is overstated. See my comment above regarding page 5, lines 14-16. The fact that some ohnologs are human disease genes is underwhelming. Of course they are. There are many more that are **not**. The studies the authors have cited are more concerned with dosage issues in anciently polyploid genomes such as ours, and that when those dosages in the re-diploidized genomes are perturbed, by copy-number variations for example, they may result in disease. This is interesting in terms of genome evolution and the constraints upon genome structure and evolution, which are revealed when disease arises. In these terms, there is a connection to the present study, and this study adds to the knowledge about constraints on genome evolution. But from there it is a big step to say that this study has “implications for understanding human genetic diseases”, which suggests implications for disease origins, disease progression or even disease treatments. Please restate this section, and the section at the end of the introduction on page 5, in terms of constraints on genome evolution, rather than by linking it to human disease.

Page 13, lines 28-32: Several statements in this concluding section need to be tempered down a bit. On line 28 - “contentious” is perhaps a bit strong. I suggest “our reconstructions address several unresolved issues”. Regarding “the origin of the adaptive immune system”, the authors have provided a brief and very general discussion about the evolution of the adaptive immune system. This statement should be understated somewhat. The reference to human diseases should be left out.

Figure 1: Most of the figure caption is not relevant for the graphical interpretation of the figure. If the results or the methodology are not described well enough in the main text, change the main text instead of adding this much information to the figure caption. For example, the whole section between lines 2-8 should be removed (“We reconstructed the...”). The final sentence of the legend also does not belong here. The caption can be shortened further by changing to “The

Trichoplax and elephant shark scaffolds were sorted...” to avoid repetition. As for the figure itself, it would be useful if the 17+PvcUn chromosomes were enumerated in the y-axis.

Figure 2: It should be clear that the figure shows examples and not the full data. Again, there is some confusion of terms between scaffolds, segments, subgroups and chromosomes. I suggest the following to perhaps clarify this - “Japanese lamprey scaffolds **(a)** were correlated with proto-vertebrate chromosomes (Pvc). Scaffolds corresponding to Pvc3 are shown in blue and to Pvc17 are shown in green. Segments of conserved synteny from the lamprey scaffolds were clustered into proto-cyclostome chromosomes **(b)** based on the distribution of paralogs vs. orthologs. The triangular plot **(c)** is a 45-degree-rotated graph of the paralog distribution **between** the 12 proto-cyclostome chromosomes that correspond to Pvc3 and Pvc17. This shows...” The description of the multiplicity table is too long, and most of it is not relevant for the graphical interpretation of the figure. The figure caption is already too long.

Figure 3: There is too much description of results and discussion in the figure caption that is not necessary for the graphical interpretation of the figure. The whole section starting “The segment lengths are longer in human...” and ending “... and the large macrochromosomes” does not belong in a figure caption. The same is true for “In general, smaller proto-gnathostome chromosomes [...] and large chromosomes with low gene densities” and “As in the gene density plot [...] with high ortholog densities.” There is also some confusion between “segment length” and “chromosome size” for this figure. The definition of “segment” should be abundantly clear in the main text as well as the figure caption.

Figure 4: I don’t think the authors should include PvcUn in the evolutionary scenario, nor mention 18 (rather than 17) ancestral chromosomes in the caption. PvcUn is a construction of many small sections with weakly conserved synteny that likely “belong” in other chromosomes. It’s a “waste basket” construction, if I’ve understood their methods correctly. The inclusion in the evolutionary schematic gives the wrong impression that it represents a pair of ancestral chromosomes. The grey areas that correspond to PvcUn can be left in the images of the modern genomes, if it’s clearly described in the caption that the grey color corresponds to PvcUn regions. How strong are the conserved synteny that indicate that elephant shark scaffold 25 and chicken chromosome 24 are derived from PvcUn? If it’s only a handful of genes, I would at the very least mark these as striped and not completely filled in with grey color.

Figure 4: The authors have not included any rearrangements or drawn lines between the proto-cyclostome chromosomes and the extant lamprey chromosomes. It’s difficult to see the evidence of the hexaploidization in the lamprey genomes otherwise. If the reader doesn’t have any sort of Then why include the lampreys at all?

Figure 4: The caption suggests that all macrochromosomes in extant gnathostomes resulted from the chromosome fusions that preceded 2R, and that all chromosomes that didn’t fuse resulted in microchromosomes. How can this be? In this figure alone I can see that, for example, chromosome 14 in humans, arguably a macrochromosome, is derived mostly from a Pvc17-derived proto-chromosome, which did not experience any fusions. Even *if* all macrochromosomes are derived from ancestral chromosome fusions, surely not all fusions occurred at the base of vertebrates?

Detailed comments on Supplementary Information:

Page 4, line 5: What was the origin of this elephant shark? The geographic area where it was caught, but also the conditions by which it was caught. The elephant shark is classified as a “Least Concern” species by the IUCN (<https://www.iucnredlist.org/species/41743/68610951>), but it occurs within protected areas, and there are conservation plans in place across its entire geographical range, so this information is important. This information also provides additional assurance that the right species has been used.

Page 11, line 5: The same as above for the Arctic lamprey. How was this animal procured and from which geographic range? In America, the Arctic lamprey could co-occur with the closely related Alaskan brook lamprey (*Lethenteron alaskense*), and in Asia it co-occurs with the Far-Eastern brook lamprey (*Lethenteron reissneri*). The Siberian brook lamprey (*Lethenteron kessleri*) is some times classified as a sub-species of the Arctic lamprey.

Page 12, line 5: How does this genome size compared with the previously publishes genome assembly of the Arctic lamprey? And of the latest assembly of the sea lamprey?

Page 14, lines 21-24: Were these TRINITY transcriptome assemblies from the same individual as the genome assembly? It’s not clear whether these transcriptome efforts were part of the same genome project described in this paper. This should be made clear in the text. The Institute of Molecular and Cell Biology at A*STAR is cited as the source of the RNA-Seq reads in the BioProjects database, which is the home institute of several of the authors.

Page 18, lines 1-9: The methods described in this paragraph are not entirely clear. For example, “We obtained orthologs and paralogs from gnathostome species...” What does this entail specifically? What kind of dataset was obtained from Ensembl? Sequences? Spreadsheets with annotation IDs and locations etc? How were these obtained from gene trees? Usually Ensembl datasets are obtained through BioMart. Was the complete set of gene trees in Ensembl 75 downloaded? If so, this dataset must have included much more data than only phylogenetic data. For example, it must have included some of the annotation data created by Ensembl, because the authors mention that they looked at whether gene duplicated were annotated as Vertebrata, Euteleostomi or Clupeocephala. Were the trees simply analyzed visually on the Ensembl website? This would be a monumental task. If only *some* Ensembl gene trees were analyzed, how were they selected for analysis. How was the tree data analyzed specifically? The authors write, for example, that small-scale duplicates were discarded. What does this entail specifically? What did their final dataset consist of? What kind of data? So much of the final evolutionary scenario hinges on these analyses, but I haven’t been able to scrutinize it to the level I would like to because I don’t find the information. For example, the analyses hinge on identifying whether gene duplicates are paralogs, but I can’t see how the authors have identified that two genes are duplicates to begin with. How did they positively identify duplicates, specifically.

In general, it would be valuable if the authors described exactly how many orthologs vs. paralogs they identified and included in their dataset. I would also urge the authors to share these datasets either as a supplementary file with the publication or in an online repository, if possible. Unless

this data includes tens or even hundreds of thousands of genes, then I would understand it is not feasible. However, it would be especially relevant for the elephant shark reciprocal BLASTP searches described on page 18, lines 7-9, because it would be important to know how many orthologs they identified, and as a reader I would like to review this list to make sure that the orthology assignments were (mostly) correct. This also goes for the amphioxus/human and lamprey/gnathostome ortholog searches described further down on the page. If it's not feasible to share the resulting datasets, at least describing the searches in more detail would help give the reader an indication of what the results were like. Because, in addition, it is not clear against which datasets/databases the BLASTP-searches described on this page were done. For example, "We performed BLASP search[es] for all species pairs, and identified orthologs and paralogs..." What species pairs? Which gene dataset was used as queries and which datasets/databases were searched? I understand the logic of simply using the top 2 or 4 scoring genes for the BLASP searches, but there is a large potential for mis-matches. I would like at least the possibility to quickly scan the resulting orthology/paralogy assignments to verify, or at the very least know which datasets were used as queries and which ones were searched in order to ensure reproducibility.

Page 18, line 29: What search were these bit-scores derived from. Describe the procedure clearly.

Page 18, line 29: All three conditions or only 1 or 2 of them? It's not clear.

Page 18, line 30: Describe that lamprey vs. amphioxus BLASTP searches were done earlier in this section. Does this refer to the same BLASTP search as the lamprey gene pair bit-scores in the preceding line? The following line also seems to refer to BLAST-searches against sea lamprey genes...?

Page 18, line 26 - page 19, line 8: This section describes the annotation of lamprey paralog genes. It is logical that the authors would consider paralogous gene pairs in lamprey, as described on page 18, lines 19-29. But it is not clear from this section, nor from the main text of the paper, how paralogous gene **pairs** helped identify **hexaploidization** in cyclostomes. I understand that the **distribution** of gene pairs across three ancestral chromosome pairs would still indicate hexaploidization, but if this was the authors' thinking, it should be better described. The information I miss from this section is whether any gene **triplets** were identified, and if so, how many?

Page 19, lines 2-5. I don't understand this reasoning at all. Please clarify. It is not clear what "the pair" are, or what "either of the lamprey genes" refers to. Remove the parenthesis around "We retained seven paralogs..." Also, clarify that the expectation of three rounds of WGD (1R, 2R and a cyclostome-specific WGD) is the hypothesis that they were working with based on the previous suggestion in *Mehta et al. (2013)*. It's important to highlight this because the actual scenario that this study resulted in is different! One WGD (1R) and one hexaploidization! The maximum expected number of paralogs after 1R and then a cyclostome-specific hexaploidization would be $1 \times 2 \times 3 = 6$? At first I was confused because I thought the authors were referring to the latter, not the initial hypothesis. Why 7 though, and not 8?

Page 19, lines 13-15: This section is similarly confusing. What does “the elephant shark gene pair” and “neither of the elephant shark genes” refer to?

Page 21, line 3 (below the algorithm): I suggest “**the** proto-vertebrate genome”.

Page 21, line 4: Clarify **which** lamprey genome.

Page 21, line 4: When the authors write simply “comparing the lamprey genomes with each other and also with four gnathostome genomes...” it reads like they are not explaining further what these comparisons entail. It is not immediately clear that they are referring to the sections that follow (3.2.1, 3.2.2 etc). Please clarify.

Page 23, line 2-2: “The reconstruction with $K = 18$ was the most significant.” Could the authors please share the full results of this? What was the significance **value** of $K = 18$? What values did other K s produce?

Page 23, line 14-15: “Syntenic to” does not mean what the authors mean here. Syntenic means that two genes are located on the same chromosome. I suggest “**A comparison of conserved synteny between these proto-vertebrate chromosomes and the scallop genome shows that Pvc17, PvcUn, Pvc8, and Pvc9 correspond to individual scallop chromosomes - chromosomes 3, 13, 6 and 4 respectively.**”

Page 23, line 18: It’s not clear what the authors mean by “in early invertebrate lineages”. Early invertebrates as in at the base of the metazoan lineage (this is very *very* early), or early as in already in an invertebrate ancestor or extant chordates/vertebrates.

Page 23, line 18: I’m still not certain that PvcUn actually represents an ancestral chromosome. Clearly, there is not perfect correspondence between the proto-vertebrate genome reconstruction and the scallop genome, as shown in Figure 4. Because the conserved synteny comparison was one-sided, i.e. proto-vertebrate \rightarrow scallop, it’s not possible to differentiate between rearrangements in the proto-vertebrate or rearrangements in the lineage leading to the scallop. Doing the analysis the other way, scallop \rightarrow proto-vertebrate, might show that parts of scallop chromosome 13 correspond to other Pvc’s. So for a large number of segments of weak synteny conservation (i.e. PvcUn) to show conserved synteny with a single scallop chromosome is not definitive evidence. Did all the segments of PvcUn correspond to scallop chromosome 13, or where there segments in PvcUn that could not be assigned? The authors have not described this. Also, they haven’t described how big the conserved synteny segments that make up PvcUn are. I suspect they are very small, which makes any conclusions very tentative.

Page 23: It is notable that the authors haven’t discussed here why these results are so different from the previous reconstruction of the vertebrate genome by the first author (*Nakatani et al. Genome Res. 17(9), 2007*), which reconstructed only 10 ancestral chromosomes. Which scenario is wrong? Is this completely due to the inclusion of a cyclostome in the reconstruction? *Putnam et al. (2008)* didn’t include lamprey synteny and still arrived at 17 ancestral (chordate) chromosomes. I have to ask, also, for the *Nakatani et al. (2007)* ancestral chromosomes to be included in Table S8. This would be very useful.

Page 24, line 7: It can't hurt to add the binomial nomenclature for the silkworm and sea anemone as well.

Page 24, lines 14-15: It is not clear what "assigned scaffolds to **the chromosome** with the largest number of markers" refers to. The proto-vertebrate chromosomes?

Page 24, lines 18-19: I'm not so sure. This suggests that the patterns of synteny are conserved, it says nothing of chromosomes themselves. For example, it does not consider chromosome fissions preceding the time point of the proto-vertebrate reconstruction. What I see in Fig. S3 is that **these particular** conserved synteny patterns, inferred to have existed in early vertebrate evolution, can be "recreated" **to some extent**, by no means perfectly, in invertebrate genomes as well. However, genomes are mixes of different patterns, syntenies and paralogies of different origins, and this study does not address other patterns that may exist in the invertebrate genomes that may indicate other ancestral chromosome configurations. The analyses in these studies were done in only one direction, proto-vertebrate → invertebrates. Starting with another lineage at the outset may reveal other chromosome configurations in the common ancestor.

Page 25, line 4: Change to "**have** remained contentious".

Page 25, line 5: Change to "**the** possibility of cyclostome-specific WGD..." I also suggest removing "intense", as this is a value judgment.

Page 25, line 8: Change to "... WGD, **followed by the** loss of two entire clusters".

Page 25, line 10: Change to "We considered that **a** reconstruction of **the** proto-cyclostome chromosomes..."

Page 25, line 12. Change "comprises" to "comprise".

Page 25, lines 14-15: Change to "Thus, **the** reconstruction..."

Page 25, line 17: Change to "**The** enumeration..."

Page 28, line 3: Change to "**in** the proto-vertebrate lineage..."

Page 28, lines 2-6: Perhaps this is unrelated, but does it then follow that for the proto-cyclostome reconstruction the most significant partition was $6 = 1R$ followed by hexaploidization?

Pages 29-30: The "red/black/white/grey" metaphor is quite long-winded and very difficult to follow. Please break up and clarify.

Page 31, lines 1-2: Please clarify that the "previous reconstruction" has the same first author as this study. Otherwise we might get the impression that Dr. Nakatani is (unfairly) disowning his previous work.

Page 31, line 5: Regarding the “nine large-scale rearrangements”, I counted nine fusions. How about fissions?

Page 31, line 26: Change “fission” to either “**the** fission” or “fissions”.

Page 32, line 10: Change “chromosomes” to the singular “chromosome” or write “For each **of the** proto-gnathostome chromosomes...”

Page 32, lines 22-23: I suggest “These chromosomes underwent **the first** WGD (1R), [comma] resulting in the **doubling** of the proto-vertebrate **genome**.” Remember that we are generally talking about the **haploid** genome here. “Doubling” of chromosomes could be misinterpreted as referring to the diploid genome.

Page 32, line 23: Change “In the gnathostome lineage” to “In the lineage leading to extant gnathostomes”, see my comment about page 8, lines 13-14, above.

Page 33, lines 6-10: I suggest “**Where** our reconstruction **produced** less than six chromosomes, the remaining chromosomes **out of the expected six are** shown as hatched bars. **Where** our reconstruction **produced** more than six chromosomes, the extra chromosomes **are not shown**. **However**, the extra chromosomes were included in all other figures, [comma] including Figures 1 and 2, although **they are very** small.”

Page 33, line 12: Change “Modern” to “Extant”.

Page 33, lines 15-16: It seems strange to me that so many, and in some cases extensive, “white regions” can be explained to be only centromeres. Perhaps if including also pericentromeric areas, which do contain **some** genes. It’s a small point, but in any case, this is only a conjecture on the authors’ part. In addition, writing “regions excluded from our reconstruction” makes it sound like the authors excluded these regions **purposely**, which I don’t think was the case. I suggest writing “**Regions of the human genome shown in white likely correspond to regions poor in genes, such as centromeres and pericentromeric regions.**” The authors should be careful not to give the false impression that they are showing the complete chromosomes in their reconstruction (Fig. 4). I don’t see centromeres/pericentromeric regions, telomeres and other “gene deserts” in the figure. These can be more closely described as conserved synteny blocks for each of the chromosomes.

Page 33, line 26-29: I suggest “... we plotted paralogs among proto-gnathostome and proto-cyclostome chromosomes **and classified them** into vertebrate **paralogs** (i.e. duplicated in the common ancestral vertebrate), **cyclostome-specific paralogs, and gnathostome-specific paralogs** as described below.”

Page 33, lines 30-31 - Page 34: I suggest removing "Paralogs in the proto-gnathostome genome were represented by human paralogs obtained from BioMart:" and simply starting the sentence as follows - “**Human** paralogs annotated as Vertebrata **in Ensembl** were classified as vertebrate paralogs (blue dots), [comma] and **human** paralogs annotated as Euteleostomi were classified as...”

Page 34, lines 2-3: I suggest "Figure S9 shows the distribution of vertebrate and gnathostome-specific paralogs **mapped onto the reconstructed** proto-gnathostome genome."

Page 34, line 21 (Step 3): "We deleted irrelevant genes from the tree" - This is a very reckless formulation. Who decides what is irrelevant? Instead, describe and defend your criteria clearly and methodically.

Page 34, line 26 (last line): Replace "branching pattern" with "tree topology".

Page 35, line 4: Replace "should be clustered" with "would cluster".

Page 35, line 6: Use the plural "annotations".

Page 35, line 20: Replace "the one third of high-GC genes" with "the third of the genes with the highest GC content".

Page 35, line 25: Make sure that you have described earlier which sea lamprey assembly you have used for these analyses. Is it the latest germline genome assembly version, or the much poorer previous assembly? In any case, it doesn't hurt to remind the reader here as well.

Page 35, lines 25-16: I suggest "The annotation of **sea lamprey paralogs was done** by using RAxML-EPA with **the** WAG matrix (method A), **and** is shown in Figure S13."

Page 35, lines 30-31: The authors refer to the supplementary figures (Fig. S9-S13, and Fig. S14 on the next page) when they write about Hox genes, yet the Hox genes are not marked out in these figures. How will the reader verify that this is correct?

Page 35, line 28 - page 36, line 4: It would be helpful if the authors could discuss the most likely alternative scenario that could explain the same results. Why isn't a shared 1R/2R at the base of vertebrates followed by independent fissions/segmentations a likely scenario? Something like this, shared 1R followed by independent chromosome-level segment duplications and fissions, has been proposed by Jeramiah Smith and co-authors, for instance, based on the synteny conservation of the latest sea lamprey germline genome. Based on the current results presented in this papers, why are these alternative scenarios less likely? This is something that I miss in this paper in general.

Page 36, lines 20-21: The sentence starting "It was previously shown..." is difficult to follow. It's not clear what the "branching patterns" of the human genome refers to. It might just be that a lot of information is packed very densely into this sentence. Please clarify.

Page 37, lines 2-3: I suggest "Figure S14 **suggests** that **a** majority of ohnologs..." It's not entirely clear how this figure shows sequence divergence. Only panel a in the figure seems to show this, is that right? Please clarify.

Page 37, line 4: The authors write “two out of four” but I can’t really see this in the cited figures. Some guidance would be good. In addition, the figure caption for Fig. S14 mentions “two out of **six**”...

Page 44, Figure S3: The y-axis designation “Proto-vertebrate/-cyclostome” is seemingly contradictory. I understand that these are the Japanese lamprey scaffolds, but it is confusing to lead with a seemingly contradictory statement. They can’t be proto-vertebrate and proto-cyclostome chromosomes at the same time. I suggest changing the formulation “proto-vertebrate/proto-cyclostome chromosomes represented by Japanese lamprey scaffolds...” to simply “The Japanese lamprey scaffolds were compared with invertebrate genomes (x-axes). In this way we could validate both the proto-vertebrate and proto-cyclostome chromosome reconstructions. Horizontal orange lines represent the boundaries of Japanese lamprey scaffolds and black horizontal lines represent the boundaries of the corresponding proto-vertebrate chromosomes.” This should be applied to all the similar figures - Fig 1, Figs. S2, S3, S4, S6, S7 - and within the figure captions and manuscript text. Name the y- and x- axes for what they actually show, not what they “represent”. In addition, I cannot see any horizontal grey lines in the figure - they are mentioned in line 5 of the figure caption. I also can’t see the difference between thick and thin vertical lines - mentioned in lines 7-8 of the caption.

Page 44, line 11 (last line of figure caption): See my comment above regarding page 24, lines 18-19. This shows that the synteny patterns can be recreated **to some extent** in invertebrate genomes, but it doesn’t definitively show that they represent ancestral metazoan chromosomes. Be careful with this conjecture.

Page 46-47: This figure caption is inordinately long. Please include only information necessary for the **graphical** interpretation the figure. Everything else should go in the supplementary information text, if it’s not there already. The description of this procedure is very good, it should be part of the main text, not a caption!

Page 48, Fig. S6: It would be very helpful to enumerate Pvc1-17 and PvcUn on the X-axis of the figure, and the proto-gnathostome chromosomes on the y-axis. The caption of this figure illustrates my comment about alternative scenarios. The authors very clearly describe their scenario, and highlight the data which illustrate their point very well. But can they disprove/falsify alternative scenarios? Can this same data illustrate any of the alternative scenarios? What would the data look like if the most likely alternative scenario were true? Could the rearrangements not be post-2R or pre-1R fusions? This analysis doesn’t differentiate between 1R-generated and 2R-generated paralogs. Help the reader navigate these alternatives.

Page 48, Fig. S6: There are some curiosities in this figure that are not mentioned. Notably, the orthology between Pvc17 and proto-gnathostome chromosome 9. Wouldn’t this result from a large-scale fission? When did this occur? The authors have not mentioned fissions in the paper.

Page 48, line 3: Correct “axe” to “axis”.

Page 49, Fig. S7: The horizontal grey lines are barely visible, even when I zoom in on the PDF.

Page 49, line 1: “Comparison with the lampreys and amphioxus genomes.” Comparison of what? Instead of writing “proto-gnathostome” at the y-axis, describe what it actually shows. Correct “lampreys” to “lamprey”.

Page 49, line 7: I can’t tell the difference between thick and thin vertical lines in the figure.

Page 49, lines 8-9: Explain that the 1:4-orthology between proto-vertebrate and proto-gnathostome genomes is shown in the amphioxus panel of the figure, if I’ve understood this correctly. Perhaps it would also be better to order the panels of the figure in the inverse order. In general, it is quite difficult to relate the caption to the figure. Doesn't the two lamprey panels show that both 1R and 2R occurred **after** the divergence of cyclostomes? It shows the same relationships as the amphioxus panel. Very tricky to know what to look at.

Page 49, line 12: None of this numbering is shown in the figure, so it’s very difficult to know what to look at.

Fig. S9 - Fig. S13: Please describe what the x- and y-axes of these figure represent.

Page 56, Fig. S14: I almost gave up trying to interpret this figure. It is incredibly information-dense and there are seemingly some missing parts? Why are there no triangular plots for the upper scatterplots? Please write out next to the rectangular scatterplots what they actually show. For example, I’ve mocked up an image for panel a...

Replace the numbering in orange for the actual chromosome numbers. This was useful for me to see the 1:4 and 1:6 relationships between the proto-vertebrate and the proto-gnathostome and proto-cyclostome, respectively. For the bottom scatterplot, it would also be clearer to use black lines, not orange to mark the boundaries of the proto-cyclostome chromosomes. Because the top and bottom scatterplots are so similar, I was expecting that Pvc1 and Pvc17 were also plotted in the bottom scatterplot. This would avoid the confusing “bottom and left”, “bottom and right”, “bottom six”, “middle two out of six”... give them numbers! I still don’t know what “middle two out of six” refers to.

Page 56, line 13-14: Perhaps it would be better to note what the figure **does** show, rather than what it **doesn’t** show? I.e. the 1:4 relationship between the proto-vertebrate and proto-gnathostome reconstructions, and the 1:6 relationship between the proto-vertebrate and the proto-cyclostome reconstructions. To be fair, only panel a shows this undoubtedly, but you can argue for panel b and c, which I suspect are the more common occurrences. Also, it would be helpful to know what it would look like if indeed there was 1:1 orthology relationship - i.e. what if the alternative hypothesis is true? Can the data be described with alternative scenarios?

Reviewer #2:

Remarks to the Author:

The manuscript "Reconstruction of proto-vertebrate, proto-cyclostome and proto-gnathostome genomes provides new insights into early vertebrate evolution", by Nakatani et al reports improved genome assemblies for two species (elephant shark and Japanese/Arctic lamprey) and uses these genomes to reconstruct whole genome duplication events, using reconstruction algorithms that have not previously been applied to the problem. These are presented as lending strong support to specific whole genome duplication scenarios. However much of the information necessary to assess the reconstructions is unavailable to the reader, and the analysis of reconstructions does not effectively test their favored hypotheses against previously-proposed hypotheses or others that seemingly emerge from their analyses. Moreover, a more thorough discussion of the biological underpinnings of their proposed evolutionary mechanisms would be welcome, and necessary for readers to understand the implications of the presented analyses. There seem to be relatively straight forward remedies to these issues, which are outlined in the comments below.

Comments:

- 1) First, use of the term "Proto-Cyclostome" is seemingly inappropriate with respect to the reconstructions that are presented in this paper. The lineages leading to sea lamprey and Japanese lamprey diverged approximately 20-30 million years ago. Therefore the hypothetical reconstructed ancestor would more appropriately be called the Proto-Petromyzontid ancestor. This refers to a branch that extends to ~250 MYA at which point the petromyzontid lineage is thought to have split from Geotria lampreys. Without data from other lampreys or hagfish, it seems like over-reaching to call the reconstruction "Proto-Cyclostome".
- 2) The authors state that "Whether microchromosomes were recently created by chromosome fission, or were present in the gnathostome ancestor has been controversial". In my impression this does not accurately reflect the recent state of literature. Multiple analyses of various genomes, including most notably amphibians, gar and lamprey in comparison to birds and elephant shark have seemingly firmly established this.
- 3) In general the authors should strive to more fully articulate alternate models and specifically test the fit of those models to observed patterns across extant genomes, not simply the reconstruction that is optimal under their algorithm. One example of this is the assertion that the numbers of Proto-Petromyzontid chromosomes/segments supports a post-1R triplication. The distribution of paralogous segment counts peaks at 6, which is considered evidence of duplication followed by triplication. However, it should be noted that a simple model of random segmental duplication would also be expected to yield a peak with mean = 6. Constraining this pattern assuming 1R substantially sharpens this peak. Based on a quick permutation test, 1R plus random duplication seems to be a better fit to the observed distribution than 1R + triplication. It is probably also worth considering 1R + duplication and other models. Admittedly, a more formal statistical approach related to the birthday problem of hash collision might provide a more elegant solution than permutation.
- 4) A second comment related to this is that the numbers presented in figure 2d should refer to the numbers of ancestral genes that are incorporated into these classes, not the number of lamprey genes (as these include duplicates).
- 5) Related to this, it would be very useful if the authors could provide the number of orthologs that define each of the presumptive Proto-Cyclostome/Petromyzontid chromosomes presented in figure 4g. It seems that some of these are very small, but it is hard to assess with the presented data.
- 6) At face value the reconstruction method seems to assume 2 rounds of duplication, this appears to impart several important features to the inferred evolutionary history of vertebrates that are worthy of discussion (outlined in more detail below). However it is not clear from the textual description of the

algorithms if some of these are artefacts of analysis since it is not completely clear how under what conditions WGD is presumed to have occurred, or how duplications are differentiated from ancient fissions/translocations under their model (both would be expected to result in the collapse of segments in the ancestor and the presence of duplicates (retained following duplication and rediploidization of neighboring genes, or separating onto derived segments after originating in cis).

7) Fuller articulation of alternate models and rigorous tests of alternatives will also be important for assessing and discussing 2R. Similar to comment 3 above.

8) As mentioned above, several features of the reconstruction are worthy of discussion with respect to their probabilistic and biological meaning. The first of these is the overarching predominance of chromosomal fusion (vs fission) between the 1R and 2R duplications. This reconstruction requires 11 fusion events and zero fissions. This seems noteworthy in light of the fact that there are more even numbers reported between 2R and the basal gnathostome split 3 fissions and 4 fusions. This may attach to comment 6 above, or may reveal an unusual aspect of vertebrate biology that arose briefly following the split of gnathostome and agnathan lineages but before 2R. The timing, mechanics and probability of this seem worthy of extensive discussion.

9) With respect to phylogenetic reconstructions, the authors raise an important point. "Intriguingly, we observed large numbers of vertebrate paralogs between most pairs of homoeologous proto-gnathostome and proto-cyclostome chromosomes, seemingly suggesting a contradictory model where quadruple proto-gnathostome chromosomes and sextuple protocyclostome chromosomes were created before the gnathostomes-cyclostomes split." It is fairly well understood that this pattern pervades these trees and was previously understood to be due to long branch attraction and similar artefactual convergence related to long term substitution biases in lampreys. The authors also mention the possibility that this is explained by allopolyploidization, but do not mention these more mundane explanations, or other alternatives such as true differences in timing of duplication events and hidden paralogy. This part of the discussion is also a bit confusing because earlier in the manuscript 2R is discussed in the context of an allopolyploidization event, whereas this seems to be focusing on peri-1R patterns (or pre-1R?).

10) Examination of the phylogenies of some 6-fold duplicated in lamprey may shed additional light on the timing of presumptive duplications. As was performed previously for sea lamprey hox clusters. It would be nice to see this done for a larger number trees that were generated as part of their analysis pipeline. This would also give readers a better sense of the underlying data.

11) The paragraph starting at the bottom of page 11 related to the asymmetric and unequal contribution from the subgenomes could use further development. Which chromosomes are thought to belong to the A and B subgenomes in Figure 4? Do the authors propose that these have evolved in a manner similar to *Xenopus* wherein one of the subgenomes has lost more paralogs than the other? Please discuss further the degree of asymmetry observed here, and compare to that of *Xenopus* and other systems where it has been observed.

12) The paragraph related to AIS and microchromosomes could also use a bit of development as it is a bit difficult to understand. Is the "immune supercomplex" idea central to the "big bang" theory? It seems that this idea should have fallen by the wayside some time ago, but perhaps this should be developed further? Additionally, the section appears to argue that more immune genes were inherited from the subdominant (b) genome. Is this correct? Some of it would be nice to see this cleared up. Additionally, this clause seems like it might be missing a reference "corroborates the view that a primordial 'adaptive' immune system emerged in the ancestral vertebrate genome and later turned into the intricate gnathostome-like AIS through 2R."

13) The Methods, or large portions thereof, should be elevated to the main body of the manuscript and presented in a manner that is accessible to a broad audience, including assumptions and caveats

that relate to inferring duplications and pre-duplication states.

14) The authors should elevate reporting of assembly improvement to the Results section and develop a figure that more effectively relays improvements. Comparing the cumulative rate of increase in assembly size across increasing scaffold lengths (often included in standard DoveTail reports) would provide important perspective.

15) Code and sequence availability: The authors state that "reconstruction software/code is available on request." However, I would strongly recommend that the code be released on GitHub (or similar) as soon as possible and that reconstructions be included as supplemental data files or placed in another permanent repository. Access to the code and reconstructions are necessary in order to properly assess their findings, and would have likely changed some of the comments made above. An embargoed release of the genomes would also be useful, and has become common practice, although I understand that this is not necessarily standard practice at this point in history.

Sincerely
Jeremiah Smith

Reviewer #3:

Remarks to the Author:

It was a delight to read the manuscript "Reconstruction of proto-vertebrate, proto-cyclostome and proto-gnathostome genomes provides new insights into early vertebrate evolution" by Nakatani and others. The study reconstructs the genome of the first vertebrates at chromosome/level, by using high-quality genomes of a lamprey and the elephant shark. The results offer a highly detailed and resolved picture of the genome of early vertebrates, gnathostomes, and cyclostomes, shedding new light on the debate about the whole genome duplications. I found the results on microchromosomes very original and interesting. The design of the analyses and the manuscript writing are great, and the conclusions highly relevant to our knowledge of vertebrate origins. I would like to commend the authors for their efforts.

My only criticism is about the discussion about the evolution of the adaptive immune system and MHC. While I think this is very interesting and the data/analyses certainly support the claims, this is only touched in the Discussion section and seems a bit out of the blue. I'd like to suggest to support this either with another section in Results or maybe a figure.

Along those lines, another suggestion to make the paper interesting to a wider audience would be to add a figure in which the different hypotheses about 1R, 2R, and cyclostome-specific WGS are mapped to a phylogeny. This would help some readers to understand better the evolutionary scenario, as well as show the phylogenetic relationships of all the animals involved, which are never shown. If the authors decide to follow this advice, I'd also add photos of the sequenced organisms here. If the paper has reached the limit of displayed items, I think Figure 3 could be easily moved to Supp data, as it is not that informative and there are enough figures with dots in the paper already (this is a very "dotty" paper!).

I do not have any major criticisms, but I have some other comments and questions that I hope the authors can kindly address:

- 1) Page 3, I wonder if the authors could add a reference to the number of vertebrate species. This number keeps creeping up as time goes!
- 2) Page 3, I'd like to suggest replacing "degenerate" by "simplified", as the first has other connotations.
- 3) Page 6 and others, I wonder if the selection of genomes to perform comparative synteny analyses

was just based on evolutionary rates or also on high contiguity genomes.

4) Page 10, first sentence, maybe I need more coffee but I did not understand the bit between parentheses "(or diverged before 1R)". I would like the authors to clarify this in the text.

5) Page 10, the sentence "the ancestral metazoan animal genome", the paper is comparing a mollusc vs a vertebrate. It should say "Bilaterian" rather than "metazoan"

Reviewers' comments:

Response: We would like to thank all three reviewers for their time and effort in reviewing our manuscript and
offering detailed and constructive suggestions. Their comments have indeed helped to improve the clarity of the
manuscript.

Reviewer #1 (Remarks to the Author):

I am happy to recommend that this article be published by Nature Communications as soon as my comments and
recommendations, as detailed below and in the attached document, are addressed. I don't think that the study
requires additional analyses and I am confident that the methods are appropriate and sufficient, based on my review
as well as previous highly-regarded publications by Nakatani and co-authors, as well as Venkatesh and co-authors.

The application of the modified probabilistic macrosynteny model to the question of cyclostome-gnathostome
divergence relative to the vertebrate tetraploidizations is especially exciting, as this method has proven its
usefulness in previous publications by Nakatani and co-authors. In addition, the sequencing and chromosome-level
assembly of a lamprey genome and an elephant shark genome will undoubtedly be useful resources for molecular
evolutionary studies in vertebrates and vertebrate genomics. These species hold key taxonomic positions that have
previously been under-addressed due to the lack of high-quality genomic resources. Congratulations on a fantastic
paper!

However, some methods and procedures are not described clearly, which made some aspects of the analyses
difficult to review. I also have some concerns about the conclusion that the cyclostome lineage underwent a
hexaploidization event. This is a very novel suggestion, and I want to make sure that the authors have done
everything possible to explain their method clearly, so that no serious doubts can be brought forward about the
conclusion.

My general comments and suggestions are included below, and more detailed comments have been attached in a
separate document.

General comments:

**[Comment 01]**- Will the new genome assemblies be shared as part of any of the commonly used public genome
browsers? Is the *Lethenteron camtschaticum* genome assembly, LetJap1.0, the same that has already been shared
through NCBI: https://www.ncbi.nlm.nih.gov/assembly/GCA_000466285.1? The submitter of LetJap1.0 matches
the home institution of several of the co-authors. If so, the authors should mention in the paper that the genome
assembly has been shared, and direct the reader towards the online databases. If a newer assembly has been made,
this should be shared in the same way. The BioProject entries for the new genomes mentioned in the paper are not
active yet, so I couldn't check them; but presumably these will only include the raw data, not the assembled
genomes. Sharing the genome assemblies in an easily browsable/searchable way is crucial.

The Japanese lamprey genome assembly used in the current study is a de novo, PacBio read-based assembly and is
different from the LetJap1.0 assembly version that was also generated by our group. The Japanese lamprey and
elephant shark genome assemblies generated in this study have been submitted to GenBank and will be available in
the public domain before the publication of our manuscript. In the revised version of the manuscript, we have
provided their GenBank accession numbers in the Data Availability section.

**[Comment 02]**- The figure legends are inordinately long. Please make sure to only include information relevant for
the graphical interpretation of the figure. As they are now, the figure legends include lengthy descriptions of the
methodology and descriptions of results. This should not be included in a figure legend. Otherwise, it may look like
the authors are not confident that their text is good enough for the reader to understand the figures. Or, perhaps
more cynically, that the authors ran out of words in the main text of the paper and are smuggling some of the text
into the paper via the figure legends. They can do better. I have suggested some changes in my detailed comments
(attached).

We shortened and simplified the figure legends, following Reviewer 1's suggestions.

**[Comment 03]**- Another smaller issue is the nomenclature of the Japanese or Arctic lamprey, *Lethenteron*
*japonicum* alt. *Lethenteron camtschaticum*. According to the World Register of Marine Species, *L. japonicum*
(Martens, 1868) is an unaccepted synonym. Source:
<http://www.marinespecies.org/aphia.php?p=taxdetails&id=298380>. The accepted name is *L. camtschaticum*
(Tilesius, 1811). Source: <http://www.marinespecies.org/aphia.php?p=taxdetails&id=101173>. This is also the case in
the FishBase database (<https://www.fishbase.se/summary/Lethenteron-camtschaticum.html>) and in the NCBI
taxonomy browser (<https://www.ncbi.nlm.nih.gov/Taxonomy/Browser/wwwtax.cgi?id=980415>). Please change all
references to the binomial name of this species throughout the manuscript to reflect the accepted nomenclature. At
the first reference to the species, on page 4, the common synonym *L. japonicum* should be mentioned. But it's not
the accepted name. It's all right if the authors use "Japanese lamprey" as the common
name throughout the paper, as long as *L. camtschaticum* is used and the other common name "Arctic lamprey" is
mentioned at the first mention of the species.

We thank Reviewer 1 for raising this point. We revised the text and mentioned the accepted nomenclature for
Japanese lamprey.

**[Comment 04]**- The authors consistently mention 18 photo-vertebrate chromosomes throughout the paper. There
are several issues with this. 1) These are reconstructed chromosomes, so they are a purely theoretical conjecture
about the karyotype of the photo-vertebrate. I understand that it would be cumbersome to clarify this each and
every time they are mentioned, and simply writing "chromosome" is a good shorthand, but the authors should be
absolutely clear, at crucial points of the text, that these are theoretical constructions.

1) In the revised manuscript, we made this point clear by writing putative/hypothetical/reconstructed chromosomes
when necessary.

2) The reconstruction of the “PvcUn” chromosome is a bit more problematic. I seems to consist of a relatively large
number of small fragments that could not be assigned to any other of the proto-vertebrate chromosome
reconstructions. It is likely that these fragments correspond to other proto-chromosomes and that there is no 18th
proto-vertebrate chromosome. Indeed, in comparing their results to those of Sacerdot et al. (2018), PvcUn seems to
match Pvc17 (Table S8). The authors should make this clear in the main text, not only the supplementary text.

2) As written in the initially submitted manuscript, our reconstruction algorithm inferred 18 proto-vertebrate
chromosomes, and we called one of them as PvcUn because it showed unclear macrosynteny conservation in the
outgroup amphioxus genome. We dealt it as the 18th chromosome because it shows one-to-one macrosynteny
correspondence to a small scallop chromosome in Supplementary Figure S4, and also to a chromosome
reconstructed by Sacerdot et al.in Supplementary Figure S3. These observations suggest the possibility that the
macrosynteny conservation was lost in the amphioxus genome due to rearrangements in the amphioxus lineage.
However, we note that the nomenclature could be confusing or misleading, and may obscure this point, so we have
relabelled this as Pvc18 and adjusted the text accordingly to make this clearer.

3) Thus, I suggest that the authors refer to 17 proto-vertebrate chromosomes, not 18, and when necessary refer to
“PvcUn” as separate from the set. For example, on page 6, lines 4-5 - I suggest “Our reconstruction of the proto-
vertebrate genome comprises 17 ancestral chromosomes, designated as Pvc1-17, as well as PvcUn, which consists
of weak macrosynteny segments that could not be assigned to Pvc1-17.” Please make sure that this is carried
through for the whole text. Regarding the analysis matching PvcUn to scallop chromosome 13 as an argument for
PvcUn representing a “true” ancestral chromosome, see my comment for page 23, line 18, in the attached
document.

3) Our reconstruction algorithm inferred 18 chromosomes and calling one of them as PvcUn was our
interpretation/speculation. In the revised manuscript we relabelled it as Pvc18. We have addressed the specific
comments in the reviewer’s separate document and also include that file in our response.

**[Comment 05]**- I have some concerns about the description of the analyses of the proto-cyclostome genome
reconstruction, and how the authors arrived at a hexaploidization scenario. My main issue is that these analyses
have not been described well enough for me to make a judgment of whether the conclusions seem correct or not.
For example, Jeremiah Smith and co-authors have suggested the involvement of a series of segmental duplications
in the cyclostome lineage. How did the authors distinguish between genome hexaploidization and genome
tetraploidization + segmental duplications? Simply calculating the “multiplicity” of genes would not address this. I
have detailed some other concerns in the detailed comments (attached document) for pages 6-8 as well as for the
supplements.

We would like to emphasize that we calculated multiplicity of chromosomes not genes: first, we counted the
number of reconstructed proto-cyclostome chromosomes that are duplicated from proto-vertebrate chromosome

Pvc1, and we found that the “multiplicity” was six; second we repeated this procedure for Pvc2, Pvc3, and so on;
third, we found that nine out of 18 proto-vertebrate chromosomes were duplicated into six proto-cyclostome
chromosomes, which cover the majority of the cyclostome genomes; fourth, we concluded that the sharp peak at
multiplicity six in Figure 3 suggests six-fold duplication of the entire genome (i.e. paleo-hexaploidization), rather
than tetraploidization plus segmental duplications. In the revised manuscript, we added detailed arguments as
follows.

“If the proto-cyclostome genome was shaped by three rounds of tetraploidization (S5 in Fig. 1), it should be
covered by chromosomes of multiplicity eight. Instead if it experienced a single tetraploidization with subsequent
chromosomal duplications (S8 in Fig. 1), the multiplicity should peak at two with gradual decrease toward larger
multiplicities. The third possibility is that if the genome went through a single tetraploidization and a
hexaploidization (genome triplication) (S6 in Fig. 1) the majority of the genome should be covered by
chromosomes of multiplicity six.”

“Although the current lamprey genomes might still be incomplete and some chromosomes might be fragmented,
such limitations are unlikely to have substantially biased our analysis. First, if the proto-cyclostome genome was
shaped by three rounds of tetraploidization, that would additionally require a large number of subsequent
chromosome fusions to explain the current genome arrangement (for example, 45 post-tetraploidization fusions are
required to obtain the chromosome number of sea lamprey germline cells: $18 \times 8 - 45 = 99$). However, we found that
the lamprey lineage had remarkably low rates of inter-chromosomal rearrangement (Supplementary Fig. S5) over
~ 500 million years⁴² of cyclostome evolution. Specifically, our proto-cyclostome genome reconstruction shows
large-scale fusions and translocations affecting only 22 out of 141 Japanese lamprey scaffolds and only 19 out of
151 sea lamprey scaffolds that have at least 10 genes. The exceptionally low rate of inter-chromosomal
rearrangement and the haploid chromosome number of ~ 99 in the germline sea lamprey genome⁴³ are consistent
with our evolutionary scenario in which the lamprey chromosome number is explained approximately as 18×6
$= 108$ with several subsequent fusions. Second, even though some tiny chromosomes might be missing in the
current proto-cyclostome reconstruction, large chromosomes (e.g. Hox-bearing chromosomes duplicated from
Pvc1) are unlikely to be missing entirely; therefore, our reconstruction is particularly reliable for the largest five
proto-vertebrate chromosomes (i.e. Pvc1, 3, 10, 13 and 17), which consistently exhibited a multiplicity of six. Thus,
the high coverage (60.3%) of the Japanese lamprey genome by six-fold duplicated proto-cyclostome chromosomes
suggests that extant cyclostome genomes are paleo-dodecaploids (i.e. the chromosome number increased as 18×6
due to tetraploidization and hexaploidization), which might be similar to the situation in sturgeon where a species
(*Acipenser brevirostrum*) with ~ 180 chromosomes is considered to be a hexaploid of a tetraploid ancestor with
~ 60 chromosomes⁴⁴⁻⁴⁶.”

**[Comment 06]**- In general, I miss a discussion of alternative scenarios in the paper. The authors mention
alternative scenarios proposed by other previous papers like Mehta et al. (2013), Smith & Keinath (2015), Smith et
al. (2018) and Sacerdot et al (2018), but I miss a discussion regarding whether any of these alternative scenarios
could be possible with another interpretation of the results presented in the paper. In other words, can the authors

definitely disprove any of the previous alternative scenarios?

See our response to a comment from Reviewer 2 [Comment 12].

It would be helpful to the reader if the authors could discuss at least the one most likely alternative scenario. Why
isn't a shared 1R/2R at the base of vertebrates followed by independent fissions/segmentations a likely scenario, for
example? Something like this has been proposed by Jeramiah Smith and co-authors, based on the meiotic map of
the previous sea lamprey genome assembly, and more recently based on synteny conservation of the latest sea
lamprey germline genome. I concede that Smith and co-authors have gone back-and-forth and suggested partly
contradictory scenarios, but it seems to boil down to one shared WGD together with chromosome-level segment
duplications and fissions, possibly both preceding and following the WGD. Based on the current results presented
in the present paper, why are these alternative scenarios less likely?

We had discussed this issue in the initially submitted manuscript, however, perhaps we were not sufficiently clear.
First, we resolved the divergence timing issue by identifying lineage-specific rearrangements. We found several
chromosome fusions occurring between 1R and 2R as shown in Fig. 6. Those fusions were observed in the proto-
gnathostome genome but not in the proto-cyclostome genome. We interpreted this observation as the evidence that
the cyclostome lineage diverged from the gnathostome lineage between 1R and 2R. Second, we favoured our
tetraploidization-plus-hexaploidization model rather than the 1R-plus-segmental-duplication model, because our
reconstruction showed a clear peak at multiplicity six (Fig. 3). This observation cannot be explained by segmental
duplications, unless we come up with a molecular mechanism through which the numbers of independently
duplicating proto-vertebrate chromosomes eventually converge to six in the proto-cyclostome genome. We also
note that the analyses by Smith et al. [Smith and Keinath, *Genome Res* (2015); Smith et al., *Nat Genet* (2018)] were
not reconstruction-based, and thus they could be affected by lineage-specific rearrangements and incompleteness of
the genome assemblies.

**[Comment 07]**- Smith et al. (2018) also have the great advantage of dealing with the germline genome of the sea
lamprey. As is well-known, lampreys greatly modify their genomes in the mature somatic cells, losing upwards of
20% of the genomic DNA. The authors describe that the DNA for the Japanese lamprey genome assembly was
extracted from the mature testis (page 4 of supplementary information), while Smith et al. (2018) specify that
germline DNA was extracted from sperm cells of sea lamprey. I'm not entirely familiar with the methods for SMRT
sequencing, but how confident are you that your Japanese lamprey genome assembly reflects the germline genome?

The testis was collected from an adult male during the peak breeding season. As shown in the photo below, the tissue
was full of sperm and the milt oozed profusely when a small incision was made. Thus, the tissue we used was
predominantly sperm and the genome assembly represents the germline genome. We have checked our lamprey
genome assembly for some genes (e.g. *WNT5A*, *HFMI* and *COBLL1*) reported to be lost in the somatic genome of
the sea lamprey (Bryant et al. *Mol. Biol. Evol.* 2016) and they are indeed present in the assembly.

**[Comment 08]**- I also have concerns regarding the annotation of orthologs vs. paralog. The method is ingenious,
although it has some limitations, and the principles behind it make sense. However, there are many pitfalls related
to the fact that it is easy to misidentify orthology and paralogy with automatic annotations and gene trees, and with
reciprocal BLASTP searches. I would want to make sure that these pitfalls have been avoided to the utmost extent.
I would like the authors to describe the methods, the procedures, and the datasets in clearer detail in the
supplementary information. As it is right now it would be nigh impossible for anyone to reproduce these analyses.
See my comment in the attached document regarding page 18 of the supplementary information.

It seems to be a prevalent misunderstanding that the utmost accuracy is required in a specific step of the
reconstruction method. In reality, what is important in our analysis is to design a robust computational method so
that minor errors (including orthology/paralogy annotation errors) do not affect our conclusions. For this purpose,
we previously developed a probabilistic macrosynteny model, and published the method as a separate paper. The
essential idea common to “macrosynteny” analyses is that the “signals” (i.e. traces of the ancestral genome
structure) remain in the modern genomes even if there is certain amount of “noise” (i.e. small-scale translocations,
small-scale segmental duplications, gene annotation errors, gene tree errors, orthology/paralogy annotation errors,
genome sequencing errors and genome assembly errors, etc). Please see [Nakatani and McLysaght, *Bioinformatics*
(2017)] for a more detailed description of our method. Please also see the figures in Supplementary Information,
because orthology/paralogy annotation errors should be visible as randomly distributed dots. In addition, the
revised manuscript includes the ortholog/paralog dataset used for our reconstructions as Supplementary Data 1.

**[Comment 09]**- The authors consistently write about implications for human disease, however, I cannot identify
anything in the study that would further our understanding of the molecular/genetic mechanisms of disease, disease
progression, treatment, etc, which is what is clearly implied by centering on human disease. Genetic diseases may
reveal some constraints on genome evolution, which the authors discuss in a relevant way. But from this, there is a
big step to talking about “implications for human disease”. This reference to human disease must be tempered and

put into the right context in the revised manuscript. Otherwise, this just looks like a transparent attempt to drive up
the significance of the study by linking it to human disease. Surely the readers of Nature Communications can see
through this, and I certainly don't think it was the author's intention.

The ancestral genome reconstruction enables us to recognise relationships between regions of modern genomes by
virtue of their shared descent from a specific macrosynteny block. This has implications for understanding genome
evolution in general, but also identification of hard-to-detect ohnologs. Because ohnologs are so frequently
associated with disease, this has implications for identification of disease genes. We had included the reference to
the link between ohnologs and human disease because we genuinely think it is of great interest, but it is also true
that the value of this paper does not depend on that, so we have removed it from the abstract and introduction, and
now it is just mentioned in passing in the discussion.

Finally, my spell checker kept changing "proto" to "photo", "port" or "protocol". I think I have identified the
majority of these mistakes, but if there is a "photo-vertebrate" chromosome here and there in my responses, please
overlook it.

It was a lot of work going through this manuscript in the detail that it deserves, but it was a pleasure to take part in
these results before they are released. I apologize if my ignorance of some specific topics made me ask for a lot of
clarification, but think of readers like myself who will benefit from this study without necessarily being experts in
the intricacies of ancient genome reconstruction and macrosynteny algorithms.

I wish my colleagues all the best in the publication of this paper and I'm excited for it to come out.

Reviewer #2 (Remarks to the Author):

The manuscript “Reconstruction of proto-vertebrate, proto-cyclostome and proto-gnathostome genomes provides
new insights into early vertebrate evolution”, by Nakatani et al reports improved genome assemblies for two
species (elephant shark and Japanese/Arctic lamprey) and uses these genomes to reconstruct whole genome
duplication events, using reconstruction algorithms that have not previously been applied to the problem. These are
presented as lending strong support to specific whole genome duplication scenarios. However much of the
information necessary to assess the reconstructions is unavailable to the reader, and the analysis of reconstructions
does not effectively test their favored hypotheses against previously-proposed hypotheses or others that seemingly
emerge from their analyses. Moreover, a more thorough discussion of the biological underpinnings of their
proposed evolutionary mechanisms would be welcome, and necessary for readers to understand the implications of
the presented analyses. There seem to be relatively straight forward remedies to these issues, which are outlined in
the comments below.

Comments:

**[Comment 10]** 1) First, use of the term “Proto-Cyclostome” is seemingly inappropriate with respect to the
reconstructions that are presented in this paper. The lineages leading to sea lamprey and Japanese lamprey diverged
approximately 20-30 million years ago. Therefore the hypothetical reconstructed ancestor would more
appropriately be called the Proto-Petromyzontid ancestor. This refers to a branch that extends to ~250 MYA at
which point the petromyzontid lineage is thought to have split from Geotria lampreys. Without data from other
lampreys or hagfish, it seems like over-reaching to call the reconstruction “Proto-Cyclostome”.

It is correct that we used the two lamprey genomes in our reconstruction, whose last common ancestor is closer to
the proto-petromyzontid ancestor than to the proto-cyclostome ancestor. On the other hand, what we reconstructed
is the post-polyploidization (i.e. post-hexaploidization in our model) genome rather than the last common ancestor
of two lamprey lineages. An analysis of the hagfish Hox gene clusters [Pascual-Anaya et al., *Nat Ecol Evol* (2018)]
suggested that the polyploidization event is shared between the hagfish and lamprey lineages. For this reason, we
favour retaining the term ‘proto-cyclostome’ for describing this reconstruction in the revised manuscript.

**[Comment 11]** 2) The authors state that “Whether microchromosomes were recently created by chromosome
fission, or were present in the gnathostome ancestor has been controversial”. In my impression this does not
accurately reflect the recent state of literature. Multiple analyses of various genomes, including most notably
amphibians, gar and lamprey in comparison to birds and elephant shark have seemingly firmly established this.
We agree with Reviewer 2 and revised the main text as follows. “Although several recent studies supported the
ancient origin of microchromosomes, it was still unknown (1) if chromosomal features characteristic to modern
avian microchromosomes (i.e. high GC-content, high gene density and high recombination rate) already presented
in the ancestral gnathostome genome, and (2) why microchromosomes have been conserved in distantly related
gnathostome species such as the chicken, spotted gar and elephant shark.”

[Comment 12] 3) In general the authors should strive to more fully articulate alternate models and specifically test
the fit of those models to observed patterns across extant genomes, not simply the reconstruction that is optimal
under their algorithm.

First of all, we need to be aware that it is not possible to reject alternative scenarios by rigorous statistical tests,
because nobody knows realistic parameters of rearrangements occurring in early vertebrate genomes. In particular,
little is known about the probability that chromosome duplications (or chromosome-scale segmental
duplications/deletions) are inherited for long generations and fixed in the population, although we know such large-
scale duplications should be extremely rare and unlikely due to the disruption of gene dosage balance. For this
reason, we could not perform statistical tests for rejecting alternative scenarios in the manuscript. Please also see
our response to [Comment 05] above for additional explanations of alternative models.

One example of this is the assertion that the numbers of Proto-Petromyzontid chromosomes/segments supports a
post-1R triplication. The distribution of paralogous segment counts peaks at 6, which is considered evidence of
duplication followed by triplication. However, it should be noted that a simple model of random segmental
duplication would also be expected to yield a peak with mean = 6. Constraining this pattern assuming 1R
substantially sharpens this peak.

We concluded that the proto-cyclostome genome was shaped by six-fold duplication of the entire proto-vertebrate
genome, because (1) the five largest proto-vertebrate chromosomes gave rise to six proto-cyclostome chromosomes
and (2) the majority of the Japanese lamprey genome was mapped to these six-fold duplicated chromosomes. Other
scenarios including the 1R-plus-segmental-duplications model are interesting, but we were unable to come up with
a convincing biological mechanism through which the numbers of independently duplicating proto-vertebrate
chromosomes converge to six. Please also see our response to [Comment 05] above.

Based on a quick permutation test, 1R plus random duplication seems to be a better fit to the observed distribution
than 1R + triplication. It is probably also worth considering 1R + duplication and other models. Admittedly, a more
formal statistical approach related to the birthday problem of hash collision might provide a more elegant solution
that permutation.

To the best of our knowledge, our analysis is the most comprehensive investigation of alternative scenarios
(including the 1R-plus-segmental-duplication scenario). In our reconstruction, we had no prior assumption of the
number of WGD events and segmental duplications: we enumerated possible combinations of lamprey segments,
and chose the combination with the most significant (i.e. non-random) distribution of paralogs and orthologs. Thus,
we do not change our conclusion that the proto-cyclostome genome was shaped by six-fold duplication of the entire
genome, unless someone proposes a convincing biological mechanism through which the numbers of
independently duplicating proto-vertebrate chromosomes converge to six. In order to show that our reconstruction
method explores alternative scenarios comprehensively, we added Supplementary Movie 1, which visualizes the
exploration of alternative scenarios during the reconstruction procedure.

[Comment 13] 4) A second comment related to this is that the numbers presented in figure 2d should refer to the

numbers of ancestral genes that are incorporated into these classes, not the number of lamprey genes (as these
 include duplicates).

As Reviewer 2 commented, the numbers of Japanese lamprey genes in Fig. 3d (Fig. 2d in the initially submitted
 manuscript) includes duplicates. If we count a family of paralogs only once, we obtain the following table.

Multiplicity	Chromosomes	Genes	Ratio	
–	PvcUn	493	0.036	■
Pvc9,14	1088	0.080	■
Pvc2,12	1769	0.130	■
Pvc1,3,4,7,8,10,13,16,17	8188	0.603	■
Pvc5,6	817	0.060	■
Pvc11	693	0.051	■
Pvc15	533	0.039	■
total		13581		

 The ratio values are almost the same as the original table, and thus the difference does not affect our arguments. In
 the revised manuscript, we kept the original table, because (1) the inference of the numbers of (de-duplicated)
 ancestral genes would impose additional uncertainty because of the possibility of small-scale duplications occurring
 before and during the polyploidization events; and (2) what we discuss with regard to this table is that more than
 60% of the Japanese lamprey genes were mapped to the six-fold duplicated proto-cyclostome chromosomes.

 **[Comment 14]** 5) Related to this, it would be very useful if the authors could provide the number of orthologs that
 define each of the presumptive Proto-Cyclostome/Petromyzontid chromosomes presented in figure 4g. It seems that
 some of these are very small, but it is hard to assess with the presented data.

The table below shows the statistics. Each line shows (1) proto-vertebrate chromosome name (PVC), (2) number of
 amphioxus genes mapped to the PVC, (3) proto-cyclostome chromosome name (PCC), (4) number of Japanese
 lamprey genes mapped to the PCC, (5) number of sea lamprey genes mapped to the PCC, and (6) number of
 amphioxus genes that are mapped to the PVC and are orthologous to lamprey genes mapped to the PCC.

Proto-vertebrate	Amphioxus	Proto-cyclostome	Japanese lamprey	Sea lamprey	Orthologous amphioxus genes on the PVC
Pvc1	1445	Pcc1A	515	473	304
Pvc1	1445	Pcc1B	502	452	283
Pvc1	1445	Pcc1C	397	344	250
Pvc1	1445	Pcc1D	303	266	177
Pvc1	1445	Pcc1E	226	217	154
Pvc1	1445	Pcc1F	158	179	121
Pvc2	891	Pcc2A	287	326	174
Pvc2	891	Pcc2B	252	228	167
Pvc2	891	Pcc2C	206	225	154

Pvc2	891	Pcc2D	172	177	115
Pvc2	891	Pcc2E	155	184	78
Pvc3	686	Pcc3A	264	265	99
Pvc3	686	Pcc3B	261	234	96
Pvc3	686	Pcc3C	237	220	89
Pvc3	686	Pcc3D	231	212	88
Pvc3	686	Pcc3E	120	131	52
Pvc3	686	Pcc3F	62	114	47
Pvc4	473	Pcc4A	171	169	89
Pvc4	473	Pcc4B	143	171	67
Pvc4	473	Pcc4C	127	177	66
Pvc4	473	Pcc4D	18	0	1
Pvc4	473	Pcc4E	0	16	1
Pvc4	473	Pcc4F	4	4	0
Pvc5	525	Pcc5A	190	201	112
Pvc5	525	Pcc5B	189	175	96
Pvc5	525	Pcc5C	50	44	20
Pvc5	525	Pcc5D	0	38	11
Pvc5	525	Pcc5E	10	22	6
Pvc5	525	Pcc5F	7	8	2
Pvc5	525	Pcc5G	9	0	1
Pvc6	385	Pcc6A	182	212	105
Pvc6	385	Pcc6B	188	171	86
Pvc6	385	Pcc6C	108	101	55
Pvc6	385	Pcc6D	46	56	33
Pvc6	385	Pcc6E	36	26	17
Pvc6	385	Pcc6F	0	61	9
Pvc6	385	Pcc6G	10	0	0
Pvc7	707	Pcc7A	271	266	173
Pvc7	707	Pcc7B	260	271	155
Pvc7	707	Pcc7C	124	112	66
Pvc7	707	Pcc7D	162	25	43
Pvc7	707	Pcc7E	11	0	2
Pvc7	707	Pcc7F	0	10	2
Pvc8	420	Pcc8A	276	252	130
Pvc8	420	Pcc8B	207	212	100

Pvc8	420	Pcc8C	40	32	10
Pvc8	420	Pcc8D	15	15	4
Pvc8	420	Pcc8E	0	11	3
Pvc8	420	Pcc8F	2	4	0
Pvc9	563	Pcc9A	355	344	174
Pvc9	563	Pcc9B	277	282	159
Pvc9	563	Pcc9C	145	145	30
Pvc9	563	Pcc9D	23	0	3
Pvc10	962	Pcc10A	257	240	151
Pvc10	962	Pcc10B	252	240	148
Pvc10	962	Pcc10C	218	228	129
Pvc10	962	Pcc10D	196	205	120
Pvc10	962	Pcc10E	172	202	115
Pvc10	962	Pcc10F	128	170	90
Pvc11	844	Pcc11A	314	296	167
Pvc11	844	Pcc11B	225	261	126
Pvc11	844	Pcc11C	107	132	74
Pvc11	844	Pcc11D	90	80	44
Pvc11	844	Pcc11E	58	106	39
Pvc11	844	Pcc11F	12	15	8
Pvc11	844	Pcc11G	0	28	7
Pvc11	844	Pcc11H	16	0	1
Pvc12	798	Pcc12A	366	361	181
Pvc12	798	Pcc12B	258	259	157
Pvc12	798	Pcc12C	225	246	151
Pvc12	798	Pcc12D	157	313	113
Pvc12	798	Pcc12E	0	14	5
Pvc13	1196	Pcc13A	470	441	234
Pvc13	1196	Pcc13B	346	342	203
Pvc13	1196	Pcc13C	251	232	151
Pvc13	1196	Pcc13D	188	217	115
Pvc13	1196	Pcc13E	141	173	101
Pvc13	1196	Pcc13F	24	0	9
Pvc14	602	Pcc14A	242	224	130
Pvc14	602	Pcc14B	175	187	104
Pvc14	602	Pcc14C	85	159	53

Pvc14	602	Pcc14D	0	25	6
Pvc15	560	Pcc15A	251	239	126
Pvc15	560	Pcc15B	164	194	99
Pvc15	560	Pcc15C	91	66	47
Pvc15	560	Pcc15D	53	96	38
Pvc15	560	Pcc15E	33	12	20
Pvc15	560	Pcc15F	16	23	14
Pvc15	560	Pcc15G	0	31	12
Pvc15	560	Pcc15H	11	0	4
Pvc15	560	Pcc15I	4	6	2
Pvc16	689	Pcc16A	283	267	180
Pvc16	689	Pcc16B	263	254	144
Pvc16	689	Pcc16C	42	9	11
Pvc16	689	Pcc16D	9	12	3
Pvc16	689	Pcc16E	0	10	3
Pvc16	689	Pcc16F	3	5	2
Pvc17	1282	Pcc17A	491	420	291
Pvc17	1282	Pcc17B	326	313	203
Pvc17	1282	Pcc17C	302	301	184
Pvc17	1282	Pcc17D	298	269	182
Pvc17	1282	Pcc17E	295	265	173
Pvc17	1282	Pcc17F	140	168	108
Pvc18	197	Pcc18A	859	569	59

**[Comment 15]** 6) At face value the reconstruction method seems to assume 2 rounds of duplication, this appears to
impart several important features to the inferred evolutionary history of vertebrates that are worthy of discussion
(outlined in more detail below). However it is not clear from the textual description of the algorithms if some of
these are artefacts of analysis since it is not completely clear how under what conditions WGD is presumed to have
occurred, or how duplications are differentiated from ancient fissions/translocations under their model (both would
be expected to result in the collapse of segments in the ancestor and the presence of duplicates (retained following
duplication and rediploidization of neighboring genes, or separating onto derived segments after originating in cis).
The algorithm compares many possible reconstructions (which were called set partitions in Supplementary
Information). In particular, the algorithm considers reconstruction into two, three, four, five, six, seven, eight, ...
duplicated chromosomes. Polyploidization is inferred if the majority of the proto-vertebrate chromosomes have the
same multiplicity in the proto-cyclostome genome (see Fig. 3) or in the proto-gnathostome genome (see Figs. 4, S6

and S7). Pre-1R fissions are expected to result in two distinct ortholog distributions (see Fig. 1 in [Nakatani and
McLysaght, *Bioinformatics* (2017)]). Fusions and fissions between 1R and 2R can be distinguished by a
comparison with outgroup genomes (see [Nakatani et al., *Genome Res* (2007)]). Post-2R rearrangements and
fragmental genome assemblies result in smaller segments, which can become small fifth and sixth chromosomes in
the proto-gnathostome reconstruction or seventh and eighth chromosome in the proto-cyclostome reconstruction.
Smaller-scale rearrangements (translocations and segmental duplications) affecting only a small number of genes
are not detectable in our macrosynteny analysis, but they are expected to be visible in paralog/ortholog plots as
isolated clusters of dots (Supplementary Figs. S3-S7, S9-S13).

**[Comment 16] 7) Fuller articulation of alternate models and rigorous tests of alternatives will also be important for**
**assessing and discussing 2R. Similar to comment 3 above.**

To the best of our knowledge, our study is the most comprehensive analysis of alternative models. Specifically, our
algorithm explores all possible reconstructions and examines the paralog distributions. During this process, the
algorithm does not exclude alternative scenarios including segmental duplications, chromosome
duplications/losses, tetraploidization, hexaploidization, and so on. In addition, rigorous tests of alternatives are not
possible at present, as explained in our reply to [Comment 12] above.

**[Comment 17] 8) As mentioned above, several features of the reconstruction are worthy of discussion with respect**
**to their probabilistic and biological meaning. The first of these is the overarching predominance of chromosomal**
**fusion (vs fission) between the 1R and 2R duplications. This reconstruction requires 11 fusion events and zero**
**fissions. This seems noteworthy in light of the fact that there are more even numbers reported between 2R and the**
**basal gnathostome split 3 fissions and 4 fusions. This may attach to comment 6 above, or may reveal an unusual**
**aspect of vertebrate biology that arose briefly following the split of gnathostome and agnathan lineages but before**
**2R. The timing, mechanics and probability of this seem worthy of extensive discussion.**

It was previously argued that early vertebrate lineages experienced two contrasting modes of genome structure
evolution: i.e., some early vertebrate lineages had a relatively stable (or slowly evolving) genome structure for a
long evolutionary time, while other lineages had many chromosome fusion events in a relatively short period of
evolutionary time [Nakatani et al., *Genome Res* (2007), Nakatani and McLysaght, *Bioinformatics* (2017)]. The
proto-gnathostome lineage might have experienced a rapid transition from a phase of stable/slow karyotype
evolution to a phase of frequent chromosome fusions. The mechanism is unknown, but karyotypic reversal (from
acrocentric chromosomes to metacentric chromosomes) by Robertsonian fusions is observed in mammals [Pardo-
Manuel de Villena and Sapienza, *Genetics* (2001)], and a similar phenomenon might have occurred in the proto-
gnathostome lineage.

We added this paragraph in Supplementary Information (Section 4.3).

**[Comment 18] 9) With respect to phylogenetic reconstructions, the authors raise an important point. “Intriguingly,**

we observed large numbers of vertebrate paralogs between most pairs of homoeologous proto-gnathostome and
proto-cyclostome chromosomes, seemingly suggesting a contradictory model where quadruple proto-gnathostome
chromosomes and sextuple protocyclostome chromosomes were created before the gnathostomes-cyclostomes
split.” It is fairly well understood that this pattern pervades these trees and was previously understood to be due to
long branch attraction and similar artefactual convergence related to long term substitution biases in lampreys. The
authors also mention the possibility that this is explained by allopolyploidization, but do not mention these more
mundane explanations, or other alternatives such as true differences in timing of duplication events and hidden
paralogy.

We added a sentence and mentioned the difficulties in gene tree inference: “This observation may be explained by
difficulties in gene tree inference due to the high GC content and strong codon bias in the lamprey genomes.”

This part of the discussion is also a bit confusing because earlier in the manuscript 2R is discussed in the context of
an allopolyploidization event, whereas this seems to be focusing on peri-1R patterns (or pre-1R?).

We deleted some text in this paragraph, because Reviewer 1 also commented that this part is confusing (see
Reviewer 1’s minor comment S62).

**[Comment 19]** 10) Examination of the phylogenies of some 6-fold duplicated in lamprey may shed additional light
on the timing of presumptive duplications. As was performed previously for sea lamprey hox clusters. It would be
nice to see this done for a larger number trees that were generated as part of their analysis pipeline. This would also
give readers a better sense of the underlying data.

We had already tried such an analysis, but we found only a small number of lamprey genes with six or more
retained ohnologs. Our analysis showed that cyclostome-specific paralogs are enriched in a few pairs of proto-
cyclostome chromosomes (see Supplementary Figures S9–S13, confirming the analysis by J.J. Smith and
colleagues described in the germline sea lamprey genome paper [Smith et al., *Nature Genet* (2018)].

**[Comment 20]** 11) The paragraph starting at the bottom of page 11 related to the asymmetric and unequal
contribution from the subgenomes could use further development. Which chromosomes are thought to belong to
the A and B subgenomes in Figure 4?

We added subgenome information in Figure 6 (Fig. 4 in the initially submitted manuscript) (i.e. proto-gnathostome
chromosomes are surrounded by thick black line if they belong to the subgenome with a higher rate of gene loss).

Do the authors propose that these have evolved in a manner similar to *Xenopus* wherein one of the subgenomes has
lost more paralogs than the other? Please discuss further the degree of asymmetry observed here, and compare to
that of *Xenopus* and other systems where it has been observed.

In our reconstruction, the ratio of retained genes between the two subgenomes is 2.25, which is considerably larger
than previously reported ratios of paleo-allopolyploids: 1.47 for *Brassica*, 1.46 for maize, 1.24 for sorghum, 1.17
for *Arabidopsis* and 1.35 for *Xenopus laevis* [Garsmeur et al., *Mol Biol Evol* (2014); Session et al., *Nature* (2016)].

We added this sentence in the main text in Subsection “Inferred scenario of early vertebrate genome evolution”.

[Comment 21] 12) The paragraph related to AIS and microchromosomes could also use a bit of development as it
is a bit difficult to understand. Is the “immune supercomplex” idea central to the “big bang” theory? It seems that
this idea should have fallen by the wayside some time ago, but perhaps this should be developed further?

The “immune supercomplex” idea is a model for explaining the “immunological big bang”. See [Flajnik *Nat Rev*
*Immunol* (2018); Kaufman, *Annu Rev Immunol* (2018), the last paragraph in Page 394] for recent reviews.

Additionally, the section appears to argue that more immune genes were inherited from the subdominant (b) genome.
Is this correct? Some of it would be nice to see this cleared up.

We appreciate this suggestion. We performed an analysis of gene ontology enrichment between the two subgenomes,
and found that the genes derived from the shorter subgenome (with a higher rate of gene loss) are enriched with
defense/immunity proteins. We added the following sentence in the main text: “In addition, we observed
functional biases between the two subgenomes: the human genes in the segment derived from the
shorter subgenome were enriched with ‘defense/immunity protein’ in PANTHER Protein Class (FDR
$q = 2.75 \times 10^{-13}$, see Supplementary Information Section 4.5).”

Additionally, this clause seems like it might be missing a reference “corroborates the view that a primordial
‘adaptive’ immune system emerged in the ancestral vertebrate genome and later turned into the intricate
gnathostome-like AIS through 2R.”

We added the following review papers: [Flajnik and Kasahara, *Nat Rev Genet* (2010); Flajnik, *Nat Rev Immunol*
(2018); Ohta et al., *J Immunol* (2019)].

[Comment 22] 13) The Methods, or large portions thereof, should be elevated to the main body of the manuscript
and presented in a manner that is accessible to a broad audience, including assumptions and caveats that relate to
inferring duplications and pre-duplication states.

Our reconstruction method consists of two steps. In the first step, we reconstructed the proto-vertebrate genome
using the probabilistic macrosynteny model, which was published as a separate paper [Nakatani and McLysaght,
*Bioinformatics* (2017)]. In the second step, we reconstructed the proto-cyclostome and proto-gnathostome
genomes. In this step, we employed the method previously described in [Nakatani et al., *Genome Res* (2007)]. We
extended the previous method so that multiple post-WGD genomes can be used for reconstruction. In addition, the
possibility of fusions/fission between 1R and 2R is also explored during the search for the optimal reconstruction
(called set partition in Supplementary Information). The basic idea of the second step is now described in the main
text and illustrated in Figure 4 (Fig. S5 in the initially submitted manuscript). The fundamental idea (or
assumptions) in our reconstruction is that paralogs are distributed non-randomly: they should be found mostly
between duplicated chromosomes [Nakatani et al., *Genome Res* (2007)]. There are several caveats. First, it would
be difficult to obtain a reliable reconstruction if the available genomes have been shuffled extensively. For
example, teleost genomes are known to have had high rates of chromosome fusions (before the teleost-specific
WGD event) and intra-chromosomal rearrangements, and thus teleost genomes are not suitable for the proto-
vertebrate reconstruction. Second, we should avoid relying too much on a single genome, since it might be affected
by lineage-specific rearrangements, genome assembly errors, limited contiguity of scaffolds, etc. For this purpose,

we used multiple post-WGD genomes in our reconstruction. Third, small chromosomes in the proto-vertebrate and
proto-cyclostome genomes tend to be less reliable than large chromosomes, because it is difficult to identify small
synteny blocks in the post-WGD genomes, especially when post-WGD genome assemblies are not complete and
chromosomes are divided into multiple short fragments. For this reason, we discussed that the majority of the
Japanese lamprey genome was covered by the six-fold duplicated proto-cyclostome chromosomes, and confirmed
that the largest five proto-vertebrate chromosomes were six-fold duplicated in the proto-cyclostome genome.
In the revised manuscript, we added Supplementary Movie 1 and added Figure 4 to the main text (adapted from the
previous supplementary figure S5) in the revised manuscript. In addition, we discussed the limitation of the proto-
cyclostome reconstruction as follows:

“Although the current lamprey genomes might still be incomplete and some chromosomes might be
fragmented, such limitations are unlikely to have substantially biased our analysis. First, if the proto-
cyclostome genome was shaped by three rounds of tetraploidization, that would additionally require
a large number of subsequent chromosome fusions to explain the current genome arrangement (for
example, 45 post-tetraploidization fusions are required to obtain the chromosome number of sea
lamprey germline cells: $18 \times 8 - 45 = 99$). However, we found that the lamprey lineage had remarkably
low rates of inter-chromosomal rearrangement (Supplementary Fig. S5) over ~ 500 million years⁴² of
cyclostome evolution. Specifically, our proto-cyclostome genome reconstruction shows large-scale
fusions and translocations affecting only 22 out of 141 Japanese lamprey scaffolds and only 19 out of
151 sea lamprey scaffolds that have at least 10 genes. The exceptionally low rate of inter-
chromosomal rearrangement and the haploid chromosome number of ~ 99 in the germline sea
lamprey genome⁴³ are consistent with our evolutionary scenario in which the lamprey chromosome
number is explained approximately as $18 \times 6 = 108$ with several subsequent fusions. Second, even
though some tiny chromosomes might be missing in the current proto-cyclostome reconstruction,
large chromosomes (e.g. Hox-bearing chromosomes duplicated from Pvc1) are unlikely to be
missing entirely; therefore, our reconstruction is particularly reliable for the largest five proto-
vertebrate chromosomes (i.e. Pvc1, 3, 10, 13 and 17), which consistently exhibited a multiplicity of
six. Thus, the high coverage (60.3%) of the Japanese lamprey genome by six-fold duplicated proto-
cyclostome chromosomes suggests that extant cyclostome genomes are paleo-dodecaploids (i.e. the
chromosome number increased as 18×6 due to tetraploidization and hexaploidization), which might
be similar to the situation in sturgeon where a species (*Acipenser brevirostrum*) with ~ 180
chromosomes is considered to be a hexaploid of a tetraploid ancestor with ~ 60 chromosomes⁴⁴⁻⁴⁶.”

**[Comment 23] 14) The authors should elevate reporting of assembly improvement to the Results section and develop**
**a figure that more effectively relays improvements. Comparing the cumulative rate of increase in assembly size across**
**increasing scaffold lengths (often included in standard DoveTail reports) would provide important perspective.**
The key statistics of the current genome assemblies (contig and scaffold N50 values) have been mentioned in the
main text. In addition, as suggested by the reviewer, we have now included a supplementary figure (Fig. S1; cited in

Supplementary Information) which shows the cumulative rate of increase in the assembly size across increasing
scaffold lengths which clearly shows a higher level of contiguity in the current assemblies. We believe that there is
no need to include this data in the main text.

**[Comment 24]** 15) Code and sequence availability: The authors state that “reconstruction software/code is available
on request.” However, I would strongly recommend that the code be released on GitHub (or similar) as soon as
possible and that reconstructions be include as supplemental data files or placed in another permanent repository.
Access to the code and reconstructions are necessary in order to properly assess their findings, and would have likely
changed some of the comments made above. An embargoed release of the genomes would also be useful, and has
become common practice, although I understand that this is not necessarily standard practice at this point in history.
At present, the code is not publicly available for download due to copy right issues involving the graphics module
integrated in our code with some modifications. Instead, we added the reconstruction dataset including information
of orthologs, paralogs and segments as Supplementary Data 1.

The Japanese lamprey and elephant shark genome assemblies generated as part of this study have been submitted to
GenBank under the accession numbers WFAB00000000 and WEZY00000000, respectively. These genome
assemblies will be available in the public domain before the publication of our manuscript.

Reviewer #3 (Remarks to the Author):

It was a delight to read the manuscript “Reconstruction of proto-vertebrate, proto-cyclostome and proto-gnathostome
genomes provides new insights into early vertebrate evolution” by Nakatani and others. The study reconstructs the
genome of the first vertebrates at chromosome/level, by using high-quality genomes of a lamprey and the elephant
shark. The results offer a highly detailed and resolved picture of the genome of early vertebrates, gnathostomes, and
cyclostomes, shedding new light on the debate about the whole genome duplications. I found the results on
microchromosomes very original and interesting. The design of the analyses and the manuscript writing are great,
and the conclusions highly relevant to our knowledge of vertebrate origins. I would like to commend the authors for
their efforts.

**[Comment 25]** My only criticism is about the discussion about the evolution of the adaptive immune system and
MHC. While I think this is very interesting and the data/analyses certainly support the claims, this is only touched in
the Discussion section and seems a bit out of the blue. I’d like to suggest to support this either with another section
in Results or maybe a figure.

We apologize that our discussion of the adaptive immune system was abrupt. We presented the reconstructions and
direct implications in Results, and discussed how our reconstructions as a whole may change our view on the
evolution of early vertebrate in Discussion section. In order to address Reviewer 3’s concern, we revised the
manuscript as follows. First, we revised Figure 6 and showed the inferred positions of the MHC, NKC and LRC
clusters in the proto-gnathostome genome, and showed that these complexes are found in the shorter subgenome.
Second, we revised the introductory sentence and clarified that adaptive immune system might have evolved through
genome hybridization as follows: “In particular, our reconstruction suggests that genome hybridization might have
contributed to the origin of the adaptive immune system (AIS), which is a prime example of a major evolutionary
innovation in early vertebrates.” Since the emergence of gnathostome-like AIS through genome hybridization is a
novel hypothesis, we believe that Discussion is the most suitable section.

**[Comment 26]** Along those lines, another suggestion to make the paper interesting to a wider audience would be to
add a figure in which the different hypotheses about 1R, 2R, and cyclostome-specific WGS are mapped to a
phylogeny. This would help some readers to understand better the evolutionary scenario, as well as show the
phylogenetic relationships of all the animals involved, which are never shown. If the authors decide to follow this
advice, I’d also add photos of the sequenced organisms here. If the paper has reached the limit of displayed items, I
think Figure 3 could be easily moved to Supp data, as it is not that informative and there are enough figures with
dots in the paper already (this is a very “dotty” paper!).

We added Figure 1 to show typical alternative scenarios and the phylogenetic relationship among representative
species used in our study.

I do not have any major criticisms, but I have some other comments and questions that I hope the authors can

kindly address:

**[Comment 27]** 1) Page 3, I wonder if the authors could add a reference to the number of vertebrate species. This
number keeps creeping up as time goes!

We have now included a link (<http://vgpdb.snu.ac.kr/splist/>) which directs the reader to a comprehensive list of all
~71,000 extant species of vertebrates.

**[Comment 28]** 2) Page 3, I'd like to suggest replacing "degenerate" by "simplified", as the first has other
connotations.

Revised as suggested.

**[Comment 29]** 3) Page 6 and others, I wonder if the selection of genomes to perform comparative synteny analyses
was just based on evolutionary rates or also on high contiguity genomes.

We chose high contiguity genomes in Ensembl.

**[Comment 30]** 4) Page 10, first sentence, maybe I need more coffee but I did not understand the bit between
parentheses "(or diverged before 1R)". I would like the authors to clarify this in the text.

The scenario of divergence before 1R is still possible at this point in the manuscript. The scenario was concluded to
be unlikely later in the manuscript because the two lineages share a large number of paralogs. Since multiple
reviewers were confused by this phrase, we deleted "(or diverged before 1R)".

**[Comment 31]** 5) Page 10, the sentence "the ancestral metazoan animal genome", the paper is comparing a
mollusc vs a vertebrate. It should say "Bilaterian" rather than "metazoan"

We wrote metazoan because we showed a macrosynteny conservation with the *Trichoplax* genome. We revised the
main text to "bilaterian animal genome".

**Specific comments for manuscript NCOMMS-19-37344-T - “Reconstruction of proto-vertebrate,**
**proto-cyclostome and proto-gnathostome genomes provides new insights into early vertebrate**
**evolution” by Nakatani et al.**

**###S1:** Page 2, line 1: Is it necessary to center humans in this conversation? We are after all a very
small part of this story. I suggest “The genomes of vertebrates, including humans, have been
shaped by...”

Revised as suggested.

**###S2:** Page 2, line 2: I suggest starting a new sentence at “... tetraploidization events. These have
had a lasting impact...”

We revised the part as “... events, which have had a lasting impact ...”

**###S3:** Page 2, line 3: Strike “However,”

Revised as suggested.

**###S4:** Page 2, line 6: The authors suggest that the lack of a proto-cyclostome genome
reconstruction has been a limitation in sorting out the timing of the cyclostome-gnathostome
divergence relative to the early vertebrate tetraploidizations. The proto-cyclostome genome
reconstruction is undoubtedly a great tool to resolve this issue, but the limitations truly lie with
the lack of a reliable, mapped, cyclostome genome as well as the unique composition of
cyclostome genomes and sequences. The authors discuss these issues in the manuscript. Can the
statement in the abstract be tempered to reflect this? I suggest that this sentence can be removed
completely without affecting the abstract.

The sentence was deleted as suggested.

**###S5:** Page 2, line 11-15: I suggest something like “**Our model suggests that** cyclostomes
diverged from **the lineage leading to** gnathostomes after a shared tetraploidization...” In this same
long sentence I suggest the following grammatical review - “; **that** the cyclostome lineage
experienced...”, “; **that** 2R in the gnathostome lineage **was an** allotetraploidization **event...**”, “; and
**that subsequently**, biased gene loss **from one of the** subgenomes...”

“Our model suggests” is confusing, because the macrosynteny model is one of several parts of our
reconstruction method. We divided the long sentence by “First, Second, Third,”

**###S6:** Page 2, line 13: It’s a tautology to write “the **cyclostome** lineage experienced a **cyclostome-**
**specific hexaploidization**”.

We paraphrased it as “the cyclostome-lineage experienced an additional hexaploidization.”

**###S7:** Page 2, last sentence of Abstract: Again, this centers humans a bit too much in the story.
The authors do mention the possibility of their findings informing our knowledge of human
disease genes (I have some additional comments about this below), but because the authors have
not identified any specific disease genes, not used any specific human disease genes as examples
in this study, I think it is misleading to mention human disease genes in the abstract.

The phrase about human disease was deleted.

**###S8:** Page 3, line 2. The word “simple” can be removed. This is a common pitfall when writing
about evolution. “Simple” in relation to what? Surely even these early chordates had some
measure of complexity?
“Simple” was deleted.

**###S9:** Page 3, line 4: Add comma - “... species, including humans.”
Revised as suggested.

**###S10:** Page 3, line 9: Change to “Osteichthyes, represented by ray-finned fishes and lobe-finned
fishes, including tetrapods”. The clade of lobe-finned fishes (Sarcopterygii) includes tetrapods, it’s
not separate from it.
Revised as suggested.

**###S11:** Page 3, line 10: I’m not sure that this opinion of cyclostomes is so general any more.
Perhaps this could be changed to “Cyclostomes are **sometimes** thought to be...”
Revised as suggested.

**###S12:** Page 3, line 13: I suggest “**seemingly** degenerate”.
We wrote “seemingly simplified”, due to the comment of Reviewer 3.

**###S13:** Page 3, line 15: Start a new sentence at “For example,”.
Revised as suggested.

**###S14:** Page 3, lines 20-22: This sentence (“Evolutionary innovations...”) is very long and tricky to
follow. Please break up and clarify.
The sentence was simplified as follows: “Evolutionary innovations at the origin of vertebrates have
been proposed to be the result of ancient tetraploidization events that generated additional
copies of the entire genome^{9,10}.”

**###S15:** Page 3, lines 22-23: “This view is now widely accepted” seems to refer to the duplication
followed by sub/neo-functionalization scenario, and not to the tetraploidizations themselves,
which I think is the point. Please clarify.
It refers to the view that evolutionary innovations at the origin of vertebrates were facilitated by
the WGD events. We simplified the previous sentence for avoiding a confusion in this sentence.

**###S16:** Page 4, lines 7-8: Isn’t “the tendency of lamprey ohnologs to cluster outside gnathostome
gene clades” what is to be expected, i.e. isn’t this the position that follows the taxonomy
correctly? I know what the authors mean - that cyclostome sequences tend to occupy
“paradoxical” positions in gene trees, but surely the position that the authors have described as
“paradoxical” is the expected one?
The branching pattern of gene trees may not reflect the correct phylogenetic relationship of those
species, because lamprey ohnologs tend to cluster together due to the high GC-content of lamprey
genes.

**###S17:** Page 4, line 26: It’s misleading to describe the species themselves as “early branching
vertebrates”. At least the lamprey is a **representative** of an early branching vertebrate lineage, but

the cartilaginous fishes are just as “early” as the bony fishes, so this description is incorrect. Please
clarify that the two species whose genomes have been sequenced and assembled **represent two**
**crucial divergence points in the evolution of vertebrates.**

We revised the sentence as Reviewer 1 suggested as follow: “These two species represent two
crucial divergence points in the evolution of vertebrates.”

**###S18:** Page 4, lines 29-32: This sentence (“The major advantage...”) is very long and tricky to
follow. Please break up and clarify.

The sentence was simplified as follows: “The major advantage of our method is that it has a high
tolerance to reconstruction uncertainty caused by small-scale rearrangements that have
accumulated over a long evolutionary time.”

**###S19:** Page 4, line 33: Syntax error - “... we were able to reconstruct **the first the** proto-
cyclostome genome...”

We fixed this.

**###S20:** Page 5, lines 1-2: The statement “In addition, our reconstruction of the proto-
gnathostome genome...” comes a bit prematurely. The authors have not yet stated that it was an
aim to do this reconstruction, as they stated with the proto-cyclostome genome reconstruction on
the previous page. I suggest “In addition, **we reconstructed the proto-gnathostome genome using**
**the same strategy, with a higher coverage of extant gnathostome genomes than previous**
**reconstructions...**” The authors have also neglected to mention that their sequencing and
assembly of a new elephant shark genome was crucially integrated into this reconstruction.
Highlight this fact - it’s one of the major advances described in this paper! Similarly, the authors
could highlight how crucial a chromosome-level assembly of a lamprey genome, compared with
previous lamprey genome assemblies, was to their reconstruction.

We had already emphasized the importance of the elephant shark and Japanese lamprey genomes
in appropriate positions in the main text. We revised the sentence as follows: “In addition, using
the elephant shark genome, we reconstructed the proto-gnathostome genome with a higher
coverage of extant gnathostome genomes than previous reconstructions”.

**###S21:** Page 5, lines 7-8: The authors write that they “provide new insights into the genetic basis
underlying evolutionary innovations”. This is an overstatement. Surely, this is a possible future
impact of this study, but as for the present paper there is only a brief and very general discussion
about the evolution of the adaptive immune system. That’s it. Please temper the tone of this
statement to something that reflects the content of this paper more truthfully.

It seems that Reviewer 1 misunderstood our arguments in Discussion (see our response to
Comment S75). We revised Figure 6 (Fig. 4 in the initially submitted manuscript) to show
presumed ancestral positions of the genes in MHC, NKC and LRC in the proto-gnathostome
genome (see also Reviewer 3’s Comment 25).

**###S22:** Page 5, lines 8-9: This statement is only true if the authors will share the new genome
assemblies in an easily searchable or browsable form, or, even better, share a detailed searchable

map of their reconstructions. These possibilities are not mentioned at all in the paper. If the
authors do not plan to share these resources, then the reconstructions will not serve as references
of any kind.

We have included GenBank accession numbers of the new genome assemblies, and the
reconstruction dataset has been made available as Supplementary Data 1.

**###S23:** Page 5, lines 14-16: This is a big overstatement. But to give this statement any credence,
the authors should at the very least provide some examples and references of where this has been
the case (I have more comments about this further down). They have not identified any specific
disease genes linked to their findings, nor used any specific human disease genes as examples in
this study. It is a pity because the study doesn't need it. There are many of us who follow the
author's work and understand its value without centering it on humans and our pathologies.
See our response to Reviewer 1's [Comment09].

**###S24:** Page 6, lines 30-32: The second clause of this sentence is tricky to follow. I suggest "... we
predicted 18,727 **protein-coding genes in the elephant shark genome assembly** and 19,455
protein-coding genes in the Japanese lamprey genome assembly." This is only 5 words longer.
Revised as suggested.

**###S25:** Page 6, line 3: If it does not make the manuscript exceed the word count, please detail
which four gnathostome genomes here. This is important because if the elephant shark is one of
them, the authors should highlight how essential their new genome assembly is for their analyses.
We added "including human, chicken, spotted gar and elephant shark".

**###S26:** Page 6, line 5: Here is the first reference to "18 chromosomes". See my general comment
about this above.
See our reply to Reviewer 1's general comment about this point.

**###S27:** Page 6, lines 11-12: Since the names "scallop" and "placozoa" are used as general terms,
and not as specific common names, the parenthesis around the binomial names *Chlamys farreri*
and *Trichoplax adhaerens* should be removed.
Revised as suggested.

**###S28:** Page 6, lines 12-14: Move this text ("also see Supplementary Fig. S3...") out of the
parenthesis and make it a new sentence.
We revised the text as "(Fig. 2, also see Supplementary Fig. S4)."

**###S29:** Page 6, line 20: Use commas around the sub-clause "that were not used in the proto-
vertebrate reconstruction".
Revised as suggested.

**###S30:** Page 6, line 25: Add "the" for "**the** Japanese lamprey".
Revised as suggested.

**###S31:** Page 6, lines 25-16: Use commas around the sub-clause “in addition to the existing
‘hybrid’ genome assembly of the sea lamprey”.
Revised as suggested.

**###S32:** Page 6, line 28: Add a comma after “contentious”.
Revised as suggested.

**###S33:** Page 6, line 29 - page 7, line 2: This section, removing “For example”, should be moved
down to just before the paragraph starting “To distinguish between different polyploidization
models...” This way, these different models, which are complex scenarios, are still fresh in the
mind of the reader. In addition, the alternative models of polyploidization seems as an aside,
“just” an example”, the way they are described now. When, in fact, the reader must be
familiarized with them to understand the rest of this section. The text can easily go from “... which
have remained contentious, even after the sequencing of the sea lamprey genome”, to “In the
present study, we have generated...” without losing clarity or jumping to a separate context (the
alternative scenarios).
We decided to keep the current presentation order. The readers need to know alternative
scenarios before they read about the reconstruction method, because the method is specifically
designed to explore the possibility of those alternative scenarios.

**###S34:** Page 6, line 32: Start a new sentence at “Another possibility...”
Revised as suggested.

**###S35:** Page 6, lines 29-34: It’s not clear that the authors are referring to 1R here, the same
tetraploidization (1R) is mentioned in two scenarios but makes it look like they are different
tetraploidizations. I suggest “... could be due to additional tetraploidization events in the
cyclostome lineage; alternatively, they could be the result of one shared tetraploidization event
(1R) at the base of vertebrates followed by segmental (chromosome) duplications in cyclostomes.
Another possibility is that the cyclostome lineage experienced a hexaploidization event (whole-
genome triplication) following the shared 1R, thus giving rise to $1 \times 2 \times 3 = 6$ Hox clusters.
At this point in the manuscript, we are not discussing if any WGD events were shared between the
proto-cyclostome and proto-gnathostome lineages.

**###S36:** Page 7: Throughout this section of the paper I had a very difficult time distinguishing
between blocks, segments, scaffolds and chromosomes. Sometimes a segment can be the same as
a scaffold, right? And several segments can be “assembled” into a proto-chromosome? Where do
“blocks” come in? Please define these terms clearly. This confusion is carried over to Figure 2.
Segments and blocks refer to chromosomal regions in general (e.g. synteny blocks). Segments are
obtained by using a “segmentation” algorithm as explained in Supplementary Information Section
3. Segments may be whole scaffolds/chromosomes, and they are the building blocks of the
reconstructed chromosomes as explained in Supplementary Information Section 3.

**###S37:** Page 7, lines 4-8: This sentence is very long and difficult to follow. The authors should
 move the parenthesis to a new sentence following this, e.g. "... by combining lamprey genomic
 segments into 104 proto-cyclostome chromosomes (Figure 2). Genomic segments in this case are
 blocks of conserved synteny that were inferred..."
 The sentence was divided and shortened.

**###S38:** Page 7, line 6: Remove "the" from "the cyclostome evolution".
 Revised as suggested.

**###S39:** Page 7, line 11: I suggest "because **each of the segments showed conserved synteny with**
 **two different sea lamprey scaffolds.**"
 Revised as suggested.

**###S40:** Page 7, lines 11- 16. Start a new sentence here, e.g. "In our reconstruction..."
 Furthermore, this sentence is very long and tricky to follow, and the references to Fig. 2 interrupt
 the flow and make it even more difficult to understand. I also have some methodological concerns
 here. I suggest the following: "In our reconstruction, the linkage of the two segments on
 Scaffold35 was restored in one of the proto-cyclostome chromosomes (green in Fig. 2b) with
 support from Pacific lamprey linkage markers. On the other hand, the two segments on Scaffold2
 were assigned to different proto-cyclostome chromosomes based on the number of paralogs
 shared between them, which indicate an origin in a whole-genome duplication"
 We do not exclude the possibility of aneuploidy (chromosome-wise duplication) at this point.
 Whole-genome duplication is argued based on Fig. 3d (Fig. 2d in the initially submitted
 manuscript).

I must say that the count of number of paralogs doesn't convince me much - I can count (roughly?)
 the same number of dots, 12, in Fig. 2c between the two Scaffold35 segments and between the
 two Scaffold2 segments.

It seems that Reviewer 1 misunderstood the figure. The numbers of paralogs are shown in the red
 rectangles in the figure below.

Where do the authors draw the line for considering a number of paralogs as evidence for or
against linkage?

We calculated the significance of the number of paralogs as explained in Supplementary
Information Section 3.

In addition - to invoke the linkage on Scaffold 35 as a proof that the segments indeed were part of
the same proto-chromosome is a circular argument.

We disagree that it is a circular argument. Lamprey genome assemblies consist of large numbers
of scaffolds and if a chromosome sequence is represented by several short scaffolds in the sea
lamprey genome, the syntenic Japanese lamprey chromosome is also partitioned into several
short segments in our analysis. In other words, lamprey segments tend to be over-fragmented.
When these segments are mapped to the same proto-vertebrate chromosome, we have two
possibilities about their origin in the proto-cyclostome genome: (1) they originate from the same
proto-cyclostome chromosome; or (2) they originate from duplicated proto-cyclostome
chromosomes. These segments are assigned (1) to the same proto-cyclostome chromosome if
they do not share significantly large numbers of paralogs in order to alleviate the
overfragmentation of the lamprey segments; or (2) to duplicated proto-cyclostome chromosomes
if they share significantly large numbers of paralogs. This was described in Supplementary
Information Section 3.

Why then wasn't the linkage on Scaffold 2 seen as an argument for the ancestral linkage of these
segments?

It was explained in Supplementary Information Section 3. The algorithm disallows two segments
that share a significantly large number of paralogs to be assigned to the same proto-cyclostome
chromosome.

This section of text as well as the paragraph that follows, makes the authors' analyses seem almost
arbitrary, with "hand-picked" results, when they should rely on carefully considered algorithms.
Please clarify this section of the paper so that the reader isn't left with the same impression.

We relied on the algorithm described in Supplementary Information Section 3. Although we
disagree that our analyses were arbitrary, we revised the section and simplified the text as follows.
"The major advantage of this reconstruction method is its robustness against lineage-specific
rearrangements and fragmentation of genome assemblies. For example, Japanese lamprey
Scaffold2 was partitioned into two segments (Fig. 3a) because each of the segments showed
conserved synteny with two different sea lamprey scaffolds; in our reconstruction (Fig. 3b), and
the two segments on Scaffold2 were assigned to different proto-cyclostome chromosomes
because they share a significantly large number of paralogs (dots in Fig. 3c). Thus, our
reconstruction-based analysis is more reliable than scaffold-based analyses used in previous
studies^{18,19,26} and provides the first opportunity to conclusively resolve the controversy over the
origin of the proto-cyclostome genome."

**###S41:** Page 7, line 21: I've already suggested that the authors should move a section of text from
the preceding page to this location of the paper. The paragraph starting here is very tricky to
follow, starting with the first sentence. I suggest something like - "To distinguish between **these**
alternative polyploidization models, we introduced a measure we have called multiplicity, i.e the
number of **reconstructed** proto-cyclostome chromosomes that **correspond to each** of the
**reconstructed** proto-vertebrate chromosomes."

The phrase was revised as "we introduced a measure we have called multiplicity".

Avoid writing that multiplicity equals "the number of proto-cyclostome chromosomes **originating**
from individual proto-vertebrate chromosomes" - This would be a circular argument. This
describes a conclusion from the analysis, not how the analysis was made.

It is a result of our reconstruction, and it is not a circular argument (see Supplementary
Information Section 3). We reconstructed duplicated chromosomes, and we concluded that they
were created by whole-genome triplication.

The authors have not written here how this multiplicity was calculated, how the correspondence
between proto-cyclostome and proto-vertebrate chromosomes was made, and I could not find a
clear description of this in the supplementary text either. This again makes the analyses seem
arbitrary and circular.

We are unsure of the source of confusion here. As we wrote, we counted the number of
duplicated proto-cyclostome chromosomes for each proto-vertebrate chromosome. The clear
description of our reconstruction method can be found in Supplementary Information Section 3.
Our macrosynteny algorithm infers the probabilities that each lamprey segment was derived from
each proto-vertebrate chromosome (see Fig. 1 in [Nakatani and McLysaght, *Bioinformatics*
(2017)]). Then, individual segments were assigned to the proto-vertebrate chromosome with the
largest reconstruction score, as described in Supplementary Information Section 3.2.2. These
segments were reconstructed into proto-cyclostome chromosomes by set partitioning, as
described in Supplementary Section 3.3.

It is briefly mentioned on page 33 of the supplement, but that's it.

It is written in Sections 3.2 and 3.3, Pages 21—28 of the initially submitted Supplementary
Information file.

Is it part of section 3.3.3 on pages 27-28 of the supplement? The only reference to this "we
extended it to also enumerating set partitions into more than 5 proto-cyclostome chromosomes."
Is this it?

The reconstruction of proto-cyclostome chromosomes was described in Section 3.3 from Page 25
to Page 28 of the initially submitted Supplementary Information file. Set partitioning is introduced
in Page 25. Significance of a set partition is explained in Section 3.3.3.

Was the set partition with 6 proto-cyclostome chromosomes the most significant?

We wrote "For each of Pvc1–Pvc17, we enumerated all set partitions of the clusters, and chose the

optimal set partition with the most significant distribution of orthologs and paralogs as the proto-
cyclostome chromosomes” in Page 26 of the initially submitted Supplementary Information file,
and Fig. 3 shows that six-fold duplication was the most significant for nine out of 18 proto-
vertebrate chromosomes.

In any case, describe briefly how this was done in the main text of the paper, and include a clearly
marked “multiplicity calculation” (or similar) description in the supplementary text.

We thank Reviewer 1 for this suggestion, but we just counted the number of proto-cyclostome
chromosomes. Instead of repeating the same explanation, we made a movie (Supplementary
Movie 1) explaining the reconstruction method.

###S42: Page 7, line 24: Here is another mention of 18 proto-vertebrate chromosomes. The
authors should write that they arrived at 17 proto-vertebrate chromosomes plus PrvUn. See my
general comment above.

See our reply to Reviewer 1’s general comment.

###S43: Page 7, line 24-25: The sentence “We found that nine out of the proto-vertebrate
chromosomes **were duplicated** into six paralogous proto-cyclostome chromosomes.” In my
opinion, the authors should not write this conclusively about their results at this point of the
paper. This statement is the **conclusion** that they arrive at, but for the reader it does nothing to
explain **how** they arrived at this conclusion.

We rephrased this as ‘Our analysis indicates that ...’ The observation (that nine out of 18 proto-
vertebrate chromosomes were duplicated into six paralogous proto-cyclostome chromosomes)
was the inference result of our reconstruction method. How the method arrived at this result is
explained in Supplementary Information Section 3.3. Our conclusion/interpretation is that the
observation indicates six-fold duplication of the entire genome through one whole-genome
duplication and one whole-genome triplication.

What did the results look like?

The resulting reconstruction of the proto-cyclostome genome was illustrated in Fig. 3, Fig. 6, Fig.
S4, Fig. S6, Fig. S7 and Figs. S1014.

Are there any alternative scenarios that could explain the same results? If so, how were
alternative scenarios discarded?

We chose the most significant reconstruction from millions of alternative scenarios as explained in
Section 3.3. The calculation of significance is explained in Section 3.3.3. See also Supplementary
Movie 1.

###S44: Page 7, line 28: Clarify that this first tetraploidization is 1R. For a moment I thought the
authors suggested that both the tetra- and hexa-ploidizations occurred at the base of cyclostomes,
which confused my reading of the paper.

At this point of the manuscript, we have no information to judge if the polyploidization events
were shared between the proto-gnathostome and proto-cyclostome lineages. Thus we described

that one tetraploidization and one hexaploidization occurred between the proto-vertebrate and
proto-cyclostome.

**###S45:** Page 7, lines 30-34: This is a very long sentence that is difficult to follow. Please break up
and clarify.

Revised as suggested.

**###S46:** Page 8, line 1: The authors have not described how many proto-cyclostome chromosomes
their reconstruction resulted in. This would seem like an obvious result to share, especially in the
context of discussing the number of chromosomes in extant lampreys.

It was already written in the main text. "In the present study, we have generated the first
reconstruction of the proto-cyclostome genome by combining lamprey segments ... into 104
proto-cyclostome chromosomes ..."

**###S47:** Page 8, line 8: I suggest changing "obtained" with "produced".

Revised as suggested.

**###S48:** Page 8, lines 10-11: It's not clear here that the authors are describing their newly
sequenced/assembled elephant shark genome. Highlight the fact that this genome assembly is
new to this study.

We added "our newly sequenced" as suggested.

**###S49:** Page 8, line 13: Change "confirmation" with "support", or "additional support".

Changed to "additional support".

**###S50:** Page 8, line 13-14: It was not the "proto-gnathostome" lineage that underwent the two
tetraploidizations. At least 1R occurred in a "proto-vertebrate". The authors found the evidence of
1R/2R in their "proto-gnathostome" genome reconstruction, but 1R occurred earlier. The authors
should also be very clear to describe that 2R occurring in the lineage leading to gnathostomes is a
new finding of this study.

We revised the text as "The reconstruction provided additional support for the previous finding of
two rounds of tetraploidization between the proto-gnathostome and its invertebrate ancestor."
Whether or not 2R is gnathostome-specific is not mentioned here, because we are not discussing
the timing of gnathostome-cyclostome divergence at this point of the manuscript. The evidence of
gnathostome-specific rearrangements occurring between 1R and 2R is discussed later in the
manuscript.

**###S51:** Page 8, lines 13-14: "The proto-gnathostome lineage" could be a confusing term. If the
time estimates for 1R and 2R that have been done previously are mostly correct, then it's not at all
certain that crown gnathostomes had emerged by the time 2R happened. A key fossil to date this
node is the (likely) lobe-finned fish Guiyu at approximately 420 million years ago. The earliest fossil
showing a bony jaw is the placoderm Entelognathus, a likely stem gnathostome also dated at
approximately 420 Mya. This marks the minimum age of gnathostomes. The maximum age of
gnathostomes is more difficult to estimate, but is bounded by the emergence in the fossil record
of ostracoderms, at approximately 468 Mya. This time window overlaps with the suggested ages

for 2R, but again it is not at all clear that crown gnathostomes had emerged at this point.
Therefore, I think that it would be more accurate to write “the lineage leading to extant
gnathostomes” instead of “the proto-gnathostome lineage”.
We thank Reviewer 1 for this information. We are aware of the problem regarding the usage of
‘proto-vertebrate’, ‘proto-cyclostome’ and ‘proto-gnathostome’. However, we also think that it
will cause more confusion if we decide to avoid using those convenient terms. For example, it
might be more accurate if we change the title to “Reconstruction of genomes of the lineage
leading to extant vertebrates, the lineage leading to extant cyclostomes and the lineage ...”, but it
is not helpful for most readers. We decided to call them proto-vertebrate, proto-cyclostome and
proto-gnathostome, and we believe this slight abuse of words is helpful for most readers.

**###S52:** Page 8, lines 16-22: This paragraph about microchromosomes seems to interrupt the flow
of the text. Perhaps it could be shortened and moved down to the following paragraph, after “...
even after ~450 million years of gnathostome evolution.” The first sentence of the paragraph
““Analysis of the proto-gnathostome genome also revealed...””) could then be removed.
We kept the two paragraphs separate: one for the background information and the other for the
results of our reconstruction analysis (also see a comment from Reviewer 2 [Comment 11]).

**###S53:** Page 9, line 17: Add comma after “hypothesis”.
Revised as suggested.

**###S54:** Page 9, line 18: I suggest “... high density of genes (**including ohnologs**) in the proto-
gnathostome chromosomes...”
Revised as suggested.

**###S55:** Page 9, lines 16 and 18: Ohnologs are mentioned, but there is no description in the main
text of the paper, however brief, of how ohnologs were identified/predicted or differentiated from
other forms of orthologous genes. There is a good description in the supplementary information,
but the main text of the paper should give some understanding of this. Especially because it is
mentioned in the introduction that “our reconstructions serve as a reliable reference for accurate
annotation of ohnologs.”
We used the paralogs described in Supplementary Information Section 2.

**###S56:** Page 9, lines 22-24: This sentence is tricky to follow I suggest - “The timing of
gnathostome-cyclostome divergence relative to the two basal vertebrate tetraploidization events
(i.e. 1R and 2R) remains an unresolved issue in the field of vertebrate **genome** evolution. Remove
the reference to 1R/2R occurring in “proto-gnathostome lineage”. This is incorrect. See also my
comment above regarding “the lineage leading to extant gnathostomes” rather than “the proto-
gnathostome lineage”.
Revised as suggested.

**###S57:** Page 9, line 24-25: I suggest “we searched **our reconstructions of the proto-vertebrate...**”
Revised as suggested.

**###S58:** Page 9, line 27: Remove the parentheses and insert a comma after “models”.
Revised as suggested.

**###S59:** Page 9, line 32: I suggest “... before 2R, **but after 1R.**”
The evidence of post-1R divergence is not discussed yet at this point in the manuscript.

**###S60:** Page 10, line 2: Regarding the text in parentheses, “or diverged even before 1R”. This is a
much bigger discussion and should not be relegated to a parenthesis. If this were true, then the
authors’ own proposed scenario would be consistent with independent 1R events in cyclostomes
and the lineage leading to gnathostomes. What in their results, and indeed in previously published
studies, suggests that this is a possibility? To the best of my knowledge, the evidence points away
from this conjecture.
We discussed the evidence of post-1R divergence later in the manuscript, so we wrote the phrase
here to show that we considered all possibilities and alternative scenarios. However, the phrase
confused multiple reviewers, and thus we deleted “or diverged even before 1R”.

**###S61:** Page 10, line 5: When the authors write “we performed a gene-tree analysis”, it gives the
faulty impression that the authors created these gene trees themselves. In fact, the authors have
analyzed automatically generated Ensembl gene trees. This is a possible weak point in the
analyses, so the authors should clearly describe what they have done.
We clarified the text by revising it to say “we performed an analysis based off Ensembl gene
trees”. We inserted lamprey genes into the existing gene trees downloaded from Ensembl, as
explained in Supplementary Information Section 5. We described it as a gene tree analysis. In our
view, Ensembl Compara is one of the most comprehensive databases for comparative genomics,
and, though not infallible, they are based on genes from many vertebrate and outgroup
invertebrate species.

**###S62:** Page 10, lines 10-22: This section is very difficult to follow. It seems like a substantial part
of the description of results and the arguments are missing. The authors state **that** they arrived at
certain conclusions, but it is not at all clear to the reader **how or why** they arrived at these
conclusions. Not all of the argumentation should be left to the supplementary text. For example,
on line 11 the authors describe “homeologous proto-gnathostome and proto-cyclostome
chromosomes”, but calling them homeologous is a conclusion in itself. How did they arrive at this.
The duplicated chromosomes were inferred by our reconstruction method (so duplicated
chromosomes are results). The discussion that those duplicated chromosomes were created by
polyploidization (and not by segmental duplications or by chromosome-wise duplications) was
already written in preceding texts in the manuscript.

The following subclause, “seemingly suggesting a contradictory model...” is very unclear. How
could both quadruple and sextuple chromosomes arise at the same time? I think they authors
simply suggest that this is evidence for a shared tetraploidization at the base of vertebrates, i.e.
1R. How is this a “contradictory model”? Contradictory to what? It is near impossible to distinguish
between paralogs generated in 1R and those generated in 2R (although the authors have made a

good attempt at dating them by analyzing Ensembl gene trees), but a large amount of 1R
generated paralogs shared between gnathostomes and cyclostomes is not contradictory to
independent chromosomes rearrangements in each of the lineages. Or have the authors been able
to date the paralogs so precisely that this set of paralogous genes includes both 1R- and 2R-
generated paralogs? Also, be sure to clarify that the hypothesis of 2R being a gnathostome-specific
event is based on **their** result and this study. The fact that 2R might be gnathostome-lineage-
specific doesn't necessarily mean that it is a **later** event. The estimations of time-points for 2R, the
emergence of crown gnathostomes, and the gnathostome-cyclostome divergence all overlap, and
the authors have not done a time-estimate calculation of their own.

We simplified this paragraph because multiple reviewers did not understand the text. See also
[Comment 18] from Reviewer 2.

**###S63:** Page 10, line 17: Add “the” before “establishment”.

Revised as suggested.

**###S64:** Page 10, line 19: I would suggest that polyploidization through hybridization is common
“to some extent” in animals.

Revised as suggested.

**###S65:** Page 10, line 27: Here is another reference to 18 ancestral chromosomes when it should
be 17 (see general comment above).

Please see our reply to Reviewer 1's general comment.

**###S66:** Page 11, lines 2-3: “, which can be explained by allotetraploidization” is a repetition and
can be removed.

Revised as suggested.

**###S67:** Page 11, line 2: Add the indefinite article “A” to “A comparison with modern...”

Revised as suggested.

**###S68:** Page 11, line 9: Another reference to 18 ancestral chromosomes. Also, the formula
$18 \times 2 \times 3$ can be misleading. It's not clear here that “x2” refers to 1R.

We cannot think of any better expressions, and the description was clear enough for Reviewer 1 to
correctly guess that $\times 2$ refers to 1R.

Also, the authors have not revealed how many proto-cyclostome chromosomes their
reconstruction ended up in. Was it as neat as $18 \times 2 \times 3 = 108$?

As already written in the main text, 104 proto-cyclostome chromosomes were reconstructed.

If so, they should mention very clearly, somewhere in the text, whether their estimation of the
number of proto-cyclostome chromosomes was constrained by the 18 (17, really) proto-
vertebrate chromosomes they had already reconstructed.

We already discussed the proto-cyclostome chromosome number in the main text and in Figure 3.

**###S69:** Page 11, line 16: “Evolutionary hexaploidy” is not an accepted term and could be
confusing. Simply removing “evolutionary” would clear it up. Alternatively, I suggest something
like “There are several documented examples of hexaploidy giving rise to new evolutionary
lineages”.

Revised as suggested.

**###S70:** Page 11, lines 25-26: The authors of this study are not the first to suggest this. See
Vertebrate evolution by interspecific hybridization – are we polyploid? by Jürgen Spring in FEBS
Letters 400, 2–8, 1997, for an early-ish example. They are not the first to suggest that
hybridization played a role at the early stage of vertebrate evolution. In more general terms,
hybridization has been part of the discussion since Susumu Ohno’s time - he writes about it in the
“Mechanisms of Gene Duplication” chapter of Evolution by Gene Duplication in reference to both
auto- and allo-tetraploidy, and he mentions triploidy, though he does write that “Such an
interesting oddity, however, is a side issue of vertebrate evolution.” At this point of the paper, the
authors should perhaps temper their discussion to reflect the long ongoing discussion surrounding
the role of hybridization in polyploidization and the origin of vertebrates. In the supplementary
text, the authors contrast “their” hybridization scenario against the “octaploidy hypothesis”. This
makes a neat and tidy way to launch hybridization as a new hypothesis, but it has in fact been
discussed previously. What’s exciting about this paper, is that it adds evidence to this ongoing
discussion.

We were aware of previous discussions of allo-polyploidization in previous papers, but we didn’t
cite those papers in the initially submitted manuscript. We added citations in the main text.

**###S71:** Page 12, lines 2-4: This sentence highlights an issue with this whole section of the
discussion: suddenly the authors are describing the proto-gnathostome genome rather than the
proto-vertebrate genome... Do they mean to say that only 2R, and not 1R, was an
allopolyploidization event? Why not 1R? This is especially confusing since the authors started the
section talking about the proto-cyclostome genome and hexaploidization. It should be **abundantly**
clear which tetraploidization events they are referring to.

We started the paragraph by mentioning polyploidization events in early vertebrate lineages
including the proto-cyclostome and proto-gnathostome. We first mention the cyclostome-specific
whole-genome triplication and then we move on to 2R.

**###S72:** Page 12, line 2: I would change “shows” to “suggests”.

Revised as suggested.

**###S73:** Page 12, lines 10-11: I suggest “... throughout most gnathostomes, **[comma]** including
cartilaginous fishes, but are **missing** in invertebrates, **[comma]** including the closest relatives **of**
**vertebrates**, such as **tunicates** and amphioxus.”

Revised as suggested.

**###S74:** Page 12, line 13: Add a comma after “events”.

Revised as suggested.

**###S75:** Page 12; lines 30-31: It's not clear whether MHC, NKC and LRC were located on **different**
microchromosomes or the same microchromosome. The authors write about cis-preserved genes
on the next page (line 2), but the context we are in as readers is tetraploidizations, which suggests
different chromosomes... The authors use microchromosomes in plural on page 12, line 31.
In this discussion, we are interested in a possibility of asymmetric contribution from one of the
two subgenomes in the proto-gnathostome genome. We revised Figure 6 to clarify that MHC, NKC
and LRC were located on different microchromosomes in the proto-gnathostome genome. Figure
6 suggests that the precursor of MHC, NKC and LRC might have emerged from one of the two
subgenomes.

**###S76:** Page 12, line 30 - page 13, line 7: The authors have traced the **locations** where there
would be MHC, NKC and LRC genes back to early vertebrate evolution, but are there any
indications that the genes themselves were present at this time? After 1R? After 2R in
gnathostomes?
A recent study discussed the origins of those immune complexes, and argued that those
complexes have emerged through 1R and 2R [Ohta et al., J Immunol, (2019)].

**###S77:** Page 13, lines 9-22: I think this section is overstated. See my comment above regarding
page 5, lines 14-16. The fact that some ohnologs are human disease genes is underwhelming. Of
course they are. There are many more that are **not**. The studies the authors have cited are more
concerned with dosage issues in anciently polyploid genomes such as ours, and that when those
dosages in the re-diploidized genomes are perturbed, by copy-number variations for example,
they may result in disease. This is interesting in terms of genome evolution and the constraints
upon genome structure and evolution, which are revealed when disease arises. In these terms,
there is a connection to the present study, and this study adds to the knowledge about constraints
on genome evolution. But from there it is a big step to say that this study has "implications for
understanding human genetic diseases", which suggests implications for disease origins, disease
progression or even disease treatments. Please restate this section, and the section at the end of
the introduction on page 5, in terms of constraints on genome evolution, rather than by linking it
to human disease.
See our response to Reviewer 1's [Comment09].

**###S78:** Page 13, lines 28-32: Several statements in this concluding section need to be tempered
down a bit. On line 28 - "contentious" is perhaps a bit strong. I suggest "our reconstructions
address several unresolved issues". Regarding "the origin of the adaptive immune system", the
authors have provided a brief and very general discussion about the evolution of the adaptive
immune system. This statement should be understated somewhat. The reference to human
diseases should be left out.
We replaced contentious with important. We don't think it is an overstatement to say that our
reconstruction offers a unique evolutionary perspective to the origin of adaptive immune system.
See Comment S75 above to clarify the confusion by Reviewer 1.

**###S79:** Figure 1: Most of the figure caption is not relevant for the graphical interpretation of the
figure. If the results or the methodology are not described well enough in the main text, change
the main text instead of adding this much information to the figure caption. For example, the
whole section between lines 2-8 should be removed (“We reconstructed the...”).
We moved the text (“We reconstructed the ...”) to the main text.

The final sentence of the legend also does not belong here.
We moved the sentence to the figure title and the main text.

The caption can be shortened further by changing to “The *Trichoplax* and **elephant shark** scaffolds
were sorted...” to avoid repetition.
Revised as suggested.

As for the figure itself, it would be useful if the 17+PvcUn chromosomes were enumerated in the
y-axis.
We added chromosome labels on the y-axis.

**###S80:** Figure 2: It should be clear that the figure shows examples and not the full data. Again,
there is some confusion of terms between scaffolds, segments, subgroups and chromosomes. I
suggest the following to perhaps clarify this - “Japanese lamprey scaffolds (a) were correlated with
proto-vertebrate chromosomes (Pvc). Scaffolds corresponding to Pvc3 are shown in blue and to
Pvc17 are shown in green. Segments of conserved synteny from the lamprey scaffolds were
clustered into proto-cyclostome chromosomes (b) based on the distribution of paralogs vs.
orthologs. The triangular plot (c) is a 45-degree-rotated graph of the paralog distribution **between**
the 12 proto-cyclostome chromosomes that correspond to Pvc3 and Pvc17. This shows...”
We did mention that our reconstruction is presented in Figure 3 (Fig. 2 in the initially submitted
manuscript) partly. The text was revised as follows: “(a) Japanese lamprey scaffolds are illustrated
with the scaffold IDs. These scaffolds were partitioned into segments of conserved synteny, and
segments corresponding to proto-vertebrate chromosome Pvc3 (blue) and Pvc17 (green) are
shown for illustrative purposes. (b) Groups of segments of the same color were organized into
several subgroups representing proto-cyclostome chromosomes based on the distribution of
paralogs and orthologs. (c) The triangular plot is a 45-degree-rotated graph of the paralog
distribution between the 12 proto-cyclostome chromosomes that correspond to Pvc3 and Pvc17. ”

The description of the multiplicity table is too long, and most of it is not relevant for the graphical
interpretation of the figure. The figure caption is already too long.
We deleted two sentences.

**###S81:** Figure 3: There is too much description of results and discussion in the figure caption that
is not necessary for the graphical interpretation of the figure. The whole section starting “The
segment lengths are longer in human...” and ending “... and the large macrochromosomes” does
not belong in a figure caption. The same is true for “In general, smaller proto-gnathostome
chromosomes [...] and large chromosomes with low gene densities” and “As in the gene density

plot [...] with high ohnolog densities.”

We moved the texts to the main text.

There is also some confusion between “segment length” and “chromosome size” for this figure.

The definition of “segment” should be abundantly clear in the main text as well as the figure

caption.

Reconstructed chromosomes consist of multiple segments and the chromosome size is the total

segment length. We revised the text as follows: “Each proto-gnathostome chromosome, consisting

of multiple segments, was mapped to modern genomes, and the total segment length in the

human genome is shown on the x-axis, whereas the total segment length in the chicken, spotted

gar and elephant shark genomes are shown on the y-axis.”

**###S82:** Figure 4: I don’t think the authors should include PvcUn in the evolutionary scenario, nor

mention 18 (rather than 17) ancestral chromosomes in the caption. PvcUn is a construction of

many small sections with weakly conserved syntenies that likely “belong” in other chromosomes.

It’s a “waste basket” construction, if I’ve understood their methods correctly. The inclusion in the

evolutionary schematic gives the wrong impression that it represents a pair of ancestral

chromosomes. The grey areas that correspond to PvcUn can be left in the images of the modern

genomes, if it’s clearly described in the caption that the grey color corresponds to PvcUn regions.

We decided to present PvcUn as one of 18 proto-vertebrate chromosomes, because (1) it is the

output of our reconstruction method; (2) it has macrosynteny conservation in the scallop genome;

and (3) there is one-to-one correspondence with a chromosome reconstructed by Sacerdot et al.

Please see our response to Reviewer 1’s [Comment 04].

How strong are the conserved syntenies that indicate that elephant shark scaffold 25 and chicken

chromosome 24 are derived from PvcUn? If it’s only a handful of genes, I would at the very least

mark these as striped and not completely filled in with grey color.

See Figure S7.

**###S83:** Figure 4: The authors have not included any rearrangements or drawn lines between the

proto-cyclostome chromosomes and the extant lamprey chromosomes.

It would make the figure too complicated.

It’s difficult to see the evidence of the hexaploidization in the lamprey genomes otherwise.

The evidence is presented as Figure. 3. Illustration of all lamprey scaffolds does not indicate paleo-

hexaploidization by itself.

**If the reader doesn’t have any sort of Then why include the lampreys at all?**

It visually shows rearrangements in the lamprey lineages. This is important because conclusions in

previous studies (including the 1R-plus-segmental-duplication model) might have been affected by

such lineage specific rearrangements, and this is why we need the proto-cyclostome

reconstruction to conclusively resolve the contentious issues in the early vertebrate genome

evolution.

**###S84:** Figure 4: The caption suggests that all macrochromosomes in extant gnathostomes
resulted from the chromosome fusions that preceded 2R, and that all chromosomes that didn't
fuse resulted in microchromosomes. How can this be?
This is a misunderstanding. We mean that a pair of fusion chromosomes became a
macrochromosome and a microchromosome by biased fractionation. We revised Figure 6 (Fig. 4 in
the initially submitted manuscript) to clarify which chromosomes belong to which subgenome.

In this figure alone I can see that, for example, chromosome 14 in humans, arguably a
macrochromosome, is derived mostly from a Pvc17-derived proto-chromosome, which did not
experience any fusions. Even if all macrochromosomes are derived from ancestral chromosome
fusions, surely not all fusions occurred at the base of vertebrates?
We apologize the lack of sufficient description of the graphical interpretation of the figure.

**Detailed comments on Supplementary Information:**

**###S85:** Page 4, line 5: What was the origin of this elephant shark? The geographic area where it
was caught, but also the conditions by which it was caught. The elephant shark is classified as a
"Least Concern" species by the IUCN (<https://www.iucnredlist.org/species/41743/68610951>), but
it occurs within protected areas, and there are conservation plans in place across its entire
geographical range, so this information is important. This information also provides additional
assurance that the right species has been used.

The adult elephant shark was collected by the senior author in Hobart, Tasmania, Australia where
this species is captured regularly on a commercial scale. Annually up to 114 tons of elephant shark
capture is permitted (<https://www.afma.gov.au/fisheries-management/species/elephant-fish>) in
Australia (and a comparable quantity is caught in New Zealand). If you order Fish & Chips in
Hobart, the chances are you will be eating elephant shark (sold as elephant fish or white fish) with
the chips. There are only three species of *Callorhinchus* in the world, with one species found each
in Australia/New Zealand, Africa and South America. Therefore, there is no confusion regarding
the identity of the species. We had mentioned the source of the elephant shark in our 2007 *PLoS*
*Biol* paper (Venkatesh et al., 5: e101) and have used DNA from the same individual for all our
publications so far, including the present paper. We have now mentioned the source of the
elephant shark in the Supplementary Information.

**###S86:** Page 11, line 5: The same as above for the Arctic lamprey. How was this animal procured
and from which geographic range? In America, the Arctic lamprey could co-occur with the closely
related Alaskan brook lamprey (*Lethenteron alaskense*), and in Asia it co-occurs with the
FarEastern brook lamprey (*Lethenteron reissneri*). The Siberian brook lamprey (*Lethenteron*
*kessleri*) is sometimes classified as a sub-species of the Arctic lamprey.

The Japanese lamprey (aka Arctic lamprey) was collected by the senior author from the Ishikari
River, Hokkaido, Japan during the breeding season. In the course of genome sequencing, we have
also determined the complete mitochondrial genome sequence and it shows 99.78% identity to
the mitochondrial genome of *Lethenteron camtschaticum* in GenBank (accession number

KF701113.1). Thus, there is no ambiguity regarding the identity of the species. We have now
mentioned the source of the Japanese lamprey in the Supplementary Information.

**###S87:** Page 12, line 5: How does this genome size compared with the previously publishes
genome assembly of the Arctic lamprey? And of the latest assembly of the sea lamprey?

The genome size of the Arctic lamprey has not been previously estimated. The previously
published genome assembly of the Arctic lamprey (Mehta et al., PNAS 2013) spanned 1.03 Gb.
The closest species with an estimated genome size is *Lethenteron appendix* (1.4 pg; see Animal
Genome Size Database). The estimate genome size of *Lampetra fluviatilis* is 1.4 pg. These values
are close to our estimated genome size of the Japanese lamprey (1.43 Gb). The estimated genome
size of the sea lamprey is 2.3 Gb (Smith et al. Nature Genetics, 2013) and its latest published
assembly measures 1.1 Gb.

**###S88:** Page 14, lines 21-24: Were these TRINITY transcriptome assemblies from the same
individual as the genome assembly? It's not clear whether these transcriptome efforts were part
of the same genome project described in this paper. This should be made clear in the text. The
Institute of Molecular and Cell Biology at A*STAR is cited as the source of the RNA-Seq reads in the
BioProjects database, which is the home institute of several of the authors.

No, not all TRINITY transcriptome were from the same individual. The transcript assemblies were
generated previously in the senior author's laboratory as part of other projects (Parahox gene
family, Nav channel genes, etc.). We have now specified accession numbers for all the mentioned
tissues.

**###S89:** Page 18, lines 1-9: The methods described in this paragraph are not entirely clear. For
example, "We obtained orthologs and paralogs from gnathostome species..." What does this entail
specifically? What kind of dataset was obtained from Ensembl? Sequences? Spreadsheets with
annotation IDs and locations etc?

Protein-coding sequences, their positional information, Ensembl Compara gene trees and
alignments were obtained from Ensembl.

How were these obtained from gene trees? Usually Ensembl datasets are obtained through
BioMart. Was the complete set of gene trees in Ensembl 75 downloaded?

All trees for protein-coding genes (Compara.75.protein.nhx.emf.gz) were downloaded.

If so, this dataset must have included much more data than only phylogenetic data. For example, it
must have included some of the annotation data created by Ensembl, because the authors
mention that they looked at whether gene duplicated were annotated as Vertebrata, Euteleostomi
or Clupeocephala.

Yes.

Were the trees simply analyzed visually on the Ensembl website? This would be a monumental

task.

We check gene trees visually/manually on the Ensembl website only when we are interested in
specific genes. (We don't describe manual browsing as analysis.)

If only some Ensembl gene trees were analyzed, how were they selected for analysis.

We did not write that only some trees were analyzed. We analyzed all protein-coding gene trees.

How was the tree data analyzed specifically?

We processed the NHX format trees (Compara.75.protein.nhx.emf.gz).

The authors write, for example, that small-scale duplicates were discarded. What does this entail
specifically?

Paralogs were identified as follows. First, a gene pair was retained if their duplication node (i.e.
their divergence point) was annotated as Vertebrata, Euteleostomi, or Clupeocephala. Second, if
at least one of the two paralogous genes experience additional "small-scale" (as distinct from
whole genome) duplications, such as mammalian-specific duplications, the pair was discarded. We
want to discard genes affected by small-scale duplication events, because we cannot tell the
original gene position if the gene was affected by segmental duplication etc.

What did their final dataset consist of? What kind of data?

Please see Supplementary Data 1.

So much of the final evolutionary scenario hinges on these analyses, but I haven't been able to
scrutinize it to the level I would like to because I don't find the information. For example, the
analyses hinge on identifying whether gene duplicates are paralogs, but I can't see how the
authors have identified that two genes are duplicates to begin with. How did they positively
identify duplicates, specifically.

If two genes in a tree come from the same species, they are duplicates. We chose duplicates that
were annotated in Ensembl as Vertebrata, Euteleostomi, or Clupeocephala, because we are
interested in WGDs. We note that it is a misunderstanding that our entire analysis hinges on the
accurate annotation of paralogs. We developed a probabilistic macrosynteny model so that the
reconstruction is not affected too much from random annotation errors, local segmentation
errors, etc. (see [Nakatani and McLysaght, *Bioinformatics* (2017)] for more details). Please also see
our response to Reviewer 1's [Comment 08].

In general, it would be valuable if the authors described exactly how many orthologs vs. paralogs
they identified and included in their dataset. I would also urge the authors to share these datasets
either as a supplementary file with the publication or in an online repository, if possible. Unless
this data includes tens or even hundreds of thousands of genes, then I would understand it is not
feasible. However, it would be especially relevant for the elephant shark reciprocal BLASTP
searches described on page 18, lines 7-9, because it would be important to know how many
orthologs they identified, and as a reader I would like to review this list to make sure that the

orthology assignments were (mostly) correct. This also goes for the amphioxus/human and
lamprey/gnathostome ortholog searches described further down on the page. If it's not feasible to
share the resulting datasets, at least describing the searches in more detail would help give the
reader an indication of what the results were like.

Please see Supplementary Data 1.

Because, in addition, it is not clear against which datasets/databases the BLASTP-searches
described on this page were done. For example, "We performed BLASP search[es] for all species
pairs, and identified orthologs and paralogs..." What species pairs? Which gene dataset was used
as queries and which datasets/databases were searched?

We revised the text as follows: "We performed BLASTP searches for all species pairs (with
vertebrate genes as query sequences and invertebrate genes as subject sequences), and identified
orthologues and paralogues ..."

I understand the logic of simply using the top 2 or 4 scoring genes for the BLASP searches, but
there is a large potential for mis-matches. I would like at least the possibility to quickly scan the
resulting orthology/paralogy assignments to verify, or at the very least know which datasets were
used as queries and which ones were searched in order to ensure reproducibility.

We submit the dataset as Supplementary Data 1.

**###S90:** Page 18, line 29: What search were these bit-scores derived from. Describe the procedure
clearly.

They were derived from an all-vs-all BLASTP search among the Japanese lamprey protein
sequences.

**###S91:** Page 18, line 29: All three conditions or only 1 or 2 of them? It's not clear.

It's clear. When only one or two conditions are satisfied, we don't say three conditions are
satisfied.

**###S92:** Page 18, line 30: Describe that lamprey vs. amphioxus BLASTP searches were done earlier
in this section. Does this refer to the same BLASTP search as the lamprey gene pair bit-scores in
the preceding line? The following line also seems to refer to BLAST-searches against sea lamprey
genes...?

BLASTP searches were performed between all species pairs, as described in Section 2.1. We added
a subsection describing orthologues between the sea lamprey genes and the Japanese lamprey
genes: "We performed a BLASTP search between the sea lamprey genes (query) and the Japanese
lamprey genes (subject) and defined reciprocal best hits as orthologues."

**###S93:** Page 18, line 26 - page 19, line 8: This section describes the annotation of lamprey paralog
genes. It is logical that the authors would consider paralogous gene pairs in lamprey, as described
on page 18, lines 19-29. But it is not clear from this section, nor from the main text of the paper,
how paralogous gene **pairs** helped identify **hexaploidization** in cyclostomes.

See Figure 3 and Supplementary Information Section 3.3.

I understand that the **distribution** of gene pairs across three ancestral chromosome pairs would
still indicate hexaploidization, but if this was the authors' thinking, it should be better described.
Three paralogous proto-cyclostome chromosomes are not enough as evidence of hexaploidization.
We did not write that three paralogous proto-cyclostome chromosomes are evidence of
hexaploidization because we did not think in that way.

The information I miss from this section is whether any gene **triplets** were identified, and if so,
how many?

Gene triplets are included, but the number of paralogous genes in a gene family is not a good
indicator of the number of WGD events. That's why we need reconstructions.

**###S94:** Page 19, lines 2-5. I don't understand this reasoning at all. Please clarify. It is not clear
what "the pair" are, or what "either of the lamprey genes" refers to.

A pair of Japanese lamprey genes were defined to be paralogs if "the pair" satisfies Conditions 1, 2
and 3. We revised "the pair" to "the gene pair".

Remove the parenthesis around "We retained seven paralogs..."

Revised as suggested.

Also, clarify that the expectation of three rounds of WGD (1R, 2R and a cyclostome-specific WGD)
is the hypothesis that they were working with based on the previous suggestion in Mehta et al.
(2013). It's important to highlight this because the actual scenario that this study resulted in is
different! One WGD (1R) and one hexaploidization! The maximum expected number of paralogs
after 1R and then a cyclostome-specific hexaploidization would be $1 \times 2 \times 3 = 6$? At first I was
confused because I thought the authors were referring to the latter, not the initial hypothesis.
Why 7 though, and not 8?

We retained seven matches because we excluded self matches. We revised the text and wrote
that seven paralogs were expected if we assume three rounds of WGD. We retained eight paralogs
because if we retained only five paralogs for each gene, then readers might think that the analysis
is biased and hexaploidization is an artefact of the assumed number of paralogs. In fact, allowing
slightly larger numbers of paralogs result in only additional random noise, which do not have much
influence on our reconstruction. The reconstruction algorithm is tolerant to random noise if there
are stronger signals.

**###S95:** Page 19, lines 13-15: This section is similarly confusing. What does "the elephant shark
gene pair" and "neither of the elephant shark genes" refer to?

We replaced "the" with "an".

**###S96:** Page 21, line 3 (below the algorithm): I suggest "**the** proto-vertebrate genome".

Revised as suggested.

**###S97:** Page 21, line 4: Clarify **which** lamprey genome.

We replaced "genome" with "genomes".

**###S98:** Page 21, line 4: When the authors write simply “comparing the lamprey genomes with
each other and also with four gnathostome genomes...” it reads like they are not explaining
further what these comparisons entail. It is not immediately clear that they are referring to the
sections that follow (3.2.1, 3.2.2 etc). Please clarify.

We added “These steps are described below.”

**###S99:** Page 23, line 2-2: “The reconstruction with $K = 18$ was the most significant.” Could the
authors please share the full results of this? What was the significance **value** of $K = 18$? What
values did other K s produce?

The table shows the significance for $K=10, \dots, 20$. We added this table in Supplementary
Information.

K	$\log(\mathbb{P}(X \geq x))$
10	-8363.48
11	-9003.43
12	-9438.88
13	-9705.24
14	-9958.95
15	-10079.3
16	-10182.1
17	-10508.8
18	-10767.3
19	-10371.5
20	-10249.1

**###S100:** Page 23, line 14-15: “Syntenic to” does not mean what the authors mean here. Syntenic
means that two genes are located on the same chromosome. I suggest “**A comparison of**
**conserved synteny between these proto-vertebrate chromosomes and the scallop genome**
**shows that Pvc17, PvcUn, Pvc8, and Pvc9 correspond to individual scallop chromosomes -**
**chromosomes 3, 13, 6 and 4** respectively.”

Revised as suggested.

**###S101:** Page 23, line 18: It’s not clear what the authors mean by “in early invertebrate lineages”.
Early invertebrates as in at the base of the metazoan lineage (this is very very early), or early as in
already in an invertebrate ancestor or extant chordates/vertebrates.

The common ancestor of vertebrates and scallop, and also *Trichoplax* to some extent.

**###S102:** Page 23, line 18: I’m still not certain that PvcUn actually represents an ancestral
chromosome. Clearly, there is not perfect correspondence between the proto-vertebrate genome
reconstruction and the scallop genome, as shown in Figure 4. Because the conserved synteny

comparison was one-sided, i.e. proto-vertebrate → scallop, it's not possible to differentiate
between rearrangements in the proto-vertebrate or rearrangements in the lineage leading to the
scallop. Doing the analysis the other way, scallop → proto-vertebrate, might show that parts of
scallop chromosome 13 correspond to other Pvc's.

Reviewer 1 appears to mistakenly believe that the analysis is one-sided. If we used one-to-one
reciprocal best hits between the scallop genes and the Japanese lamprey genes, then we might
have missed some synteny blocks. In our analysis, however, we identified one-to-multiple co-
orthologs as described in Supplementary Information Section 2.

So for a large number of segments of weak synteny conservation (i.e. PvcUn) to show conserved
synteny with a single scallop chromosome is not definitive evidence. Did all the segments of PvcUn
correspond to scallop chromosome 13, or were there segments in PvcUn that could not be
assigned? The authors have not described this. Also, they haven't described how big the conserved
synteny segments that make up PvcUn are. I suspect they are very small, which makes any
conclusions very tentative.

As we wrote, the comparison with the previous reconstruction by Sacerdot et al. also supports the
presence of PvcUn (now referred to as Pvc18). One segment has many orthologs on scallop chr13,
but the remaining segments have unclear synteny in the scallop genome.

**###S103:** Page 23: It is notable that the authors haven't discussed here why these results are so
different from the previous reconstruction of the vertebrate genome by the first author (Nakatani
et al. *Genome Res.* 17(9), 2007), which reconstructed only 10 ancestral chromosomes. Which
scenario is wrong? Is this completely due to the inclusion of a cyclostome in the reconstruction?
Putnam et al. (2008) didn't include lamprey synteny and still arrived at 17 ancestral (chordate)
chromosomes. I have to ask, also, for the Nakatani et al. (2007) ancestral chromosomes to be
included in Table S8. This would be very useful.

The reconstruction by Nakatani et al. (2007) consisted of 10 to 13 chromosomes depending on
rearrangement events occurring between 1R and 2R. We wrote in the initially submitted
manuscript that many segments were assigned to chrUn in [Nakatani et al., *Genome Res* (2007)],
and we found more chromosome fusion events between 1R and 2R than in [Nakatani et al.,
*Genome Res* (2007)]. Putnam et al. did not describe why they chose specific threshold values in
their analysis (although different threshold values result in different numbers of chromosomes).
Genome sequence and annotation versions are different so comparison is not straightforward.

**###S104:** Page 24, line 7: It can't hurt to add the binomial nomenclature for the silkworm and sea
anemone as well.

Revised as suggested.

**###S105:** Page 24, lines 14-15: It is not clear what "assigned scaffolds to **the chromosome** with the
largest number of markers" refers to. The proto-vertebrate chromosomes?

The freshwater snail scaffolds were assigned to snail chromosomes. We added "then".

**###S106:** Page 24, lines 18-19: I'm not so sure. This suggests that the patterns of synteny are
conserved, it say nothing of chromosomes themselves. For example, it does not consider
chromosome fissions preceding the time point of the proto-vertebrate reconstruction.
Fissions and fusions result in different ortholog distributions. See for example [Jaillon et al., *Nature*
(2004); Nakatani et al., *Genome Res* (2007); Nakatani and McLysaght, *Bioinformatics* (2017)].

What I see in Fig. S3 is that **these particular** conserved synteny patterns, inferred to have existed
in early vertebrate evolution, can be “recreated” **to some extent**, by no means perfectly, in
invertebrate genomes as well. However, genomes are mixes of different patterns, syntenies and
paralogies of different origins, and this study does not address other patterns that may exist in the
invertebrate genomes that may indicate other ancestral chromosome configurations. The analyses
in these studies were done in only one direction, proto-vertebrate → invertebrates. Starting with
another lineage at the outset may reveal other chromosome configurations in the common
ancestor.

We don't think the direction from proto-vertebrate to invertebrates affects the conclusion. Please
see previous papers for similar discussions about macrosynteny conservation [Jaillon et al., *Nature*
(2004); Nakatani et al., *Genome Res* (2007); Putnam, et al., *Nature* (2008), etc.].

**###S107:** Page 25, line 4: Change to “**have** remained contentious”.
Revised as suggested.

**###S108:** Page 25, line 5: Change to “**the** possibility of cyclostome-specific WGD...” I also suggest
removing “intense”, as this is a value judgment.
Deleted.

**###S109:** Page 25, line 8: Change to “... WGD, **followed by the** loss of two entire clusters”.
Revised as “WGD followed by the loss of two entire clusters.”

**###S110:** Page 25, line 10: Change to “We considered that **a** reconstruction of **the** proto-
cyclostome chromosomes...”
Revised as suggested.

**###S111:** Page 25, line 12. Change “comprises” to “comprise”.
Why?

**###S112:** Page 25, lines 14-15: Change to “Thus, **the** reconstruction...”
Revised as suggested.

**###S113:** Page 25, line 17: Change to “**The** enumeration...”
Revised as suggested.

**###S114:** Page 28, line 3: Change to “**in** the proto-vertebrate lineage...”
Revised as suggested.

**###S115:** Page 28, lines 2-6: Perhaps this is unrelated, but does it then follow that for the proto-
cyclostome reconstruction the most significant partition was $6 = 1R$ followed by hexaploidization?
It is related. Please also see Figure 3 for the discussion of hexaploidization.

**###S116:** Pages 29-30: The “red/black/white/grey” metaphor is quite long-winded and very
difficult to follow. Please break up and clarify.

We revised the text as follows: “Figure 4a illustrates the case of a chromosome fusion occurring
between the two WGD events. As the result of the fusion, the grey post-2R chromosomes share
large numbers of ohnologs with the black and white chromosomes (represented by red regions in
Figure 4c); on the other hand, there are no ohnologs between black and white chromosomes
(white regions). In addition to the case of a chromosome fusion between the two WGD events, our
reconstruction method considered other rearrangement scenarios: namely, (A) a chromosome
fission event occurring in the period between 1R and 2R and (B) a fusion or translocation after 2R.
Scenario A results in the same paralog distribution pattern as in the case of a fusion between the
two WGD events, but the two scenarios can be distinguished by checking the ortholog distribution
in invertebrate genomes. In Scenario B, the paralog distribution is different from the scenario of a
fusion between 1R and 2R. In general, we expect to see a large number of paralogs between a pair
of proto-gnathostome chromosomes, only if the two chromosomes (1) are duplicated
chromosomes or (2) inherit duplicated chromosomes or duplicated segments through
rearrangements (fusions, fissions and translocations). These proto-gnathostome chromosome
pairs are called ‘red chromosome pairs’ (as in Fig. 4c) in the subsequent texts.”

**###S117:** Page 31, lines 1-2: Please clarify that the “previous reconstruction” has the same first
author as this study. Otherwise we might get the impression that Dr. Nakatani is (unfairly)
disowning his previous work.

The reason we wrote “previous reconstruction” is because it is not appropriate to write “our
previous reconstruction” nor “their previous reconstruction”. We feel that it is clear from the
citation information that the first author is the same person.

**###S118:** Page 31, line 5: Regarding the “nine large-scale rearrangements”, I counted nine fusions.
How about fissions?

We didn’t find any fissions in the proto-gnathostome genome.

**###S119:** Page 31, line 26: Change “fission” to either “**the** fission” or “fissions”.

Changed to “fissions”.

**###S120:** Page 32, line 10: Change “chromosomes” to the singular “chromosome” or write “For
each of the proto-gnathostome chromosomes...”

We thank Reviewer 1 for finding this error.

**###S121:** Page 32, lines 22-23: I suggest “These chromosomes underwent **the first** WGD (1R),
[comma] resulting in the **doubling** of the proto-vertebrate **genome**.” Remember that we are
generally talking about the **haploid** genome here. “Doubling” of chromosomes could be

misinterpreted as referring to the diploid genome.
Revised as suggested.

**###S122:** Page 32, line 23: Change “In the gnathostome lineage” to “In the lineage leading to
extant gnathostomes”, see my comment about page 8, lines 13-14, above.
Revised as suggested.

**###S123:** Page 33, lines 6-10: I suggest “**Where** our reconstruction **produced** less than six
chromosomes, the remaining chromosomes **out of the expected six are** shown as hatched bars.
**Where** our reconstruction **produced** more than six chromosomes, the extra chromosomes **are not**
**shown. However,** the extra chromosomes were included in all other figures, **[comma]** including
Figures 1 and 2, although **they are very** small.”
Revised as suggested.

**###S124:** Page 33, line 12: Change “Modern” to “Extant”.
Revised as suggested.

**###S125:** Page 33, lines 15-16: It seems strange to me that so many, and in some cases extensive,
“white regions” can be explained to be only centromeres. Perhaps if including also
pericentromeric areas, which do contain **some** genes. It’s a small point, but in any case, this is only
a conjecture on the authors’ part. In addition, writing “regions excluded from our reconstruction”
makes it sound like the authors excluded these regions **purposely**, which I don’t think was the
case. I suggest writing “Regions of the human genome **shown in white likely correspond to**
**regions poor in genes, such as centromeres and pericentromeric regions.**”
Revised as suggested.

The authors should be careful not to give the false impression that they are showing the complete
chromosomes in their reconstruction (Fig. 4). I don’t see centromeres/pericentromeric regions,
telomeres and other “gene deserts” in the figure. These can be more closely described as
conserved synteny blocks for each of the chromosomes.
The figure presents the correspondence between several extant vertebrate genomes and the
three reconstructed genomes.

**###S126:** Page 33, line 26-29: I suggest “... we plotted paralogs among proto-gnathostome and
proto-cyclostome chromosomes **and classified them** into vertebrate **paralogs** (i.e. duplicated in
the common ancestral vertebrate), **cyclostome-specific paralogs, and gnathostome-specific**
**paralogs** as described below.”
Revised as suggested.

**###S127:** Page 33, lines 30-31 - Page 34: I suggest removing "Paralogs in the proto-gnathostome
genome were represented by human paralogs obtained from BioMart:" and simply starting the
sentence as follows - “**Human** paralogs annotated as Vertebrata **in Ensembl** were classified as
vertebrate paralogs (blue dots), **[comma]** and **human** paralogs annotated as Euteleostomi were
classified as...”
Revised basically as suggested.

**###S128:** Page 34, lines 2-3: I suggest "Figure S9 shows the distribution of vertebrate and
gnathostome-specific paralogs **mapped onto** the **reconstructed** proto-gnathostome genome."
Revised as suggested.

**###S129:** Page 34, line 21 (Step 3): "We deleted irrelevant genes from the tree" - This is a very
reckless formulation. Who decides what is irrelevant? Instead, describe and defend your criteria
clearly and methodically.

Revised as follows: "In order to reduce the computation time, we retained genes from ..., and
deleted the remaining genes from the tree. Then, we inserted the lamprey genes ..."

**###S130:** Page 34, line 26 (last line): Replace "branching pattern" with "tree topology".
Revised as suggested.

**###S131:** Page 35, line 4: Replace "should be clustered" with "would cluster".
Revised as suggested.

**###S132:** Page 35, line 6: Use the plural "annotations".
Revised as suggested.

**###S133:** Page 35, line 20: Replace "the one third of high-GC genes" with "the third of the genes
with the highest GC content".
Revised as suggested.

**###S134:** Page 35, line 25: Make sure that you have described earlier which sea lamprey assembly
you have used for these analyses. Is it the latest germline genome assembly version, or the much
poorer previous assembly? In any case, it doesn't hurt to remind the reader here as well.
We used the germline sea lamprey genome as written in the main text.

**###S135:** Page 35, lines 25-16: I suggest "The annotation of **sea lamprey paralogs was done by**
using RAxML-EPA with **the** WAG matrix (method A), **and** is shown in Figure S13."
Revised as suggested.

**###S136:** Page 35, lines 30-31: The authors refer to the supplementary figures (Fig. S9-S13, and
Fig. S14 on the next page) when they write about Hox genes, yet the Hox genes are not marked
out in these figures. How will the reader verify that this is correct?
It was explained in the figure legend.

**###S137:** Page 35, line 28 - page 36, line 4: It would be helpful if the authors could discuss the
most likely alternative scenario that could explain the same results. Why isn't a shared 1R/2R at
the base of vertebrates followed by independent fissions/segmentations a likely scenario?
Something like this, shared 1R followed by independent chromosome-level segment duplications
and fissions, has been proposed by Jeremiah Smith and co-authors, for instance, based on the
synteny conservation of the latest sea lamprey germline genome. Based on the current results
presented in this papers, why are these alternative scenarios less likely? This is something that I
miss in this paper in general.

This comment is the same as Reviewer 1's [Comment 06], so please see our response to
[Comment 06].

**###S138:** Page 36, lines 20-21: The sentence starting "It was previously shown..." is difficult to
follow. It's not clear what the "branching patterns" of the human genome refers to. It might just
be that a lot of information is packed very densely into this sentence. Please clarify.
The sentence was revised as follows: "It was previously shown that clustered human ohnologs do
not always have the same branching pattern (or duplication timing)."

**###S139:** Page 37, lines 2-3: I suggest "Figure S14 **suggests** that a majority of ohnologs..." It's not
entirely clear how this figure shows sequence divergence. Only panel a in the figure seems to
show this, is that right? Please clarify.
This is a misunderstanding. The triangular plots show the presence of many Vertebrata ohnologs.

**###S140:** Page 37, line 4: The authors write "two out of four" but I can't really see this in the cited
figures. Some guidance would be good. In addition, the figure caption for Fig. S14 mentions "two
out of six" ...
Figure 6 shows chromosome fusions between 1R and 2R, resulting in two fusion chromosomes out
of four chromosomes that were duplicated from a proto-vertebrate chromosome. If we focus on a
chromosome fusion event involving two proto-vertebrate chromosomes, we get two fusion
chromosomes and four non-fusion chromosomes in the proto-gnathostome genome.

**###S141:** Page 44, Figure S3: The y-axis designation "Proto-vertebrate/-cyclostome" is seemingly
contradictory. I understand that these are the Japanese lamprey scaffolds, but it is confusing to
lead with a seemingly contradictory statement. They can't be proto-vertebrate and proto-
cyclostome chromosomes at the same time. I suggest changing the formulation "proto-
vertebrate/proto-cyclostome chromosomes represented by Japanese lamprey scaffolds..." to
simply "The Japanese lamprey scaffolds were compared with invertebrate genomes (x-axes). In
this way we could validate both the proto-vertebrate and proto-cyclostome chromosome
reconstructions. Horizontal orange lines represent the boundaries of Japanese lamprey scaffolds
and black horizontal lines represent the boundaries of the corresponding proto-vertebrate
chromosomes." This should be applied to all the similar figures - Fig 1, Figs. S2, S3, S4, S6, S7 - and
within the figure captions and manuscript text. Name the y- and x- axes for what they actually
show, not what they "represent".
We revised the figure legend as follows: "Japanese lamprey segments that were mapped to the
proto-vertebrate/-cyclostome chromosomes are shown on the y-axis. Black and orange horizontal
lines indicate boundaries of proto-vertebrate chromosomes and proto-cyclostome chromosomes,
respectively." We also changed the y-axis label from "Proto-vertebrate/-cyclostome" to "Proto-
vertebrate chromosomes" and enumerated from Pvc1 to Pvc18. The reconstruction method is
described in detail in Supplementary Information, so there should be no confusions between
Japanese lamprey segments and proto-vertebrate chromosomes.

In addition, I cannot see any horizontal grey lines in the figure - they are mentioned in line 5 of the
figure caption. I also can't see the difference between thick and thin vertical lines - mentioned in

lines 7-8 of the caption.
The horizontal gray lines and vertical thin gray lines were removed just before the initial
submission, but we forgot to edit the figure legend. Those texts were deleted in the revised
manuscript.

**###S142:** Page 44, line 11 (last line of figure caption): See my comment above regarding page 24,
lines 18-19. This shows that the synteny patterns can be recreated to **some extent** in invertebrate
genomes, but it doesn't definitively show that they represent ancestral metazoan chromosomes.
Be careful with this conjecture.
See our comment to Comment S106.

**###S143:** Page 46-47: This figure caption is inordinately long. Please include only information
necessary for the **graphical** interpretation the figure. Everything else should go in the
supplementary information text, if it's not there already. The description of this procedure is very
good, it should be part of the main text, not a caption!
In the revised manuscript, the figure was moved to the main text with a concise description.

**###S144:** Page 48, Fig. S6: It would be very helpful to enumerate Pvc1-17 and PvcUn on the X-axis
of the figure, and the proto-gnathostome chromosomes on the y-axis.
We enumerated Pvc1-18 on the x-axis. The proto-gnathostome chromosomes are not numbered
in our analysis.

The caption of this figure illustrates my comment about alternative scenarios. The authors very
clearly describe their scenario, and highlight the data which illustrate their point very well. But can
they disprove/falsify alternative scenarios? Can this same data illustrate any of the alternative
scenarios? What would the data look like if the most likely alternative scenario were true? Could
the rearrangements not be post-2R or pre-1R fusions? This analysis doesn't differentiate between
1R-generated and 2R-generated paralogs. Help the reader navigate these alternatives.
It is also possible that all chromosome fusion events occurred after 2R, but in that case, we need
to assume that the chromosome fusions occurred non-randomly. Specifically, the proto-
gnathostome chromosomes duplicated from Pvc2 must have been preferentially fused with the
proto-gnathostome chromosomes duplicated from Pvc3. We favor a more parsimonious scenario
in which such chromosome fusions occurred between 1R and 2R.

**###S145:** Page 48, Fig. S6: There are some curiosities in this figure that are not mentioned.
Notably, the orthology between Pvc17 and proto-gnathostome chromosome 9. Wouldn't this
result from a large-scale fission? When did this occur? The authors have not mentioned fissions in
the paper.
We illustrated it in Figure 6e as a post-2R translocation (or chromosome fission followed by a
fusion). It is inferred to have occurred after 2R because only one out of four duplicated proto-
gnathostome chromosomes was affected by this rearrangement.

**###S146:** Page 48, line 3: Correct "axe" to "axis".
Revised as suggested.

**###S147:** Page 49, Fig. S7: The horizontal grey lines are barely visible, even when I zoom in on the
PDF.
This must have happened during the manuscript processing on the journal website. We inserted
vector image figures in our submission and we expect the final images to be high resolution.

**###S148:** Page 49, line 1: “Comparison with the lampreys and amphioxus genomes.” Comparison
of what? Instead of writing “proto-gnathostome” at the y-axis, describe what it actually shows.
Correct “lampreys” to “lamprey”.
The figure title is revised as “Comparison of the proto-gnathostome genome with the lamprey and
amphioxus genomes.” The y-axis shows the proto-gnathostome chromosomes represented by the
human segments.

**###S149:** Page 49, line 7: I can’t tell the difference between thick and thin vertical lines in the
figure.
Again this is possibly caused by journal website processing for review and we suspect that our
vector image figures were converted to raster images.

**###S150:** Page 49, lines 8-9: Explain that the 1:4-orthology between proto-vertebrate and proto-
gnathostome genomes is shown in the amphioxus panel of the figure, if I’ve understood this
correctly.
We inserted “(shown in the amphioxus panel)”.

Perhaps it would also be better to order the panels of the figure in the inverse order.
We thank Reviewer 1 for this suggestion, and we revised the figure as suggested. The figure legend
was also edited accordingly.

In general, it is quite difficult to relate the caption to the figure. Doesn't the two lamprey panels
show that both 1R and 2R occurred **after** the divergence of cyclostomes? It shows the same
relationships as the amphioxus panel. Very tricky to know what to look at.
This figure is not enough for discussing the divergence timing. We need to investigate the
distribution of paralogs (Vertebrata paralogs, gnathostome-specific paralogs and cyclostome-
specific paralogs) as shown in Figs. S9-S14.

**###S151:** Page 49, line 12: None of this numbering is shown in the figure, so it’s very difficult to
know what to look at.
We added x-axis labels (i.e. Pvc1-18 on the x-axis) in the revised figure.

**###S152:** Fig. S9 - Fig. S13: Please describe what the x- and y-axes of these figure represent.
See Figure 4 for the meaning of the x- and y-axes.

**###S153:** Page 56, Fig. S14: I almost gave up trying to interpret this figure. It is incredibly
information-dense and there are seemingly some missing parts? Why are there no triangular plots
for the upper scatterplots? Please write out next to the rectangular scatterplots what they actually
show. For example, I’ve mocked up an image for panel a...

 There are no missing parts. There is no WGD in the amphioxus genome or the proto-vertebrate
 genome, and thus we do not discuss their paralog distributions.

Replace the numbering in orange for the actual chromosome numbers. This was useful for me to
 see the 1:4 and 1:6 relationships between the proto-vertebrate and the proto-gnathostome and
 proto-cyclostome, respectively. For the bottom scatterplot, it would also be clearer to use black
 lines, not orange to mark the boundaries of the proto-cyclostome chromosomes. Because the top
 and bottom scatterplots are so similar, I was expecting that Pvc1 and Pvc17 were also plotted in
 the bottom scatterplot. This would avoid the confusing “bottom and left”, “bottom and right”,
 “bottom six”, “middle two out of six”... give them numbers! I still don’t know what “middle two
 out of six” refers to.

Reviewer 1 was confused because he looked at a wrong panel. We revised the text from “(b,c)” to
 “In panels b and c,” to emphasize the panels, because we found this is the main source of the
 confusion. We also changed “bottom and left” to “bottom” and “top and right” to “top”. We used
 orange lines because we previously got a comment that black lines looked confusing in this figure.

**###S154:** Page 56, line 13-14: Perhaps it would be better to note what the figure **does** show,
 rather than what it **doesn’t** show? I.e. the 1:4 relationship between the proto-vertebrate and
 proto-gnathostome reconstructions, and the 1:6 relationship between the proto-vertebrate and
 the proto-cyclostome reconstructions. To be fair, only panel a shows this undoubtedly, but you
 can argue for panel b and c, which I suspect are the more common occurrences. Also, it would be
 helpful to know what it would look like if indeed there was 1:1 orthology relationship - i.e. what if

the alternative hypothesis is true? Can the data be described with alternative scenarios?
It seems Reviewer 1 misunderstood the meaning of the figure. The figure shows that (1) there is
no clear one-to-one or two-to-three relationship between the four proto-gnathostome
chromosomes and the proto-cyclostome chromosomes; and (2) fusions are not shared with the
proto-cyclostome lineage. Panel a shows the case of no fusions in the proto-gnathostome lineage,
while Panes b and c shows fusions between 1R and 2R. If there was one-to-one orthology
relationship, we should see a non-uniform distribution of ortholog dots. In the revised manuscript
we deleted the discussion of alternative scenarios, since Reviewer 1 commented that the
description is unclear (see Reviewer 1's [Comment S62]). We can also delete this figure if it is so
difficult.

Reviewers' Comments:

Reviewer #2:

Remarks to the Author:

The manuscript "Reconstruction of proto-vertebrate, proto-cyclostome and proto-gnathostome genomes provides new insights into early vertebrate evolution" reports what appear to be two high quality assemblies and uses these assemblies to gain further insights into the history of vertebrate evolution. My previous review of this paper was performed without access to the code that was used to generate analyses or important details on the assembly and analysis pipelines. The author's efforts to share code and edit the manuscript to clarify the methods has made it much easier to review. I do have several comments that I think need to be thoroughly addressed and outline these below.

1) Access to code used in synteny analyses – The authors should make the code publicly available and equally importantly provide user documentation that has become standard practice in code repositories. Given that they mention in their example scripts that components of the code (e.g. variables) are different from the previously published, (but still not public) version it would be nice to see that code released as well along with usage information for the older version and details of what has changed between versions. I spent two afternoons going through the code (other comments below) and I did not note any components of the code that should impinge on copyright. It is my impression that release of well documented code is essential to this paper and essential to ensure reproducibility.

2) Access to code used in genome assembly - The use of HiRise (Dovetail) in the lamprey and shark assembly pipelines raises some issues with reproducibility as that program is maintained as closed source code by Dovetail. As such it will be impossible for anyone to independently replicate the published assembly using the same methods reported in the paper. This may change in the future if the code is released, and I encourage the authors to request its release. If this request is not granted, the authors should make sure to include the software version used for this assembly and all relevant assembly/filtration parameters, as well as .agp (or similar) files that relays mapping evidence and weights that were used in the scaffolding process. Dovetail can provide this.

3) The analyses seem to have been run in three parts that define a specific hypothetical duplication scenario. Why was the reconstruction not performed using all of the data in a single run? What happens if they do this? Is the reconstruction the same or different?

4) The authors state "To distinguish between alternative polyploidization models (i.e. S5–S8 in Fig. 1) we introduced a measure we have called multiplicity" this is clearly not a new idea and they should consider rewording (e.g. Putnam, N.H. et al. 2007). In addition, assessment of multiplicity cannot (as implemented) define the mode of duplication that gave rise to the patterns without explicit statistical tests. I laid out how to perform these tests in the previous review, but these were not performed. From examining the code it appears that they used clustering method seeded with 18 clusters to assign lamprey chromosomes to their ancestral chromosomes. It is therefore even less surprising now that they would observe a peak at 6. Please provide explicit tests of multiple duplication scenarios as laid out in the previous review.

5) I again request that they change "proto-cyclostome" to "proto-petromyzontid". The observation of six *hox* clusters in hagfish may be consistent with their observations, but hardly raises to the level where one might imagine that hexaploidy should be assumed for the entire hagfish genome with certainty.

6) Part of the justification that their "cyclostome" reconstruction is plausible is that there have been other described instances of hexaploidy in vertebrates, despite the obvious issues this raises for obligately sexually reproducing species. However, this assertion seems to be a misinterpretation of those bodies of literature. Shortnose sturgeon have been called "functionally hexaploid" due to

pervasive loss of duplicates (microsatellites) following WGDs, but they are clearly of octaploid origin (Symonová R et al BMC Genet. 2017). Prussian carp hexaploids reproduce only by gynogenesis and are sexual parasites on diploid and tetraploid populations. Justifying their model from a biological standpoint will require substantial alteration of the current discussion and should address whether they are invoking gynogenesis in the origin of cyclostomes/petromyzontids and how this might have transitioned back to a stably meiotic lineage.

7) The authors state "Although several recent studies supported the ancient origin of microchromosomes, it was still unknown (1) if chromosomal features characteristic to modern avian microchromosomes (i.e. high GC content, high gene density and high recombination rate) were already present in the ancestral gnathostome genome" although this seems to not to acknowledge analyses of the spotted gar genome that resolved many of these feature for the ancestral euteolostome, which is only ~40 million years divergent from the ancestral gnathostome they are reconstructing (Braasch, et al. Nat Genet, 2016). Additionally, the spotted gar genome paper is not cited at all despite use of the assembly for their reconstructions.

8) The authors state "In order to verify the timing of the gnathostome-cyclostome divergence with respect to 1R and 2R, we performed an analysis based off Ensembl gene trees on the reconstructed chromosomes ..." These analyses should be re-done from scratch as ENSEMBL trees are forced to a pre-defined topology (with lamprey splits specifically designated as basal gnathostome splits) and are therefore cannot be directly used to perform the tests. This is laid out in (Smith et al. Nat Genet, 2013), but has been notably been erroneously used by others to test similar ideas to those presented in this manuscript.

9) The authors should present, in the main manuscript, more detail regarding the numbers and distribution of ohnologs (and other duplicates) across presumptive paralogous segments (particularly those in Figure 6d/e/g). This is essential to evaluating evidence favoring duplication vs fission in the origin of these segments, which is in turn essential for evaluating evidence as it relates to proposed duplication scenarios (both gnathostome and cyclostome). The authors should be able to gain some inspiration as to how to do this by looking at another paper that is generally similar to this one (Simakov, et al. Nat Ecol Evol 2020: Fig 3, 4b) and even improve upon that presentation. I requested something similar in the previous review (and prior to seeing the Simakov paper) but this request was not satisfactorily addressed, and the revision makes the need for this even clearer.

10) Given that the Simkakov paper was released after the initial submission of this manuscript it may be unfair to require that the authors consider the specific models proposed by that paper, but I am certain readers will welcome it and perhaps expect it. Details of these reconstructions differ in profound ways.

11) Related to the above comments, please also provide numbers of orthologs that support each of the conserved segments in Figure 6E and 6G (after addressing other points). Presentation in the main manuscript will provide essential detail to the reader.

11) Related to point 9, and with apologies for the length of this comment, it appears that the program used for these analyses makes a statistical faux pas in assessing evidence that that two ancestrally linked segments are derived from fission vs. duplication. If I am interpreting the code correctly, the authors use a statistically appropriate test to identify segments that have an excess of shared homologs or ohnologs relative to random. Many of the other studies mentioned in the manuscript and above others have used similar approaches to although the use of the hypergeometric distribution for these tests is laudable. However, it appears that the ohnolog statistic is compared to the ortholog statistic as part of the assessment of whether a segment is likely to be derived from duplication vs fission which does not really shed light on the question at hand if this is true, and appears to not be an appropriate use of these values. Though admittedly the code here is a bit hard to follow given the layout and the supplement seems not to clearly address this. Issues with p-value/ test probability

comparison may not be immediately obvious to the casual observer, but were pointed out by Fisher and subsequently by many others due to pervasive misuse (a couple of modern examples: <https://www.ncbi.nlm.nih.gov/pmc/articles/PMC5804470/> , <https://www.tandfonline.com/doi/abs/10.1080/00031305.1996.10474380>).

A more appropriate approach might be to compare the observed frequency of duplicates on presumptive paralogous segments to the distribution of similarly-aged duplicates within conserved segments. It is necessary to infer that some duplications will be present between fissioned segments because, 1) intrachromosomal duplication is known to occur frequently, even within the human population, and 2) intrachromosomal rearrangements have effectively randomized gene orders over the timescales that are analyzed here; one would expect that two pieces of a fissioned chromosome will carry paralogs that are derived from ancient intrachromosomal duplications (i.e. not WGDs). Therefore, one should be able specifically test whether the presumptive WGD paralogous segments carry more duplicates than would be expected for the average fission event. A test like this is seemingly critical given the definitive statements that are made throughout the manuscript. Perhaps also clearly state in the methods how duplication are differentiated from fissions, especially if I have made some error in interpretation here.

12) The supplemental movie seems to show progress in defining clusters in the cyclostome-centric analysis?? But does not really seem to shed much light into the inner workings of the programs they use.

Given these large issues I will withhold comment on other specific details (e.g. discussions of immunology, discussion of ancient hybridization – or alternately incomplete lineage sorting - in the supplement) for the moment since many details could change depending on how these above comments are addressed. I am certain that all of these requests can be addressed with statistical rigor and in a way that facilitates reproducibility. I hope that the comments above make that easier.

Sincerely,

Jeramiah Smith

Reviewer #3:

Remarks to the Author:

I wanted to congratulate the authors for their efforts addressing the extensive comments from all the reviewers. I sincerely think that a manuscript that was already great has improved a lot.

We thank the two reviewers for their time and valuable constructive comments about our
manuscript. We provide a point-by-point response to the reviewers' comments below and describe
the additional work and changes made to the manuscript. We have labeled Reviewer 2's comments
as [R2 Comment 01], which stands for review round two, comment number one, to distinguish from
the comments in the previous round of peer-review. Previous review comments are mentioned as
Comment 14, etc. References are listed at the end of the file.

8 **Reviewer #2 (Comments to the Authors)**

The manuscript "Reconstruction of proto-vertebrate, proto-cyclostome and proto-gnathostome
genomes provides new insights into early vertebrate evolution" reports what appear to be two high
quality assemblies and uses these assemblies to gain further insights into the history of vertebrate
evolution. My previous review of this paper was performed without access to the code that was used
to generate analyses or important details on the assembly and analysis pipelines. The author's efforts
to share code and edit the manuscript to clarify the methods has made it much easier to review. I do
have several comments that I think need to be thoroughly addressed and outline these below.

**[R2 Comment 01]** 1) Access to code used in synteny analyses – The authors should make the code
publicly available and equally importantly provide user documentation that has become standard
practice in code repositories. Given that they mention in their example scripts that components of the
code (e.g. variables) are different from the previously published, (but still not public) version it
would be nice to see that code released as well along with usage information for the older version
and details of what has changed between versions.

There are no older versions and the code is the same as the one used in our previous publication
[Nakatani and McLysaght, *Bioinformatics* (2017)]. The comment in our user documentation
(README.txt) actually means that the symbols used in the program are different from the symbols
written in the previously published paper [Nakatani and McLysaght, *Bioinformatics* (2017)].

I spent two afternoons going through the code (other comments below) and I did not note any
components of the code that should impinge on copyright. It is my impression that release of well
documented code is essential to this paper and essential to ensure reproducibility.

We have edited the code document extensively in order to make it publicly accessible; however
copyright issues have not been cleared completely (see
<http://numerical.recipes/licenses/redistribute.html>). To ensure reproducibility, we have indicated that
the programs are available upon request to the authors.

**[R2 Comment 02]** 2) Access to code used in genome assembly - The use of HiRise (Dovetail) in the

lamprey and shark assembly pipelines raises some issues with reproducibility as that program is
maintained as closed source code by Dovetail. As such it will be impossible for anyone to
independently replicate the published assembly using the same methods reported in the paper. This
may change in the future if the code is released, and I encourage the authors to request its release. If
this request is not granted, the authors should make sure to include the software version used for this
assembly and all relevant assembly/filtration parameters, as well as .agp (or similar) files that relays
mapping evidence and weights that were used in the scaffolding process. Dovetail can provide this.

As suggested by the reviewer we contacted Dovetail and obtained the software version used for the
HiRise assemblies: version v2.1.3-5ce4af34ac25 for the elephant shark genome assembly and
version v2.1.2-ad17ecf8bf57 for the Japanese lamprey genome assembly. This is now mentioned in
the manuscript (Supplementary Information line 97 and 250, respectively). Dovetail has also informed
49 us “Unfortunately, were not able to release the HiRise code and do not anticipate this changing in the
50 future. Our pipeline does not have variable parameters and only uses MQ>50 for scaffolding.” We
note that more than 75 genome papers that have used Dovetail services have been published in
reputable journals including *Nature* and *Nature Genetics* and none of them have provided the HiRise
code.

**[R2 Comment 03]** 3) The analyses seem to have been run in three parts that define a specific
hypothetical duplication scenario. Why was the reconstruction not performed using all of the data in
a single run?

One of the major aims of our study is to investigate if the proto-gnathostome and the proto-
cyclostome lineages share the same duplication events (including 1R). The best way to achieve this
is to compare the genome structure between independently reconstructed genomes of the proto-
cyclostome and proto-gnathostome lineages. Otherwise it would be difficult to distinguish shared
rearrangements from inference artefacts.

**What happens if they do this? Is the reconstruction the same or different?**

To our knowledge, single-run approaches for multiple WGDs in different lineages have never been
proposed before. We think that it requires development of suitable evolutionary models and
inference algorithms that account for the lack of clear orthology relationship between proto-
cyclostome chromosomes and proto-gnathostome chromosomes.

**[R2 Comment 04]** 4) The authors state “To distinguish between alternative polyploidization models
(i.e. S5–S8 in Fig. 1) we introduced a measure we have called multiplicity” this is clearly not a new
idea and they should consider rewording (e.g. Putnam, N.H. et al. 2007).

We have revised the text as follows: "To distinguish between alternative polyploidization models
(i.e. S5–S8 in Fig. 1) we followed ref. [Nakatani et al. *Genome Res* (2007)] and used a measure we
have called multiplicity..."

The idea was initially proposed by us in Nakatani et al., *Genome Res* (2007), which was an extension
of the idea proposed in [Dehal and Boore, *PLoS Biol* (2005)]. In Nakatani et al. (2007), candidate
reconstructions into two, three, four, or five post-2R chromosomes were compared and the optimal
reconstruction to define the proto-gnathostome chromosomes was chosen. A similar optimality
analysis was performed in [Muffato, PhD Thesis (2012)], but not in [Putnam et al. *Science* (2007);
Putnam et al. *Nature* (2008)].

In addition, assessment of multiplicity cannot (as implemented) define the mode of duplication that
gave rise to the patterns without explicit statistical tests. I laid out how to perform these tests in the
previous review, but these were not performed.

In the revised manuscript submitted previously, we had added a detailed discussion that the clear
peak at multiplicity six suggests six-fold duplication of the entire genome between the proto-
vertebrate and proto-cyclostome lineages (please also see our response to Comment 12 in the
previous review). This time, we have developed (see below) a framework for calculating the
probability that the multiplicities of the proto-vertebrate chromosomes converge toward six through
a series of independent chromosome-scale duplication events.

From examining the code it appears that they used clustering method seeded with 18 clusters to
assign lamprey chromosomes to their ancestral chromosomes. It is therefore even less surprising
now that they would observe a peak at 6.

As described in the Supplementary Information, we tried reconstructions with $K = 10, \dots, 20$ and
then the optimal value ($K=18$) was chosen (see Supplementary Information Section 3.2.3 and
Supplementary Table S8). In fact the choice of the value of K has limited influence on the
multiplicity peak at six. This is because changing the value of K mainly affects smaller proto-
vertebrate chromosomes, and it doesn't affect our observation that each of the largest five proto-
vertebrate chromosomes were duplicated into six proto-cyclostome chromosomes.

Please provide explicit tests of multiple duplication scenarios as laid out in the previous review.

To test chromosome-scale duplication scenarios, we have now introduced a framework for
calculating the probability that multiplicities of independently duplicating chromosomes converge
toward a given ploidy level, where the convergence is measured in terms of the deviation (δ) from
the given ploidy level. Application to the proto-cyclostome genome shows that the observed peak of

multiplicity at six is unlikely to be created by chance through accumulation of chromosome-scale
duplications.

Let us consider the following situation. The proto-vertebrate genome with K chromosomes
underwent one or two polyploidization events, producing X_k ($k = 1, \dots, K$) duplicates for each
proto-vertebrate chromosome ($X_k = 2$ for all k after 1R or $X_k = 4$ after two rounds of
tetraploidization). Subsequently, those $X = \sum_{k=1}^K X_k$ chromosomes were duplicated by a series of
independent chromosome-scale duplications, eventually creating Y_k duplicates for each proto-
vertebrate chromosome ($k = 1, \dots, K$). As a measure of deviation from a polyploidization-only
model, we define $\delta(Y_k) = \sum_{k=1}^K |Y_k - M|$, where M is the expected multiplicity ($M = 6$ in our
model). Assuming that all chromosomes are equally likely to be duplicated, we calculate
$P(\delta(Y_k) \leq D | \sum_{k=1}^K Y_k = Y)$, the probability that the deviation is smaller than or equal to the
observed deviation D (i.e. $D = 13$ in our reconstruction) conditioned by the total number of proto-
cyclostome chromosomes Y (i.e. $Y = 103$ in our reconstruction).

The desired probability is calculated as follows. First, the total number of duplication scenarios is
given by $T = \Gamma(Y)/\Gamma(X)$, where $\Gamma(n) = (n-1)(n-2) \cdots 1$ is the gamma function. Second, for
given Y_k ($k = 1, \dots, K$), the number of duplication scenarios in which individual proto-vertebrate
chromosomes are eventually duplicated into Y_k proto-cyclostome chromosomes is given by
$S(Y_1, \dots, Y_K) = (Y_1, \dots, Y_K)! \prod_{k=1}^K \Gamma(Y_k)/\Gamma(X_k)$, where $(Y_1, \dots, Y_K)!$ is the multinomial coefficient.
Then, by enumerating all Y_k values, we can calculate the desired probability (i.e. independently
duplicating proto-vertebrate chromosomes converging to multiplicity M by chance alone) as
$P(\delta(Y_k) \leq D | \sum_{k=1}^K Y_k = Y) = \sum_{\{Y_k\}} S(Y_1, \dots, Y_K)/T$, where the summation is taken over all Y_k that
satisfy $\delta(Y_k) \leq D$ and $\sum_{k=1}^K Y_k = Y$.

In our reconstruction, we have $K = 17$, $Y = 103$, $D = 13$ and $M = 6$ (see Table below). We
evaluate the following five evolutionary scenarios: (A) chromosome-scale duplications with no
tetraploidization, (B) one tetraploidization followed by chromosome-scale duplications, (C) two
tetraploidizations followed by chromosome-scale duplications, (D) chromosome-scale duplications
followed by one tetraploidization, and (E) first tetraploidization followed by chromosome-scale
duplications followed by second tetraploidization. In these scenarios we assume that $X_k = N$ for all
k , where we set $N = 1$ and $M = 6$ for Scenario A; $N = 2$ and $M = 6$ for Scenario B; $N = 4$
and $M = 6$ for Scenario C; $N = 1$ and $M = 3$ for Scenario D; and $N = 2$ and $M = 3$ for
Scenario E. We set $(Y_1, \dots, Y_{17}) = (6, 5, 6, 6, 7, 7, 6, 6, 4, 6, 8, 5, 6, 4, 9, 6, 6)$ for Scenarios A/B/C and
$(Y_1, \dots, Y_{17}) = (3, 2, 3, 3, 3, 3, 3, 2, 3, 4, 2, 3, 2, 4, 3, 3)$ for Scenarios D/E, based on the proto-cyclostome
genome reconstruction. In addition, we evaluate the case of $K = 5$, $Y = 30$ and $D =$

0 (see Table below), because larger proto-vertebrate chromosomes are more reliable in our
 reconstruction and the largest five proto-vertebrate chromosomes have multiplicity six, as we have
 discussed in the main text.

Scenario	K	Y	D	N	M	P
A	17	103	13	1	6	0.0000000018
B	17	103	13	2	6	0.0000030304
C	17	103	13	4	6	0.0214209597
D	17	49	6	1	3	0.0000002044
E	17	49	6	2	3	0.0038115884
A	5	30	0	1	6	0.0000421035
B	5	30	0	2	6	0.0003120318
C	5	30	0	4	6	0.0049925087
D	5	15	0	1	3	0.0009990010
E	5	15	0	2	3	0.0159840160

 The table shows small probabilities of observing convergence of multiplicities through independent
 chromosome-scale duplications. Thus, it is unlikely that the proto-cyclostome genome was shaped
 by a series of independently occurring chromosome-scale duplications.

 Based on this analysis, we have revised the main text as follows.
 "In addition, we confirmed that the observed peak of multiplicity (Fig. 3d) is unlikely to have been
 created by accumulation of chromosome-scale or segmental duplications after a tetraploidization
 event (Scenario S8 in Fig. 1) by statistical test ($P=0.0000030304$, see Supplementary Information
 Section 3.5 for details). Thus, the clear peak at multiplicity of six is compelling evidence of six-fold
 duplication of the entire genome, probably through a tetraploidization and a hexaploidization event."

 **[R2 Comment 05]** 5) I again request that they change "proto-cyclostome" to "proto-petromyzontid".
 The observation of six hox clusters in hagfish may be consistent with their observations, but hardly
 raises to the level where one might imagine that hexaploidy should be assumed for the entire hagfish
 genome with certainty.

The sea lamprey genome paper [Smith et al., *Nat Genet* (2018)] and our analysis found the absence
 of clear distinction between ancient duplication and more recent duplication events among lamprey
 chromosomes, suggesting that the more recent duplication (that we call cyclostome-specific
 hexaploidization) occurred shortly after 1R. In addition, our analysis showed strong gene order
 conservation between Japanese lamprey scaffolds and sea lamprey scaffolds, whereas little gene
 order conservation was observed between paralogous scaffolds generated by the hexaploidization
 event. These observations suggest that the hexaploidization event occurred in an ancestral lineage
 considerably more ancient than the proto-petromyzontid.

Nevertheless, the exact phylogenetic position of the hexaploidization event should eventually be
determined by analyzing the chromosome-level hagfish genome assembly, and it would be an
exciting discovery if the lamprey and hagfish genomes underwent independent hexaploidization
events as suggested by Reviewer 2. At present, however, we do not have supporting evidence for
two independent hexaploidization events in the two lineages as opposed to a single hexaploidization
event shared by the two lineages. For this reason, we would prefer to retain the original description
and call the reconstruction “proto-cyclostome”, following the study of Hox clusters in hagfish which
suggested that the hagfish and lamprey lineages share the same cyclostome-specific duplication
event [Pascual-Anaya et al., *Nat Ecol Evol* (2018)].

**[R2 Comment 06]** 6) Part of the justification that their “cyclostome” reconstruction is plausible is
that there have been other described instances of hexaploidy in vertebrates, despite the obvious
issues this raises for obligately sexually reproducing species.

We would like to make it clear that we didn’t present the cases of hexaploidization in vertebrates as
a justification of our reconstruction. We mentioned those cases of hexaploidization to inform readers
that hexaploidization is not impossible in vertebrates.

However, this assertion seems to be a misinterpretation of those bodies of literature. Shortnose
sturgeon have been called “functionally hexaploid” due to pervasive loss of duplicates
(microsatellites) following WGDs, but they are clearly of octaploid origin (Symonová R et al *BMC*
*Genet.* 2017).

This seems to be a misunderstanding by Reviewer 2. It has been considered that the shortnose
sturgeon with ~360 diploid chromosomes is a functional hexaploid of dodecaploid origin [see
Fontana et al., *Genome* (2008); Trifonov et al., *Chromosoma* (2016)], and the diploid chromosome
number was inferred to have increased as 60-120-240-360 from the ancestral sturgeon to the
shortnose sturgeon (see Fig. 5 in [Symonová et al. *BMC Genet* (2017)], Fig. 3 in [Trifonov et al.,
*Chromosoma* (2016)], and Fig. 3 in [Fontana et al., *Genome* (2008)]).

Prussian carp hexaploids reproduce only by gynogenesis and are sexual parasites on diploid and
tetraploid populations. Justifying their model from a biological standpoint will require substantial
alteration of the current discussion and should address whether they are invoking gynogenesis in the
origin of cyclostomes/petromyzontids and how this might have transitioned back to a stably meiotic
lineage.

We didn’t try to justify our model by mentioning the hexaploidization in carp. We have mentioned
several cases of documented hexaploidization in vertebrates for facilitating discussions from a

biological standpoint, as in this reviewer comment. At present, we do not have sufficient information
for discussing the possibility of gynogenesis in cyclostomes, but we thank Reviewer 2 for raising
this interesting point for future discussion.

**[R2 Comment 07]** 7) The authors state “Although several recent studies supported the ancient origin
of microchromosomes, it was still unknown (1) if chromosomal features characteristic to modern
avian microchromosomes (i.e. high GC content, high gene density and high recombination rate)
were already present in the ancestral gnathostome genome” although this seems to not to
acknowledge analyses of the spotted gar genome that resolved many of these feature for the
ancestral euteleostome, which is only ~40 million years divergent from the ancestral gnathostome
they are reconstructing (Braasch, et al. *Nat Genet*, 2016).

As Reviewer 2 mentioned, the phylogenetic distance seems very short between proto-gnathostome
(jawed vertebrate) and proto-euteleostome (bony vertebrate). However, it was not obvious if two
phylogenetically close lineages share the same chromosomal features, especially when chromosomal
structures do not evolve at a constant rate. We acknowledge that Braasch and colleagues had made a
significant progress, as we had already mentioned in Supplementary Information Section 4.1 of the
previously submitted manuscript. In the revised manuscript, we have amended the main text as
below and added the citation of [Braasch et al., *Nat Genet* (2016)].

"Although several recent studies supported the ancient origin of microchromosomes (for example, a
comparison between the chicken and spotted gar genomes suggested that the origin of
microchromosomes goes back to the ancestral bony vertebrate [Braasch et al. *Nat Genet* (2016)]), ..."

Additionally, the spotted gar genome paper is not cited at all despite use of the assembly for their
reconstructions.

Some of the genomes used in our reconstructions were cited only in Supplementary Information due
to space limitation, but we have now added references for the chicken and spotted gar genomes in
the revised main text.

**[R2 Comment 08]** 8) The authors state “In order to verify the timing of the gnathostome-cyclostome
divergence with respect to 1R and 2R, we performed an analysis based off Ensembl gene trees on the
reconstructed chromosomes ...” These analyses should be re-done from scratch as ENSEMBL trees
are forced to a pre-defined topology (with lamprey splits specifically designated as basal
gnathostome splits) and are therefore cannot be directly used to perform the tests. This is laid out in
(Smith et al. *Nat Genet*, 2013), but has been notably been erroneously used by others to test similar
ideas to those presented in this manuscript.

Ensembl Compara actually utilizes a tree inference method that reconciles a gene tree and a species

tree (thus the description “forced to a pre-defined topology” is inaccurate), and it is fine to assume
that lamprey diverged from the base of gnathostomes. As suggested by Reviewer 2, we have now
additionally performed a gene tree analysis in which trees were inferred from sequence alignments
without using species trees. We used the resulting trees to plot vertebrate-specific and cyclostome-
specific paralogs on the proto-cyclostome genome and examined if the cyclostome-specific paralogs
were enriched between certain chromosome pairs (Supplementary Figure S13). The result was
largely consistent with our previous gene tree analyses (Supplementary Figures S10–S12), and thus
our conclusions remain unchanged. The method is described in Supplementary Information Section
5.1, and the results are presented as Supplementary Figure S13 as shown below.

Supplementary Information Section 5.1

“**D.** Confirmation by gene tree inference with RAxML without using Ensembl gene trees. We
excluded *P. marinus* in Step 3, and inferred gene trees from the alignments in Step 2 (using RAxML
with the WAG substitution matrix, instead of just inserting lamprey genes into Ensembl gene trees
using RAxML-EPA). To exclude tandem duplications and partially annotated genes, we retained
only one-to-one orthologues between Japanese lamprey and sea lamprey (i.e. a pair of lamprey genes
are one-to-one orthologues if the two lamprey genes are only descendants of their common ancestor
node). The result is shown in Figure S13.”

**Figure S13.** Distribution of Japanese lamprey paralogs annotated using gene trees inferred by
RAxML. Instead of inserting lamprey genes into existing Ensembl gene trees using RAxML-EPA,
gene trees were inferred using RAxML with the WAG matrix, and one-to-one orthologues between
Japanese lamprey and sea lamprey were retained for paralogue annotation.

To address the concerns raised by Reviewer 2 on the reliability of gene tree analysis, we have
revised the paragraph of gene tree analysis in the main text as shown below. In short, we enumerated
possible interpretations of our observations, and deleted the supplementary section on the possibility
of hybridizations between genetically diverse subpopulations. Those discussions only showed
possible interpretations of our results, so the current revisions do not affect the results and
conclusions of our manuscript.

“In order to verify the timing of duplications and the gnathostome-cyclostome divergence, we
performed gene tree analyses by inserting lamprey genes into Ensembl gene trees or re-computing
the gene trees (see Supplementary Section 5). Then, we classified human and lamprey paralogue
pairs by their duplication timing and plotted vertebrate paralogues (i.e. paralogues duplicated before
the gnathostome-cyclostome split), gnathostome-specific paralogues and cyclostome-specific
paralogues on the proto-gnathostome and proto-cyclostome genomes (Supplementary Figs. S9–S15).
Intriguingly, we observed a mixture of vertebrate paralogues and cyclostome-specific paralogues
between most pairs of homoeologous proto-cyclostome chromosomes, making it difficult to
conclusively determine the duplication timing of individual chromosomes. This observation may be
explained by (1) difficulties in gene tree inference due to the high GC content and strong codon bias
in the lamprey genomes^{22,26,33}, (2) differential gene loss between cyclostome and gnathostome
lineages²⁹, (3) delayed rediploidization^{28,31,32} creating cyclostome-specific paralogues between proto-
cyclostome chromosomes duplicated by 1R, and (4) tetraploidization through hybridization and
doubling⁵⁴⁻⁵⁶, which may have created both vertebrate-specific and cyclostome-specific paralogues
due to recurrent hybridization among genetically diverse subpopulations^{54,55} and subsequent genetic
drift⁵⁷. Although these factors may have obscured the duplication timing, the presence of
chromosome pairs enriched either with vertebrate-specific paralogues or cyclostome-specific
paralogues is consistent with the model that the proto-cyclostome lineage diverged from the proto-
gnathostome lineage shortly after 1R.”

**[R2 Comment 09]** 9) The authors should present, in the main manuscript, more detail regarding the
numbers and distribution of ohnologs (and other duplicates) across presumptive paralogous
segments (particularly those in Figure 6d/e/g). This is essential to evaluating evidence favoring
duplication vs fission in the origin of these segments, which is in turn essential for evaluating
evidence as it relates to proposed duplication scenarios (both gnathostome and cyclostome).

We have already presented distributions of paralogs and orthologs among reconstructed
chromosomes comprehensively in Figures 2, 3, 4, Supplementary Figures S7, S9, S10, S11, S12,

S13, S14 and S15. These figures provide essential information for evaluating the accuracy of our
reconstructions. Please also see our response to [R2 Comment 11] below.
The table below shows the numbers of paralogs between pairs of proto-gnathostome chromosomes.
The four numbers in each cell indicate human, chicken, spotted gar, and elephant shark paralogs.
Empty cells indicate no paralogs. (Please magnify this file to see the numbers. Old versions of Word
might not be able to display the tables as vector graphics.)

The table below shows the numbers of paralogs between pairs of proto-cyclostome chromosomes.
The two numbers in each cell indicate Japanese lamprey and sea lamprey paralogs. Empty cells
indicate no paralogs. (Please magnify this file to see the numbers.)

The above two tables are included in Supplementary Data 1 as a PDF document
(ChromosomeStatistics.pdf).

The authors should be able to gain some inspiration as to how to do this by looking at another paper
that is generally similar to this one (Simakov, et al. *Nat Ecol Evol* 2020: Fig 3, 4b) and even improve
upon that presentation.

Actually Simakov et al. (*Nat Ecol Evol* 2020) have not presented the numbers of paralogs among
reconstructed chromosomes. Figure 3 in [Simakov et al., *Nat Ecol Evol* (2020)] shows only the
reconstructed chromosomes and the numbers of genes. In our manuscript, the corresponding
information is presented as Supplementary Figure S7. Figure 4b in [Simakov et al., *Nat Ecol Evol*
(2020)] shows biased rates of gene retention between the two subgenomes. In our manuscript, the
corresponding information is presented as Figure 6e and Supplementary Figure S7.

In fact, tables of paralogs were not presented in previous lamprey genome papers [Smith et al., *Nat*
*Genet* (2013); Smith and Keinath, *Genome Res* (2015); Smith et al., *Nat Genet* (2018)] nor in
previous reconstruction papers [Nakatani et al., *Genome Res* (2007); Putnam et al., *Nature* (2008);
Sacerdot et al., *Genome Biol* (2018)] including Simakov et al. *Nat Ecol Evol* (2020), because long
and complicated tables of numbers are uninformative and unhelpful for readers. Instead, previous
papers presented figures plotting orthologs and paralogs so that synteny evidence can be examined
visually. This is how developers of reconstruction programs check the accuracy and performance of
their code. We have already presented Figures 2, 3, 4, Supplementary Figures S7, S9, S10, S11, S12,
S13, S14 and S15, and we believe that such visualization is the key to evaluating the accuracy of our
reconstructions and reliability of evolutionary scenarios. We also provided reconstruction
information as Supplementary Data 1, which makes it easy to confirm the numbers of genes,
orthologs, paralogs, etc. in more detail.

I requested something similar in the previous review (and prior to seeing the Simakov paper) but this
request was not satisfactorily addressed, and the revision makes the need for this even clearer.

In our previous response to reviewer comments, we had presented a table showing the number of
genes and orthologs as requested by Reviewer 2 (see our response to Comment 14 in the previous
review). The same information had already been visualized as Supplementary Figure S4 and S7 in
the manuscript, which we consider to be more informative than a long table of numbers. Please also
see our response to [R2 Comment 11] below.

**[R2 Comment 10]** 10) Given that the Simakov paper was released after the initial submission of this
manuscript it may be unfair to require that the authors consider the specific models proposed by that
paper, but I am certain readers will welcome it and perhaps expect it. Details of these reconstructions
differ in profound ways.

Supplementary Table 7 in [Simakov et al., *Nat Ecol Evol* (2020)] shows one-to-one correspondence

between the reconstruction in [Simakov et al., *Nat Ecol Evol* (2020)] and the reconstruction in
 [Putnam et al., *Nature* (2008)]. We have presented a comparison between our reconstruction and the
 reconstructions in [Putnam et al., *Nature* (2008)] and [Simakov et al., *Nat Ecol Evol* (2020)] in
 Supplementary Table S9 (reproduced below), and discussed the differences in Supplementary
 Information Section 3.2.4.

This study	Putnam et al.	Sacerdot et al.	Simakov et al.	Scallop
Pvc1	CLG16	chr1	CLGB	chr15,18,19
Pvc2	CLG3	chr10	CLGD	chr1
Pvc3	CLG4	chr11	CLGJ	chr5,10
Pvc4	CLG5	chr12	CLGK	chr2
Pvc5	CLG10	chr17	CLGP	chr11
Pvc6	CLG9	chr16	CLGN	chr17
Pvc7	CLG8	chr15	CLGF	chr8
Pvc8	CLG6	chr14	CLGQ	chr6
Pvc9	CLG7	chr14	CLGI	chr4
Pvc10	CLG13	chr5	CLGE	chr7
Pvc11	CLG14	chr6	CLGO	chr2,16
Pvc12	CLG15	chr4	CLGH	chr6
Pvc13	CLG2	chr7	CLGC	chr12,14
Pvc14	CLG1	chr8	CLGL	chr5
Pvc15	CLG12	chr9	CLGM	chr15
Pvc16	CLG17	chr2	CLGG	chr9
Pvc17	CLG11	chr3	CLGA	chr3
Pvc18	CLG11	chr13	CLGA	chr13

[R2 Comment 11] 11) Related to the above comments, please also provide numbers of orthologs
 that support each of the conserved segments in Figure 6E and 6G (after addressing other points).
 Presentation in the main manuscript will provide essential detail to the reader.

Please see our response to Comment 14 in the previous review. The essential details are shown in
 Supplementary Figure S7, which are more intuitive and informative than the long tables of numbers.
 The table below shows the numbers of human, chicken, spotted gar and elephant shark genes
 mapped to the proto-gnathostome chromosomes and amphioxus orthologs that were mapped to
 corresponding proto-vertebrate chromosomes as visualized in Figure 6e. The order of the proto-
 gnathostome chromosomes corresponds to Figure 6e. This table is included in Supplementary Data
 1.

Proto-gnathostome	Human genes	Chicken genes	Spotted gar genes	Elephant shark genes	Amphioxus orthologs
Pgc1	695	568	1127	676	496
Pgc2	335	251	288	306	252
Pgc3	280	42	227	229	194
Pgc4	541	471	564	611	473
Pgc5	413	349	391	436	335
Pgc6	227	26	0	0	61
Pgc7	774	702	258	802	569
Pgc8	395	328	362	356	278
Pgc9	741	600	640	651	627
Pgc10	268	216	279	298	165
Pgc11	318	245	281	329	207
Pgc12	142	85	93	16	60
Pgc13	498	421	528	515	333
Pgc14	132	0	201	49	117
Pgc15	792	635	1166	836	720
Pgc16	437	327	361	409	343
Pgc17	802	702	763	963	735
Pgc18	444	329	386	415	322
Pgc19	589	501	529	591	400
Pgc20	373	118	265	205	204
Pgc21	333	325	366	393	299
Pgc22	174	0	130	20	90
Pgc23	783	687	741	845	663
Pgc24	303	240	317	329	248
Pgc25	447	295	306	354	290
Pgc26	0	0	26	0	37
Pgc27	389	17	0	0	35
Pgc28	493	398	442	438	359
Pgc29	332	307	341	366	281
Pgc30	143	168	144	234	128
Pgc31	434	380	434	441	367
Pgc32	268	160	350	0	160
Pgc33	910	739	790	721	644
Pgc34	287	237	319	367	296
Pgc35	562	487	518	565	447
Pgc36	276	0	187	186	123
Pgc37	323	272	297	345	260
Pgc38	215	45	0	157	53
Pgc39	451	341	376	445	290
Pgc40	429	305	333	334	261
Pgc41	134	85	44	0	81
Pgc42	81	7	38	0	38
Pgc43	614	496	520	547	482
Pgc44	519	517	569	693	504
Pgc45	182	0	150	35	137
Pgc46	371	0	211	59	153
Pgc47	367	353	0	286	147
Pgc48	236	173	222	201	12
Pgc49	100	4	120	55	2

The table below shows the statistics for the proto-cyclostome chromosomes. This table was presented
in our response to [Comment 14] in the previous review, but we found one amphioxus scaffold was
excluded by mistake. We have fixed the table and presented it below. Each line shows (1) proto-
vertebrate chromosome name (Pvc), (2) number of amphioxus genes mapped to the Pvc, (3) proto-
cyclostome chromosome name (Pcc), (4) number of Japanese lamprey genes mapped to the Pcc, (5)
number of sea lamprey genes mapped to the Pcc, and (6) number of amphioxus genes that are mapped
to the Pvc and are orthologous to lamprey genes mapped to the Pcc. This table is included in
Supplementary Data 1.

Proto-vertebrate	Amphioxus genes	Proto-cyclostome	Japanese lamprey genes	Sea lamprey genes	Orthologous amphioxus genes
Pvc1	1445	Pcc1A	515	473	304
Pvc1	1445	Pcc1B	502	452	283
Pvc1	1445	Pcc1C	397	344	250
Pvc1	1445	Pcc1D	303	266	177
Pvc1	1445	Pcc1E	226	217	154
Pvc1	1445	Pcc1F	158	179	121
Pvc2	891	Pcc2A	287	326	174
Pvc2	891	Pcc2B	252	228	167
Pvc2	891	Pcc2C	206	225	154
Pvc2	891	Pcc2D	172	177	115
Pvc2	891	Pcc2E	155	184	78
Pvc3	686	Pcc3A	264	265	99
Pvc3	686	Pcc3B	261	234	96
Pvc3	686	Pcc3C	237	220	89
Pvc3	686	Pcc3D	231	212	88
Pvc3	686	Pcc3E	120	131	52
Pvc3	686	Pcc3F	62	114	47
Pvc4	473	Pcc4A	171	169	89
Pvc4	473	Pcc4B	143	171	67
Pvc4	473	Pcc4C	127	177	66
Pvc4	473	Pcc4D	18	0	1
Pvc4	473	Pcc4E	0	16	1
Pvc4	473	Pcc4F	4	4	0
Pvc5	525	Pcc5A	190	201	112
Pvc5	525	Pcc5B	189	175	96
Pvc5	525	Pcc5C	50	44	20
Pvc5	525	Pcc5D	0	38	11
Pvc5	525	Pcc5E	10	22	6
Pvc5	525	Pcc5F	7	8	2
Pvc5	525	Pcc5G	9	0	1
Pvc6	586	Pcc6A	182	212	105
Pvc6	586	Pcc6B	188	171	86
Pvc6	586	Pcc6C	108	101	55
Pvc6	586	Pcc6D	46	56	33
Pvc6	586	Pcc6E	36	26	17
Pvc6	586	Pcc6F	0	61	9
Pvc6	586	Pcc6G	10	0	0
Pvc7	707	Pcc7A	271	266	173
Pvc7	707	Pcc7B	260	271	155
Pvc7	707	Pcc7C	124	112	66
Pvc7	707	Pcc7D	162	25	43
Pvc7	707	Pcc7E	11	0	2
Pvc7	707	Pcc7F	0	10	2
Pvc8	420	Pcc8A	276	252	130
Pvc8	420	Pcc8B	207	212	100
Pvc8	420	Pcc8C	40	32	10
Pvc8	420	Pcc8D	15	15	4
Pvc8	420	Pcc8E	0	11	3
Pvc8	420	Pcc8F	2	4	0
Pvc9	563	Pcc9A	355	344	174
Pvc9	563	Pcc9B	277	282	159
Pvc9	563	Pcc9C	145	145	30
Pvc9	563	Pcc9D	23	0	3
Pvc10	962	Pcc10A	257	240	151
Pvc10	962	Pcc10B	252	240	148
Pvc10	962	Pcc10C	218	228	129
Pvc10	962	Pcc10D	196	205	120
Pvc10	962	Pcc10E	172	202	115
Pvc10	962	Pcc10F	128	170	90
Pvc11	844	Pcc11A	314	296	167
Pvc11	844	Pcc11B	225	261	126
Pvc11	844	Pcc11C	107	132	74
Pvc11	844	Pcc11D	90	80	44
Pvc11	844	Pcc11E	58	106	39
Pvc11	844	Pcc11F	12	15	8
Pvc11	844	Pcc11G	0	28	7
Pvc11	844	Pcc11H	16	0	1
Pvc12	798	Pcc12A	366	361	181
Pvc12	798	Pcc12B	258	259	157
Pvc12	798	Pcc12C	225	246	151
Pvc12	798	Pcc12D	157	313	113
Pvc12	798	Pcc12E	0	14	5
Pvc13	1196	Pcc13A	470	441	234
Pvc13	1196	Pcc13B	346	342	203
Pvc13	1196	Pcc13C	251	232	151
Pvc13	1196	Pcc13D	188	217	115
Pvc13	1196	Pcc13E	141	173	101
Pvc13	1196	Pcc13F	24	0	9
Pvc14	602	Pcc14A	242	224	130
Pvc14	602	Pcc14B	175	187	104
Pvc14	602	Pcc14C	85	159	53
Pvc14	602	Pcc14D	0	25	6
Pvc15	560	Pcc15A	251	239	126
Pvc15	560	Pcc15B	164	194	99
Pvc15	560	Pcc15C	91	66	47
Pvc15	560	Pcc15D	53	96	38
Pvc15	560	Pcc15E	33	12	20
Pvc15	560	Pcc15F	16	23	14
Pvc15	560	Pcc15G	0	31	12
Pvc15	560	Pcc15H	11	0	4
Pvc15	560	Pcc15I	4	6	2
Pvc16	689	Pcc16A	283	267	180
Pvc16	689	Pcc16B	263	254	144
Pvc16	689	Pcc16C	42	9	11
Pvc16	689	Pcc16D	9	12	3
Pvc16	689	Pcc16E	0	10	3
Pvc16	689	Pcc16F	3	5	2
Pvc17	1282	Pcc17A	491	420	291
Pvc17	1282	Pcc17B	326	313	203
Pvc17	1282	Pcc17C	302	301	184
Pvc17	1282	Pcc17D	298	269	182
Pvc17	1282	Pcc17E	295	265	173
Pvc17	1282	Pcc17F	140	168	108
Pvc18	197	Pcc18A	859	569	59

**[R2 Comment 12]** 11) Related to point 9, and with apologies for the length of this comment, it appears
that the program used for these analyses makes a statistical faux pas in assessing evidence that that
two ancestrally linked segments are derived from fission vs. duplication. If I am interpreting the code
correctly, the authors use a statistically appropriate test to identify segments that have an excess of
shared homologs or ohnologs relative to random. Many of the other studies mentioned in the
manuscript and above others have used similar approaches to although the use of the hypergeometric
distribution for these tests is laudable. However, it appears that the ohnolog statistic is compared to
the ortholog statistic as part of the assessment of whether a segment is likely to be derived from
duplication vs fission which does not really shed light on the question at hand if this is true, and appears
to not be an appropriate use of these values. Though admittedly the code here is a bit hard to follow
given the layout and the supplement seems not to clearly address this. Issues with p-value/ test
probability comparison may not be immediately obvious to the casual observer, but were pointed out
by Fisher and subsequently by many others due to pervasive misuse (a couple of modern
examples: <https://www.ncbi.nlm.nih.gov/pmc/articles/PMC5804470/>, <https://www.tandfonline.com/doi/abs/10.1080/00031305.1996.10474380>).

A more appropriate approach might be to compare the observed frequency of duplicates on
presumptive paralogous segments to the distribution of similarly-aged duplicates within conserved
segments. It is necessary to infer that some duplications will be present between fissioned segments
because, 1) intrachromosomal duplication is known to occur frequently, even within the human
population, and 2) intrachromosomal rearrangements have effectively randomized gene orders over
the timescales that are analyzed here; one would expect that two pieces of a fissioned chromosome
will carry paralogs that are derived from ancient intrachromosomal duplications (i.e. not WGDs).

Therefore, one should be able specifically test whether the presumptive WGD paralogous segments
carry more duplicates than would be expected for the average fission event. A test like this is seemingly
critical given the definitive statements that are made throughout the manuscript.

Perhaps also clearly state in the methods how duplication are differentiated from fissions, especially
if I have made some error in interpretation here.

As described in Supplementary Information Section 3.3, we defined the proto-cyclostome and proto-
gnathostome chromosomes by the optimal set partition with the most non-random distribution of
paralogs and orthologs. The underlying assumption is that genome rearrangements increase
randomness by scattering the distribution of paralogs, and the most non-random configuration
represents the ancestral genome organization [Nakatani et al., *Genome Res* (2007); Muffato, PhD
Thesis (2010)]. Following this idea, we defined nonrandomness as described in Supplementary
Information Section 3.3 and used it as our optimization criterion, but other researchers may choose
different optimization criteria (including the ‘more appropriate approach’ suggested by Reviewer 2).

In this step, we didn't perform hypothesis testing. Ortholog information is necessary, for example, for
assigning human HoxA and mouse HoxA to the same proto-gnathostome chromosome. Otherwise we
might get a proto-gnathostome chromosome with human HoxA and mouse HoxB, and another proto-
gnathostome chromosome with human HoxB and mouse HoxA. In addition, ortholog information is
helpful for correctly assigning short segments with few paralogs to the correct proto-gnathostome
chromosomes. For example, if a short mouse segment with no paralogs is orthologous to a large human
segment, we can find the optimal assignment of the mouse segment through the paralog information
of the orthologous human segment.

In our reconstruction program, hypothesis testing was performed when we identify significantly
paralogous segment pairs and significantly orthologous segment pairs with $p < 10^{-5}$ (see
Supplementary Information Sections 3.3.1 and 3.3.2). In this step, a conservative threshold was chosen
because identification of a small number of clearly paralogous and clearly orthologous segment pairs
was sufficient for reducing the search space and computation time (see Supplementary Information
Section 3.3). We would like to emphasize that the purpose of this step is reduction of computation
time and not classification of all segment pairs into duplication pairs and fission pairs accurately.

Reviewer 2 suggested using only intra-segment paralog frequency to classify segment pairs into
duplicate pairs and fission pairs. In reality it doesn't work because many of partially annotated genes
tend to be classified incorrectly as paralogs. In addition, it is important to recognize that not all
duplicated segment pairs share significantly large numbers of paralogs, especially when multiple
rounds of WGD are involved, as previously discussed in [Simillion et al., *PNAS* (2002); Vandepoele
et al., *Trends Genet* (2002)]. Even in such cases, the true paralogy can be detected by a multi-way
comparison of multiple paralogous segments from multiple species [Simillion et al., *Genome Res*
(2004); Van de Peer, *Nat Rev Genet* (2004)]. Our reconstruction method addressed these issues
(including the presence of paralogs between fission segments) by optimizing the non-random
distribution of orthologs and paralogs through set partitioning of multiple segments from multiple
species (Supplementary Information Section 3.3).

As for classification of fission and duplication, we admit that there can be difficult cases.

**Example:** Suppose that an entire chromosome arm (arm1) of chr1 was duplicated, producing
"arm1+arm2+arm1" chromosome. If this chromosome undergoes fission into "arm1+arm2"
chromosome and "arm1" chromosome, we should observe a large number of paralogs between
"arm1+arm2" and "arm1" chromosomes. We call it duplication of chr1 into "arm1+arm2" and
"arm1" chromosomes.

Thus, we should be aware of the possibility that not all fissions are classified as fissions. In our analysis,
we reconstructed proto-cyclostome chromosomes and proto-gnathostome chromosomes as described
in Supplementary Information Section 3.3, and if two segments on different chromosomes in a genome
are mapped to the same proto-cyclostome chromosome or the same proto-gnathostome chromosome,
they are considered to be created by fission or translocation from the proto-cyclostome or proto-
gnathostome chromosome. Fissions between 1R and 2R can be detected by comparison with outgroup
genomes as we mentioned in our response to Comment 15 in the previous round of review (see also
[Nakatani et al., *Genome Res* (2007)]), although we found no fissions between 1R and 2R.

Finally, we would like to emphasize that our conclusions remain unchanged even if some of the
reconstructed proto-cyclostome chromosomes were actually created by fission, for the following
reasons. First, even if a small number of reconstructed proto-cyclostome chromosomes may have been
created by fission, paralog plots (Supp Figs. S9–15) show most of the chromosomes are likely to have
been created by duplication. Second, as we have already discussed in the main text (see our response
to Comment 22 in the previous round of review), smaller proto-cyclostome chromosomes are less
reliable and some of them may have been reconstructed inaccurately. Nevertheless, our conclusion of
cyclostome-specific hexaploidization remains unchanged since the conclusion is supported by the
reconstruction as a whole (e.g. supported by the clear peak of multiplicity at six). Third, our statistical
analysis for testing chromosome-scale duplication scenarios (see our response to [R2 Comment 04]
above) is not affected if individual chromosomes were created by duplication or by fission. Since it is
statistically unlikely that the chromosome number increased one-by-one, the observed convergence of
multiplicity should be explained by a biological mechanism through which all chromosomes were
broken into multiple parts simultaneously or duplicated simultaneously. Since there is no such
mechanism like whole-genome fission, we conclude that the proto-cyclostome genome was shaped by
polyploidization.

[R2 Comment 13] 12) The supplemental movie seems to show progress in defining clusters in the
cyclostome-centric analysis?? But does not really seem to shed much light into the inner workings of
the programs they use.

The supplementary movie is provided as an additional resource and visualizes the essential idea
behind the algorithm (please see [Nakatani et al., *Genome Res* (2007); Muffato, PhD Thesis (2010)]
and Supplementary Information Section 8 for details). We would like to thank Reviewer 2 again for
reviewing our code in detail.

In addition to the revisions described above, we have revised Supplementary Data 1, because some
data files were missing in the previously submitted version (i.e. we added orthologs from elephant

shark to human, mouse, dog, opossum). We have also updated Figure 6, since one proto-gnathostome
chromosome was missing in the previous version. In addition, we have corrected some spelling errors
(e.g. paralog/paralogue) including Figures 4 and 5, and have replaced raster images with vector
graphics in Figure 4.

**[R2 Comment 14]** Given these large issues I will withhold comment on other specific details (e.g.
discussions of immunology, discussion of ancient hybridization – or alternately incomplete lineage
sorting - in the supplement) for the moment since many details could change depending on how
these above comments are addressed. I am certain that all of these requests can be addressed with
statistical rigor and in a way that facilitates reproducibility. I hope that the comments above make
that easier.

The discussion of hybridizations between genetically diverse subpopulations is only briefly mentioned
in the current revision (due to our response to [R2 Comment 08] above), and a more detailed discussion
in the previously submitted Supplementary Information was deleted in the current revision to avoid
additional rounds of peer-review. To avoid possible confusions between our actual results and the
general discussions, we also made a minor revision to Figure 6 and deleted the information about
immune complexes in the proto-gnathostome genome, so that Results and Discussion sections are
clearly separated and the texts on the origin of the immune complexes are now restricted to Discussion
section.

As discussed in our responses above, essentially there are no changes to our results and conclusions.
Furthermore, we have deleted detailed discussion about hybridization and removed the mention of
immune system genes in Results section. As such we believe there is no need to further revise the
current manuscript, including any of the subjects mentioned by the reviewer in [R2 Comment 14].

**Reviewer #3 (Remarks to the Author):**

I wanted to congratulate the authors for their efforts addressing the extensive comments from all the
reviewers. I sincerely think that a manuscript that was already great has improved a lot.

We thank the reviewer for the kind words. We are pleased that the reviewer finds the manuscript is of
great quality.

**References:**

Braasch, I. et al. The spotted gar genome illuminates vertebrate evolution and facilitates human-teleost
comparisons. *Nat Genet* 48, 427–437 (2016).

Dehal, P. & Boore, J. L. Two rounds of whole genome duplication in the ancestral vertebrate. *PLoS*
*Biol* 3, e314 (2005).

Fontana, F. et al. Evidence of hexaploid karyotype in shortnose sturgeon. *Genome* 51, 113–119 (2008).

Muffato, M. Reconstruction de génomes ancestraux chez les vertébrés. (Université d'Evry-Val
d'Essonne, 2010).

Nakatani, Y., Takeda, H., Kohara, Y. & Morishita, S. Reconstruction of the vertebrate ancestral
genome reveals dynamic genome reorganization in early vertebrates. *Genome Res* 17, 1254–
1265 (2007).

Nakatani, Y. & McLysaght, A. Genomes as documents of evolutionary history: a probabilistic
macrosynteny model for the reconstruction of ancestral genomes. *Bioinformatics* 33, i369–
i378 (2017).

Pascual-Anaya, J. et al. Hagfish and lamprey Hox genes reveal conservation of temporal colinearity
in vertebrates. *Nat Ecol Evol* 2, 859–866 (2018).

Putnam, N. H. et al. Sea anemone genome reveals ancestral eumetazoan gene repertoire and genomic
organization. *Science* 317, 86–94 (2007).

Putnam, N. H. et al. The amphioxus genome and the evolution of the chordate karyotype. *Nature* 453,
1064–1071 (2008).

Simakov, O. et al. Deeply conserved synteny resolves early events in vertebrate evolution. *Nat Ecol*
*Evol* 4, 820–830 (2020).

Simillion, C., Vandepoele, K., Van Montagu, M. C. E., Zabeau, M. & Van de Peer, Y. The hidden
duplication past of Arabidopsisthaliana. *PNAS* 99, 13627–13632 (2002).

Simillion, C., Vandepoele, K., Saeys, Y. & Van de Peer, Y. Building Genomic Profiles for Uncovering
Segmental Homology in the Twilight Zone. *Genome Res* 14, 1095–1106 (2004).

Smith, J. J. et al. Sequencing of the sea lamprey (*Petromyzon marinus*) genome provides insights into
vertebrate evolution. *Nat Genet* 45, 415–421 (2013).

Smith, J. J. & Keinath, M. C. The sea lamprey meiotic map improves resolution of ancient vertebrate
genome duplications. *Genome Res* 25, 1081–1090 (2015).

Smith, J. J. et al. The sea lamprey germline genome provides insights into programmed genome
rearrangement and vertebrate evolution. *Nat Genet* 50, 270–277 (2018).

Symonová, R. et al. Molecular cytogenetic differentiation of paralogs of Hox paralogs in duplicated
and re-diploidized genome of the North American paddlefish (*Polyodon spathula*). *BMC Genet*
18, 19 (2017).

Trifonov, V. A. et al. Evolutionary plasticity of acipenseriform genomes. *Chromosoma* 125, 661–668
(2016).

- Van de Peer, Y. Computational approaches to unveiling ancient genome duplications. *Nat Rev Genet*
5, 752–763 (2004).
- Vandepoele, K., Simillion, C. & Van de Peer, Y. Detecting the undetectable: uncovering duplicated
segments in Arabidopsis by comparison with rice. *Trends Genet* 18, 606–608 (2002).

Reviewers' Comments:

Reviewer #2:

Remarks to the Author:

Please see attachment

The clarity of the manuscript is much improved and I am glad to see that the authors seem to be interested in moving toward open source sharing of their program in line with modern reporting standards. I am also heartened to see that the authors have adopted a hypothesis testing framework that can be used to assess the validity of alternate models. I think there is room for expansion/modification of this test and a few additional points that should be addressed prior to publication. These are outlined in more detail below.

- 1) Code sharing – The authors should discuss this with the editor, ideally the code would be released on GitHub or similar, but it is stated that “the reconstruction software/code is available on request.” due to apparent copyright issues. I looked at the link the authors sent and it seems like there are several solutions to release (<http://numerical.recipes/licenses/redistribute.html>), perhaps it can be better explained why this is not possible in this instance and why Netlib would not be a solution. In general, I am happy to defer to the editor here in determining the correct course of action here.
- 2) Code sharing – I understand that the issue with DoveTail is problematic. In the cases that I am familiar with as a reviewer (Nature, NG, Genome research ..) or Author (one of the Nature Genetics articles cited in their letter) DoveTail data have been reanalyzed using another program or heavily vetted with an orthogonal method. Perhaps the editor could consult with Dr Henry Gee on policies related to non-open source assemblers. I understand there is likely little the authors can do themselves to resolve this issue so I do not anticipate an author response to this bullet.
- 3) I applaud the authors for identifying a statistical test to assess alternatives to the hexaploidization and would strongly recommend that they expand the tests to consider a broader range of possibilities. In addition, the pattern of presumptive paralog retention in lampreys still needs some attention. Specifically, the chromosomes that retain zero (or close to zero) homologs with the presumptive pre-1R ancestor. I apologize for missing the table with these data in the earlier rebuttals, but did not find it in the previous supplement and the information was

not integrated into the main text figure as requested. To make my concern a bit clearer, I am including a quick figure to illustrate the issue (generated from their supplemental table). Specifically, many of the chromosomes annotated as duplicates of a PVC have very small numbers (even zero) of genes that are ohnologous to the reconstructed PVCs, at least as it can be understood in the context of amphioxus homologs (not to mention reciprocal zeros across lamprey segments). Notably, these form a distribution of homolog counts that appears to be distinct from the broader distribution (and does not include six missing chromosomes – or 11 if PVC18 is counted – marked with hashes in Figure 6). Considering the possibility that chromosomal segments with small numbers of homologs are the product of translocations or other small events, the observed deviation “D” as used in their statistical tests should probably be $13+19=32$ if they wanted to perform a minimally conservative test and perhaps also the 5 missing PVG18 chromosomes ($D=37$, $K=18$) if they wanted consider other conservative tests. The tests with $K=5$ seem unjustifiably biased. Furthermore, the authors should permit duplications to occur before, between and after WGDS in a single model, and/or groups of models. Finally, in text reporting of p-values should include all hypotheses, not just a single test with a low p-value. It is not clear why the P reported in the main text was chosen versus say model C which is marginally rejected with $D=13$.

- 4) A similar test can and should be used to test for loss/degradation of chromosomes following their hexaploidization model. Although p-values cannot be directly compared to the above-mentioned tests, this would give the readers a better sense of the degree to which their model fits larger patterns observed in their data. Similar comparisons could also be performed on gnathostome duplications, though these might be difficult to execute given their reconstruction method.
- 5) As part of their argument for downplaying non-WGD mechanisms in this manuscript, the authors state in their reply that “Since there is no such mechanism like whole-genome fission, we conclude that the proto-cyclostome genome was shaped by polyploidization”. In conceptualizing the potential influence of fissions or other small events the authors should more carefully consider karyotype variation in mammals (for example <https://pubmed.ncbi.nlm.nih.gov/3073914/> and <https://pubmed.ncbi.nlm.nih.gov/15004472/>) and the degree to which these mirror both aspects the lamprey karyotype (large numbers of small acrocentrics) and details of the author’s reconstructions.
- 6) Line 189 - “Importantly, the algorithm explores all alternative models including segmental duplications, chromosome duplications/losses, tetraploidization and hexaploidization, under the assumption that duplicated chromosomes share significantly large numbers of paralogues.” I think readers could benefit for a little more detail in their explanation of how duplications are differentiated from fissions or other events, and how/if the relative timing (or simultaneity) of

duplications is assessed in their reconstruction algorithm. These can probably be gleaned from the supplements but I don't think one can expect that the average reader will dig into the supplements.

- 7) Line 264 - "Although several recent studies supported the ancient origin of microchromosomes (for example, a comparison between the chicken and spotted gar genomes suggested that the origin of microchromosomes dates back to the ancestral bony vertebrate)," - perhaps the authors could reference more than one study for the sake of scholarship. I am also a bit concerned about the presentation in that the wording makes it sound like the origins of microchromosomes are currently in question, rather than this simply being a historically interesting discussion.
- 8) Line 279 - "the total length of segments originating from individual proto-gnathostome chromosomes is highly conserved in chicken, spotted gar and elephant shark, suggesting that the ancestral gnathostome already possessed the tiny microchromosomes and the large macrochromosomes" - the authors should be aware that this exact feature was highlighted in comparisons between chicken and gar in Braasch et al and perhaps acknowledge that they are confirming this observation.
- 9) Line 363 - "Indeed, the ratio of retained genes between the two subgenomes in the proto-gnathostome genome is 2.25, which is considerably larger than previously reported ratios of paleo-allopolyploids: 1.47 for Brassica, 1.46 for maize, 1.24 for sorghum, 1.17 for Arabidopsis and 1.35 for *Xenopus laevis*." Can the authors speculate why the protognathostome might have evolved so differently from all of the other allopolyploid examples provided, with on average ~4X higher rates of biased paralog loss? Seemingly this large difference is worth discussing. Can the authors estimate a similar rate for lamprey/cyclostomes under their preferred hypothesis(es)?

Minor/Optional suggestions

- 10) I still do not like the use of the term "cyclostome" as presented in the paper since they use no data whatsoever from hagfish and it lends/justifies a biased interpretation to their results. If the authors address the above comments and still want to use the term it might be acceptable, but they should at least address the caveat that we don't have much information from hagfish yet.
- 11) The authors satisfactorily addressed my previous query about species trees with the revised analyses, although they may like to know that ENSEMBL trees are fit to a species tree in a way that will impinge on the signals that authors are interested in here. The methods have changed since publication of the original lamprey paper, so I sent an inquiry to ENSEMBL, here is their reply

Hi Jeremiah

I'm sorry for the delay in getting back to you. It involved a lot of digging through our code.

Treebest is still using a species tree to guide homology inference. Over the last 10 years what has changed is that our species tree reconstruction method has been improved via our species tree pipeline which integrates the NCBI taxonomy and mash distances between genomes calculated on the whole genome sequence. The tree we use is here:

https://github.com/Ensembl/ensembl-compara/blob/release/103/conf/vertebrates/species_tree.branch_len.nw

All the best

- 12) It is difficult to understand the authors' reluctance to share comparative maps that anchor to defined gene names. While they assert that this is not common practice, my groups have routinely provided these as supplements (e.g. Smith et al 2018 ST3&5; Smith et al 2015 ST2&3). Ultimately this is a courtesy to the average reader who might care about the evolution of specific gene families or wish to delve into other details of the analyses presented, but I am happy to let the authors choose how to handle this.
- 13) The discussion on immune system evolution seems to be essentially a just-so story, which is OK, but perhaps they should also acknowledge that there are gaps to fill in the story. Only a suggestion.

Sincerely,

Jeremiah Smith

Point-by-point response to reviewer's comments

We would like to thank all reviewers and the editor again for their time and effort in reviewing our manuscript and offering detailed and constructive suggestions. The following are point-by-point responses to the comments from Reviewer 2.

The clarity of the manuscript is much improved and I am glad to see that the authors seem to be interested in moving toward open source sharing of their program in line with modern reporting standards. I am also heartened to see that the authors have adopted a hypothesis testing framework that can be used to assess the validity of alternate models. I think there is room for expansion/modification of this test and a few additional points that should be addressed prior to publication. These are outlined in more detail below.

1) Code sharing - The authors should discuss this with the editor, ideally the code would be released on GitHub or similar, but it is stated that "the reconstruction software/code is available on request." due to apparent copyright issues. I looked at the link the authors sent and it seems like there are several solutions to release (<http://numerical.recipes/licenses/redistribute.html>), perhaps it can be better explained why this is not possible in this instance and why Netlib would not be a solution. In general, I am happy to defer to the editor here in determining the correct course of action here. We thank Reviewer 2 for this advice. We will consider using Netlib in the next version of the reconstruction program, and we will be mentioning this in our response to the editor.

2) Code sharing - I understand that the issue with DoveTail is problematic. In the cases that I am familiar with as a reviewer (Nature, NG, Genome research ..) or Author (one of the Nature Genetics articles cited in their letter) DoveTail data have been reanalyzed using another program or heavily vetted with an orthogonal method. Perhaps the editor could consult with Dr Henry Gee on policies related to non-open source assemblers. I understand there is likely little the authors can do themselves to resolve this issue so I do not anticipate an author response to this bullet.

No response required.

3) I applaud the authors for identifying a statistical test to assess alternatives to the hexaploidization and would strongly recommend that they expand the tests to consider a broader range of possibilities. In addition, the pattern of presumptive paralog retention in lampreys still needs some attention. Specifically, the chromosomes that retain zero (or close to zero) homologs with the presumptive pre-1R ancestor. I apologize for missing the table with these data in the earlier rebuttals, but did not find it in the previous supplement and the information was not integrated into the main text figure as requested. To make my concern a bit clearer, I am including a quick figure to illustrate the issue

(generated from their supplemental table).

Specifically, many of the chromosomes annotated as duplicates of a PVC have very small numbers (even zero) of genes that are ohnologous to the reconstructed PVCs, at least as it can be understood in the context of amphioxus homologs (not to mention reciprocal zeros across lamprey segments). Notably, these form a distribution of homolog counts that appears to be distinct from the broader distribution (and does not include six missing chromosomes - or 11 if PVC18 is counted - marked with hashes in Figure 6). Considering the possibility that chromosomal segments with small numbers of homologs are the product of translocations or other small events, the observed deviation "D" as used in their statistical tests should probably be $13+19=32$ if they wanted to perform a minimally conservative test and perhaps also the 5 missing PVG18 chromosomes ($D=37$, K18) if they wanted consider other conservative tests.

We thank Reviewer 2 for this analysis of small chromosomes. As Reviewer 2 discussed, there exist small segments that have only small numbers of orthologs or paralogs, and we agree that those small segments should be treated carefully. In our statistical analysis, we tested whether or not the proto-cyclostome genome was shaped by independent chromosome-number-increasing events. We described such chromosome-number-increasing events as "duplication" events, but it was just for simplifying the discussion. In fact, our probability model deals equally with all types of chromosome-number-increasing events, including duplications, fissions and translocations (if translocations increase the number of chromosomes). Therefore, our statistical test remains unchanged if some of the proto-cyclostome chromosomes were produced by fissions or translocations. In addition, we did not conclude that all proto-cyclostome chromosomes were created by polyploidization events: Some of them (the smaller chromosomes in particular) may be produced by fissions or translocations as Reviewer 2 discussed here.

Regarding Pvc18, we excluded it because it was left as a single proto-cyclostome chromosome due to the large number of lamprey segments. If we include it, we can calculate the

probability for Scenarios A (no WGDs) or D (chromosome-scale duplications followed by one WGD): $p=0.0000000480$ for Scenario A and $p=0.00000118$ for Scenario D. For other scenarios, we need to consider chromosome deletion events in addition to duplications, but our framework does not allow inclusion of deletions (see below for more discussion). To avoid this problem, we set $Y_{18} = \max(1, N)$ for Scenarios B, C and E so that we can calculate the probability as in the case of $K=17$.

In the revised manuscript, we added the tests with $K = 18$ and revised Methods, Supplementary Table 10 and the main text as follows.

“In addition, we evaluated the case of $K = 18$ by setting $Y_{18} = \max(1, N)$, since our model requires $Y_k \geq N$ for all $k = 1, \dots, K$; we also evaluated the case of $K = 5$, $Y = 30$ and $D = 0$ since larger proto-vertebrate chromosomes are more reliable in our reconstruction and the largest five proto-vertebrate chromosomes have multiplicity six.”

Scenario	K	Y	D	N	M	P
A	17	103	13	1	6	0.0000000018
B	17	103	13	2	6	0.0000030304
C	17	103	13	4	6	0.0214209597
D	17	49	6	1	3	0.0000002044
E	17	49	6	2	3	0.0038115884
A	18	104	18	1	6	0.0000000480
B	18	105	17	2	6	0.0000371775
C	18	107	15	4	6	0.0487599825
D	18	50	8	1	3	0.0000011843
E	18	51	7	2	3	0.0067631372
A	5	30	0	1	6	0.0000421035
B	5	30	0	2	6	0.0003120318
C	5	30	0	4	6	0.0049925087
D	5	15	0	1	3	0.0009990010
E	5	15	0	2	3	0.0159840160

“In addition, we confirmed by statistical test (see Methods) that the observed peak of multiplicity (Fig. 3d) is unlikely to have been created by accumulation of chromosome-scale or segmental duplications after one ($P < 4 \times 10^{-5}$) or two ($P < 0.05$) tetraploidization events.”

We also revised an equation because it was written incorrectly in the previously submitted manuscript. We confirmed that the probability calculation was performed correctly using the correct equation below.

Wrong: $S(Y_1, \dots, Y_K) = (Y_1, \dots, Y_K)! \prod_{k=1}^K \Gamma(Y_k) / \Gamma(X_k)$

Correct: $S(Y_1, \dots, Y_K) = (Y_1 - X_1, \dots, Y_K - X_K)! \prod_{k=1}^K \Gamma(Y_k) / \Gamma(X_k)$

The tests with K=5 seem unjustifiably biased.

The tests with $K=5$ were shown because the largest five proto-vertebrate chromosomes are expected

to be more reliable than the other reconstructed chromosomes. In addition, larger chromosomes are especially informative for distinguishing polyploidy and aneuploidy than smaller chromosomes, because chromosome-scale duplication of a larger chromosome is more deleterious than duplication of a smaller chromosome with only a small number of genes as discussed in the germline sea lamprey genome paper [Smith et al. *Nat Genet* (2018)]. Besides, the information of the tests with $K=5$ is helpful if someone wants to manually check the correctness of our probability calculation, because the case of $K=5$ is easier to calculate than the other cases.

Furthermore, the authors should permit duplications to occur before, between and after WGDS in a single model, and/or groups of models.

As Reviewer 2 mentioned, we assumed in our analysis that chromosome-scale duplications occur mainly before, between, or after WGDs in a single model. We did not consider pre-1R chromosome-scale duplications because the proto-gnathostome genome shows that all chromosomes were quadrupled by two rounds of WGDs with no chromosome-scale duplication events (see Supplementary Figs. 6 and 9). For Scenarios D and E, we chose the numbers of chromosomes (Y_1, \dots, Y_{17}) such that the number of chromosome-scale duplications is minimized after the last WGD (i.e., cyclostome-specific WGD). Therefore, our calculation of convergence probability should be a conservative estimate compared with the suggested models that allow chromosome-scale duplications after WGD.

Finally, in text reporting of p-values should include all hypotheses, not just a single test with a low p-value. It is not clear why the P reported in the main text was chosen versus say model C which is marginally rejected with $D=13$.

We chose Model B because (1) the choice of scenarios does not affect our conclusion, (2) Model B was proposed in the sea lamprey germline genome paper [Smith et al. *Nat Genet* (2018)], and (3) Reviewer 2 wrote “1R plus random duplication seems to be a better fit to the observed distribution than 1R+triplication,” in a previous review round.

In the revised manuscript, we revised the main text as follows.

“In addition, we confirmed by statistical test (see Methods) that the observed peak of multiplicity (Fig. 3d) is unlikely to have been created by accumulation of chromosome-scale or segmental duplications after one ($P < 4 \times 10^{-5}$) or two ($P < 0.05$) tetraploidization events.”

4) A similar test can and should be used to test for loss/degradation of chromosomes following their hexaploidization model. Although p-values cannot be directly compared to the above-mentioned tests, this would give the readers a better sense of the degree to which their model fits larger patterns observed in their data.

The strength of our analysis lies in the probability calculation without using the unknown rate of chromosome-scale duplications. On the other hand, if we allow loss of chromosomes in addition to duplications, we cannot calculate the convergence probability without knowing the rates of duplication and loss. Nevertheless, we speculate that the probability of staying close to multiplicity six after six-fold duplication should be larger than the probability of convergence to multiplicity six from lower multiplicity values, especially when nine out of 18 proto-vertebrate chromosomes have multiplicity six in the proto-cyclostome genome.

Although our framework cannot calculate the probability, our proto-cyclostome reconstruction provided genome-scale evidence of six-fold duplication for the first time, and our statistical analysis showed that the previous model is highly unlikely. Thus, we believe that our analysis already made a significant progress toward a better understanding of the origin of cyclostome genomes, considering the previous lack of ancestral genome reconstruction and rigorous statistical analysis.

Similar comparisons could also be performed on gnathostome duplications, though these might be difficult to execute given their reconstruction method.

Due to the chromosome fusion events between 1R and 2R, we cannot assume that the multiplicities of individual proto-vertebrate chromosomes increased independently. Therefore, our framework does not allow calculation of the convergence probability for the proto-gnathostome genome.

Nevertheless, it seems clear from our reconstruction and from previous reconstructions [Sacredot et al. *Genome Biol* (2018); Simakov et al. *Nat Ecol Evol* (2020)] that two WGD events occurred between the proto-vertebrate and the proto-gnathostome, and the two WGDs were separated by several chromosome fusion events as illustrated in Figure 6.

5) As part of their argument for downplaying non-WGD mechanisms in this manuscript, the authors state in their reply that "Since there is no such mechanism like whole-genome fission, we conclude that the proto-cyclostome genome was shaped by polyploidization". In conceptualizing the potential influence of fissions or other small events the authors should more carefully consider karyotype variation in mammals (for example <https://pubmed.ncbi.nlm.nih.gov/3073914/> and <https://pubmed.ncbi.nlm.nih.gov/15004472/>) and the degree to which these mirror both aspects the lamprey karyotype (large numbers of small acrocentrics) and details of the author's reconstructions. We apologize for the careless argument in our previous response. It is indeed important to consider the possibility of karyotype reversal by Robertsonian fusions and centric fissions, but we concluded that it cannot explain the observed paralog distribution in the proto-cyclostome genome. The figure below shows the distribution of paralogs among proto-cyclostome chromosomes duplicated from Pvc1, Pvc10 and Pvc17. In this figure, we used paralogs identified with less stringent criteria so that

we can distinguish chromosome duplications and fissions. We see large numbers of paralogs between all pairs of reconstructed proto-cyclostome chromosomes below, and it is unlikely that these sextuple chromosomes were created by centric fission.

6) Line 189 - "Importantly, the algorithm explores all alternative models including segmental duplications, chromosome duplications/losses, tetraploidization and hexaploidization, under the assumption that duplicated chromosomes share significantly large numbers of paralogs." I think readers could benefit for a little more detail in their explanation of how duplications are differentiated from fissions or other events, and how/if the relative timing (or simultaneity) of duplications is assessed in their reconstruction algorithm. These can probably be gleaned from the supplements but I don't think one can expect that the average reader will dig into the supplements. We thank Reviewer 2 for this advice. We have now moved the description of reconstruction method from Supplementary Information to Methods in the main manuscript.

7) Line 264 - "Although several recent studies supported the ancient origin of microchromosomes (for example, a comparison between the chicken and spotted gar genomes suggested that the origin of microchromosomes dates back to the ancestral bony vertebrate)," - perhaps the authors could reference more than one study for the sake of scholarship.

We sincerely apologize that we forgot to cite one of the most important papers on the origin of microchromosomes [Voss et al. *Genome Res* (2011)], which was co-authored by Reviewer 2. We thank Reviewer 2 for letting us notice it.

We revised Supplementary Note 4 as follows.

"Recent studies tend to support this ancient-origins hypothesis: It was argued that many avian microchromosomes represent ancient chromosomes in the ancestral land vertebrate [Burt D.W.

Cytogenet Genome Res (2002)], and that many proto-gnathostome chromosomes are retained as microchromosomes in the chicken genome without inter-chromosomal rearrangements [Nakatani et al., *Genome Res* (2007)]. The strong conservation in gene content was confirmed in several studies [Voss et al. *Genome Res* (2011); Louis et al. *Brief Func Genomics* (2012); Uno et al. *PLoS ONE* (2012); Venkatesh et al. *Nature* (2014)], but little was known about the origin of chromosomal features that characterize avian microchromosomes (i.e. chromosome length, GC contents, etc). Comparative analysis between the spotted gar genome and chicken genome showed that the chromosomal features already presented in the common ancestor of bony-vertebrate [Braasch et al. *Nat Genet* (2016)], and our analysis with the chromosome-scale elephant shark genome showed that the origin dates back further to the proto-gnathostome, suggesting that those chromosomal features were likely to be associated with the subgenome fractionation after 2R.”

We revised the main text as follows.

“Although several recent studies supported the ancient origin of microchromosomes [Burt D.W. *Cytogenet Genome Res* (2002); Nakatani et al., *Genome Res* (2007); Voss et al. *Genome Res* (2011); Louis et al. *Brief Func Genomics* (2012); Uno et al. *PLoS ONE* (2012); Venkatesh et al. *Nature* (2014); Braasch et al. *Nat Genet* (2016)], it was still unknown ...”.

I am also a bit concerned about the presentation in that the wording makes it sound like the origins of microchromosomes are currently in question, rather than this simply being a historically interesting discussion.

It might be misleading to say that the origin of microchromosomes was just a historically interesting discussion: the argument/evidence that microchromosomes were derived from a subgenome in the proto-gnathostome genome appears only recently in [Simakov et al. *Nat Ecol Evol* (2020)] and in this manuscript.

In order to write a more accurate description of previous studies, we revised Supplementary Note 4 and added citations to several relevant papers (see above for the revised text in Supplementary Note 4).

8) Line 279 - "the total length of segments originating from individual proto-gnathostome chromosomes is highly conserved in chicken, spotted gar and elephant shark, suggesting that the ancestral gnathostome already possessed the tiny microchromosomes and the large macrochromosomes" - the authors should be aware that this exact feature was highlighted in comparisons between chicken and gar in Braasch et al and perhaps acknowledge that they are confirming this observation.

We apologize if the previous text gave an impression that we do not properly acknowledge the previous study by Braasch et al. We are aware of the paper as we explained in our previous response

comment (see [R2 Comment 07]). The major difference is that we reconstructed the proto-gnathostome chromosomes and we discussed chromosomal features in the proto-gnathostome genome (not the ancestral bony vertebrate), using the chromosome-scale elephant shark genome. Thus, our argument about chromosomal features of proto-gnathostome chromosomes is not just a confirmation, but we clarified this point by inserting a description of the work by Braasch et al. in the main text as follows.

“.., it was still unknown (1) if chromosomal features characteristic to modern avian microchromosomes (i.e. high GC-content, high gene density and high recombination rate) were already present in the ancestral gnathostome genome (cf. the chromosomal features were previously reported to be conserved between the spotted gar and chicken genomes [Braasch et al. *Nat Genet* (2016)], ...”

9) Line 363 - "Indeed, the ratio of retained genes between the two subgenomes in the proto-gnathostome genome is 2.25, which is considerably larger than previously reported ratios of paleo-allopolyploids: 1.47 for Brassica, 1.46 for maize, 1.24 for sorghum, 1.17 for Arabidopsis and 1.35 for *Xenopus laevis*." Can the authors speculate why the protognathostome might have evolved so differently from all of the other allopolyploid examples provided, with on average ~4X higher rates of biased paralog loss? Seemingly this large difference is worth discussing. Can the authors estimate a similar rate for lamprey/cyclostomes under their preferred hypothesis(es)?

We thank Reviewer 2 for this suggestion of an interesting analysis. We could speculate that the proto-gnathostome might have had a higher level of sequence divergence and expression bias between the subgenomes than in other allopolyploids, but it seems difficult to verify such speculations. Regarding the bias of gene retention rate in the proto-cyclostome genome, it is difficult to classify proto-cyclostome chromosomes into subgenomes. Therefore, we don't have good answers to these questions at present, although they are interesting questions.

Minor/Optional suggestions

10) I still do not like the use of the term "cyclostome" as presented in the paper since they use no data whatsoever from hagfish and it lends/justifies a biased interpretation to their results. If the authors address the above comments and still want to use the term it might be acceptable, but they should at least address the caveat that we don't have much information from hagfish yet.

Our analysis of lamprey paralogs (and human-lamprey orthologs) showed that there is no clear, genome-wide distinction between chromosome pairs duplicated by 1R and chromosome pairs duplicated by the later event that we call cyclostome-specific hexaploidization. This observation suggests that the later event occurred shortly after 1R, which is indeed cyclostome-specific.

In the revised manuscript, we added the following sentence in the legend of Figure 1, in

which cyclostome-specific duplication scenarios are described.

“It is presently considered that the hagfish and lamprey lineages share the same duplication history [Pascual-Anaya et al. *Nat Ecol Evol* (2018)], but this argument should eventually be confirmed by sequencing the hagfish genome.”

11) The authors satisfactorily addressed my previous query about species trees with the revised analyses, although they may like to know that ENSEMBL trees are fit to a species tree in a way that will impinge on the signals that authors are interested in here. The methods have changed since publication of the original lamprey paper, so I sent an inquiry to ENSEMBL, here is their reply

Hi Jeremiah

I'm sorry for the delay in getting back to you. It involved a lot of digging through our code.

Treebest is still using a species tree to guide homology inference. Over the last 10 years what has changed is that our species tree reconstruction method has been improved via our species tree pipeline which integrates the NCBI taxonomy and mash distances between genomes calculated on the whole genome sequence. The tree we use is here:

https://github.com/Ensembl/ensembl-compara/blob/release/103/conf/vertebrates/species_tree.branch_len.nw

All the best

We thank Reviewer 2 for this information.

12) It is difficult to understand the authors' reluctance to share comparative maps that anchor to defined gene names. While they assert that this is not common practice, my groups have routinely provided these as supplements (e.g. Smith et al 2018 ST3&5; Smith et al 2015 ST2&3). Ultimately this is a courtesy to the average reader who might care about the evolution of specific gene families or wish to delve into other details of the analyses presented, but I am happy to let the authors choose how to handle this.

We had already provided such information in Supplementary Data 1. Our description of the Supplementary Data 1 might have been unclear, so we revised the text in the Data availability section and clarified that information on orthologs, paralogs and gene names in individual chromosomal segments are included in Supplementary Data 1.

13) The discussion on immune system evolution seems to be essentially a just-so story, which is OK, but perhaps they should also acknowledge that there are gaps to fill in the story. Only a suggestion. We thank Reviewer 2 for this suggestion, and we apologize if our discussion was overly assertive. Our discussion shows how the previous hypotheses can be updated or revised based on our ancestral

genome reconstruction, and we do not think that the origin of adaptive immunity is resolved completely.

To address the concern raised by Reviewer 2, we deleted phrases about adaptive immunity from the concluding sentence in Abstract and concluding sentence of Discussion which now read as follows.

Abstract: “Thus, our reconstructions reveal the major evolutionary events and offer new insights into the origin and evolution of vertebrate genomes.”

Discussion: “The resulting model offers unique perspectives on the origin and evolution of vertebrate genomes.”

Minor revisions.

In addition to the revisions described above, we have edited the manuscript for fixing minor errors as follows.

1. The asterisk symbol was fixed in Figure 1.
2. A phrase about thin vertical lines was deleted in the legend of Figure 2, because thin vertical lines were already deleted in the figure.
3. Figure 6 was fixed because there was an unnecessary horizontal line at the bottom of the figure in the previously submitted manuscript.
4. Supplementary Table 9 was fixed, because we presented the updated table in our response to the previous comments from Reviewer 2 but the table was not updated in the previous manuscript.
5. Fonts in Supplementary Tables 11, 12 and 13 were updated (table contents are the same).
6. We added an explanation of S and G_S after Equation 1 in Methods for improved readability. We revised the name of Algorithm 1 as CVB0, following relevant papers on topic models.

We hope that our responses and revisions described above satisfactorily addressed all the concerns raised by Reviewer 2.

References:

- Braasch, I. et al. The spotted gar genome illuminates vertebrate evolution and facilitates human-teleost comparisons. *Nat Genet* **48**, 427–437 (2016).
- Burt, D. W. Origin and evolution of avian microchromosomes. *Cytogenet Genome Res* **96**, 97–112 (2002).
- Louis, A., Roest Crolius, H. & Robinson-Rechavi, M. How much does the amphioxus genome represent the ancestor of chordates? *Brief Func Genomics* **11**, 89–95 (2012).

- Nakatani, Y., Takeda, H., Kohara, Y. & Morishita, S. Reconstruction of the vertebrate ancestral genome reveals dynamic genome reorganization in early vertebrates. *Genome Res* **17**, 1254–1265 (2007).
- Pascual-Anaya, J. et al. Hagfish and lamprey Hox genes reveal conservation of temporal colinearity in vertebrates. *Nat Ecol Evol* **2**, 859–866 (2018).
- Sacerdot, C., Louis, A., Bon, C., Berthelot, C. & Roest Crolius, H. Chromosome evolution at the origin of the ancestral vertebrate genome. *Genome Biol* **19**, 166 (2018).
- Simakov, O. et al. Deeply conserved synteny resolves early events in vertebrate evolution. *Nat Ecol Evol* **4**, 820–830 (2020).
- Smith, J. J. et al. The sea lamprey germline genome provides insights into programmed genome rearrangement and vertebrate evolution. *Nat Genet* **50**, 270–277 (2018).
- Uno, Y. et al. Inference of the protokaryotypes of amniotes and tetrapods and the evolutionary processes of microchromosomes from comparative gene mapping. *PLoS ONE* **7**, e53027 (2012).
- Venkatesh, B. et al. Elephant shark genome provides unique insights into gnathostome evolution. *Nature* **505**, 174–179 (2014).
- Voss, S. R. et al. Origin of amphibian and avian chromosomes by fission, fusion, and retention of ancestral chromosomes. *Genome Res* **21**, 1306–1312 (2011).